

# A 2700-year annual timescale and accumulation history for an ice core from Roosevelt Island, West Antarctica

Mai Winstrup[1], Paul Vallelonga[1], Helle A. Kjær[1], Tyler J. Fudge[2], James E. Lee[3], Marie H. Riis[1], Ross Edwards[4], Nancy A.N. Bertler[5,6], Thomas Blunier[1], Ed J. Brook[3], Christo Buizert[3], Gabriela Ciobanu[1], Howard Conway[2], Dorthe Dahl-Jensen[1], Aja Ellis[4], B. Daniel Emanuelsson[5,6], Elizabeth D. Keller[6], Andrei Kurbatov[7], Paul Mayewski[7], Peter D. Neff[8], Rebecca Pyne[6], Marius F. Simonsen[1], Anders Svensson[1], Andrea Tuohy[5,6], Ed Waddington[2], Sarah Wheatley[7]

1: Centre for Ice and Climate, Niels Bohr Institute, University of Copenhagen, Copenhagen, Denmark

2: Earth and Space Sciences, University of Washington, Seattle, WA, USA

3: College of Earth, Ocean, and Atmospheric Sciences, Oregon State University, Corvallis, OR, USA

4: Physics and Astronomy, Curtin University, Perth, Western Australia, Australia

5: Antarctic Research Centre, Victoria University Wellington, Wellington, New Zealand

6: GNS Science, Lower Hutt, New Zealand

7: Climate Change Institute, University of Maine, Orono, Maine, USA

8: Earth and Environmental Sciences, University of Rochester, Rochester, NY, USA

*Correspondance to: Mai Winstrup (mai@gfy.ku.dk)*

## Abstract

We present a 2700-year annually resolved timescale for the Roosevelt Island Climate Evolution (RICE) ice core, and reconstruct a snow accumulation history for the coastal sector of the Ross Ice Shelf in West Antarctica. The timescale was constructed by identifying annual layers in multiple ice-core impurity records, employing both manual and automated counting approaches, and constitutes the top part of the Roosevelt Island Ice Core Chronology 2017 (RICE17). The maritime setting of Roosevelt Island results in high sulfate influx from sea salts and marine biogenic emissions, which prohibits a routine detection of volcanic eruptions in the ice-core records. This led to the use of non-traditional chronological techniques for validating the timescale: RICE was synchronized to the WAIS Divide ice core, on the WD2014 timescale, using volcanic attribution based on direct measurements of ice-core acidity, as well as records of globally-synchronous, centennial-scale variability in atmospheric methane concentrations.

The RICE accumulation history suggests stable values of 0.25 m water equivalent (w.e) per year until around 1260 CE. Uncertainties in the correction for ice flow thinning of annual layers with depth do not allow a firm conclusion about long-term trends in accumulation rates during this early period but from 1260 CE to the present, accumulation rate trends have been consistently negative. The decrease in accumulation rates has been increasingly rapid over the last centuries, with the decrease since 1950 CE being more than 7 times greater than the average over the last 300 years. The current accumulation rate of $0.22 \pm 0.06$ m w.e $y^{-1}$ (average since 1950 CE, $\pm 1\sigma$) is 1.49 standard deviations (86th percentile) below the mean of 50-year average accumulation rates observed over the last 2700 years.



# 1. Introduction

Accurate timescales are fundamental for reliable interpretation of paleoclimate archives. A wide variety of dating methods are available for producing ice-core chronologies, with the technique to be applied primarily depending on data availability. Data availability is influenced by the analytical capabilities at the time of measurement as well as the limitations specific to the ice-core site, with important site parameters including core quality, accumulation rates, temperature, ice flow effects and surface snow remobilization.

Where annual snow deposition is sufficiently high and reasonably regular throughout the year, seasonal variations in site temperature and atmospheric impurity deposition lead to annual cycles in the ice-core water isotope and impurity records (Rasmussen et al., 2014). By identifying and counting these annual cycles, a high-resolution annual-layer-counted ice-core timescale can be produced (Sigl et al., 2016; Steig et al., 2005; Svensson et al., 2008; Taylor et al., 2004). This technique is commonly employed for producing ice-core timescales at sites with moderate to high snow accumulation, such as central Greenland, coastal Antarctica and mountain glaciers, but has been demonstrated also for ice from Dome Fuji on the low-accumulation East Antarctic Plateau (Svensson et al., 2015). Annual-layer-counted ice-core timescales have traditionally been obtained by manual counting, but this task can now be performed using machine-learning algorithms for pattern recognition (Winstrup et al., 2012).

Where possible, identification of annual layers allows the development of a high-resolution ice-core chronology, but unless constrained by other data, the uncertainty of such timescale will increase with depth, as the number of uncertain layers encountered accumulate to produce some age uncertainty (Andersen et al., 2006; Rasmussen et al., 2007). Established marker horizons found in the ice-core records can be used to evaluate the precision of a layer-counted timescale, or, alternatively, to constrain the timescale. Such marker horizons include layers of enhanced radioactivity resulting from nuclear bomb tests in the 1950s and 1960s (Arienzo et al., 2016), sulfuric acids (Hammer, 1980) and/or tephra (Abbott et al., 2012) from volcanic eruptions, and layers of enhanced flux of cosmogenic radionuclides resulting from abrupt changes in the Earth's magnetic field (Raisbeck et al. 2007), short-lived cosmic events (Sigl et al., 2015), as well as decadal variability in solar activity (Muscheler et al. 2014). Given the global nature of many of these events, they form strata for synchronization across ice cores and, in some cases, from ice cores to other climate archives such as tree-ring records (Sigl et al. 2015) and lake and marine sediment cores (Davies et al., 2010).

Ice cores can also be stratigraphically matched using records of atmospheric composition of trapped air. Mixing in the atmosphere causes variations in atmospheric composition of relatively long-lived gases to be globally synchronous, and large-scale changes in gas composition associated with abrupt climate events have been used as bi-polar ice-core stratigraphic markers (Bender et al., 1994; Blunier et al., 1998; Blunier and Brook, 2001; EPICA community members, 2006). Globally-synchronous multi-decadal fluctuations in atmospheric composition also exist during periods of stable climate (Bender et al., 1994). Recent improvements in measurement methods allow these to be measured far back in time (Mitchell et al., 2013), whereby the ice-core gas records can be used for synchronizing ice cores on sub-centennial timescales. During the snow densification process, there is a continuous transfer of contemporary air down to the gas lock-in depth, resulting in an offset between the ages of ice and gas at a given depth (Schwander and Stauffer, 1984). This age difference, Δage, and how it changes with depth, can be calculated from firn densification models (Herron and Langway, 1980), whereby the ice-core gas records also can be used to synchronize ice-core records measured on the ice matrix (Buizert et al., 2015).



Annually-resolved ice-core chronologies provide long-term reconstructions of the annual
snowfall accumulation (Alley et al., 1993; Dahl-Jensen et al., 1993). Based on the ice-core
chronology, annual layer thicknesses at a given depth can be converted to past accumulation
rates by applying corrections due to density changes during the transformation from snow to
ice (Herron and Langway, 1980), and subsequent thinning of annual layers caused by ice flow
(Nye, 1963). Reconstructions of past snow accumulation on glaciers and ice sheets are
important for improving our understanding of natural fluctuations in regional accumulation
rates and their dependency on climate. This knowledge is essential to accurately evaluate the
current and future surface mass balance of glaciers and ice sheets, a critical and currently
under-constrained factor in sea level assessments (Shepherd et al., 2012).
Ice cores provide long-term absolute accumulation histories from a single location. They
thereby provide a context for other observations of past and current accumulation rates: Direct
surface observations allow detailed assessment of current spatial accumulation patterns
(Frezzotti et al., 2007) but do not inform about past variability, and internal layers of equal
age (isochrones) observed using ice-penetrating radar provide spatial coverage of past relative
accumulation patterns (Medley et al., 2013), but additional information is required for
converting these to absolute accumulation rates. Regional-scale reconstructions of past
surface mass balance can be obtained by coupling spatial information from climate models
and surface observations with point estimates of past accumulation rates derived from ice
cores (Frezzotti et al., 2013; Frieler et al., 2015). Within the PAGES network, the
Antarctica2k consortium seeks to produce Antarctica-wide reconstructions of temperature and
ice-core snow accumulation for the past 2000 years. The ice-core accumulation record
developed here is a contribution to the Antarctica2k network (Stenni et al., 2017; Thomas et
al., 2017) from an otherwise poorly-constrained sector of the Antarctic continent.

## 25  2. Site characteristics

From 2010 to 2014, the Roosevelt Island Climate Evolution (RICE) project extensively
sampled snow and ice at the summit of Roosevelt Island (Bertler et al., 2017). Roosevelt
Island is located within the eastern part of the Ross Ice Shelf (Fig. 1), from which it protrudes
as an ice dome that is grounded 200 m below sea level. The annual mean temperature derived
for Roosevelt Island from the ERA-Interim (ERAi) reanalysis data set of the European Centre
for Medium-Range Weather Forecasts (Dee et al., 2011) is -27.4±2.4° C. An automated
weather station at the RICE camp measured the current accumulation rate to be 0.20 m w.e
$yr^{-1}$, consistent with previous estimates of 0.16-0.20 m w.e $yr^{-1}$ (Conway et al., 1999; Herron
and Langway, 1980). An array of 144 snow stakes monitored from 2010-2013 and covering a
200 $km^2$ region showed that snow accumulation across Roosevelt Island Dome ranged
spatially from 0.09-0.30 m w.e $yr^{-1}$ (Bertler et al., 2017). The drill site has an altitude of 550
m above sea level, and the ice thickness is 765 m.
The RICE project builds on preliminary site studies conducted in the 1970s and 90s. As part
of the US Ross Ice Shelf Project (RISP; 1973-1979) 95 firn cores were collected across the
Ross Ice Shelf, including two firn cores from Roosevelt Island summit, for evaluation of
present accumulation rates and its spatial variability over the region (Clausen et al., 1979).
The shortest (11 m) firn core from near the summit was measured for water isotopes and total
β-activity; here we refer to it as RID-75 (Table 1).
In 1997, ice-penetrating radar surveys across Roosevelt Island demonstrated extensive
internal layering which were assumed to be isochrones. Notably, the isochrones bend up
underneath the ice divide forming a so-called Raymond bump (Conway et al., 1999), a
glaciological feature indicative of stable ice divide flow over long (kyr) timescales (Raymond,



1983). Model reconstructions based on the shape and size of the Raymond bump suggest that
the current divide-type flow in the Roosevelt Island ice cap started between 2300 and 4200
years ago, with the best fit corresponding to onset of divide flow around 3000 years ago
(Martín et al., 2006). The stable flow regime since then facilitates the interpretation of past
accumulation rates from the annual layering in the RICE ice core. It has been hypothesized
that divide flow started as the grounding line in the Ross Sea Embayment moved south of
Roosevelt Island during the last deglaciation (Conway et al., 1999). Roosevelt Island is an
ideal location to investigate the timing of grounding-line retreat in the eastern Ross Sea, and
also to investigate changes in climate subsequent to the retreat of the grounding line.

The RICE ice cores were drilled at the present location of the Roosevelt Island ice divide
(Fig. 1), and less than 1 km from where the two RISP firn cores were drilled in 1974/75
(Clausen et al., 1979). The RICE deep core was drilled in austral summer 2011/12 and
2012/13. During the first season, the core was dry-drilled down to 130 m, and subsequently
extended during the second season using an Estisol-240/Coasol drilling fluid mixture.
Bedrock was reached at 764.6 m depth. The upper 344m of the core spans the last 2700 years;
the period for which an annual-layer counted timescale can be constructed. In addition to the
RICE deep core, several shallow cores were drilled in the vicinity. During the 2012/13 field
season, a 20m firn core (RICE-12/13-B) was drilled near the main core. This shallow core has
here been used here to preserve continuity of the ice-core records up to the 2012/13 snow
surface. A summary of the various relevant firn and ice cores collected at Roosevelt Island
summit is provided in Table 1.

In this paper, we first construct a layer-counted chronology for the RICE core, and validate it
against the WAIS Divide ice core WD2014 chronology (Sigl et al., 2015; Sigl et al., 2016).
We then use the annual layer thickness profile to reconstruct the accumulation history over
the last 2700 years, and consider the implications for mass balance in the Ross Ice Shelf
region. According to ERAi climate reanalysis data, precipitation rates at Roosevelt Island are
strongly influenced by the Amundsen Sea Low and associated ridging, and anti-correlated
with those in Ellsworth Land and the Antarctic Peninsula (Bertler et al., 2017; Emanuelsson
et al., 2017b; Yuan and Martinson, 2000). These differences illustrate the importance of high
spatial and temporal information about past accumulation when reconstructing regional mass
balance. With its coastal location and low altitude, Roosevelt Island represents a quite
different climatic setting than the inland, high-altitude Antarctic ice-core sites; we expect that
the RICE core is more representative of climate evolution in the coastal Victoria Land sector
of Antarctica, as defined by Thomas et al. (2017). As one of a very few ice cores within this
sector of Antarctica, the RICE accumulation history is particularly relevant for calculations of
changes in Antarctic mass balance over time.

## 3. Methods

### 3.1. Ice core processing and Continuous Flow Analysis (CFA)

The RICE ice cores were processed and analyzed at the GNS Science ice-core facility in
Lower Hutt, New Zealand. The cores were cut longitudinally to produce, among others, a
15x35 mm triangular piece for water isotope analysis and two 35x35 mm square sticks for
CFA (Fig. 2). The second CFA piece was for use in case the core quality of the primary piece
was compromised or for repeat measurements to test measurement accuracy and system
stability. The remaining core was used for measurements of gas composition, dust, tephra,
biology and physical properties, with a quarter of the ice core left as an archive piece.



In parallel with ice core cutting and processing, CFA and electrical conductivity
measurements (ECM) were carried out during measurement campaigns in 2012-2014 (Table
1). ECM was measured using a low-power hand-held ECM instrument (Icefield Instruments
Inc.) directly on the ice-core surfaces after the initial cutting of the ice core. In August 2012,
the uppermost section (8.57-40 m) of the RICE main core was processed and analyzed using
the GNS melter system, in which melt water was continuously collected into vials for
subsequent discrete analysis of various ion concentrations. In May 2013, this set-up was
replaced by an expanded version of the Copenhagen CFA system (Bigler et al., 2011),
providing high-resolution continuous measurements of liquid conductivity, calcium ($Ca^{2+}$),
insoluble dust particles, acidity ($H^+$), and black carbon (BC), as well as stable water isotopes
($\delta D$, $\delta^{18}O$) and methane gas concentrations. Using this system, the RICE-12/13-B firn core
was analyzed. Next, the RICE main core was melted and analyzed from 40 m to 475 m, at this
depth the ice brittle zone was reached. Subsequently, repeat measurements of the top section
(8.57-40 m) of the main core were made using the second, parallel CFA stick. Ice from the
brittle zone, encountered from 475 m depth to bedrock, was allowed to relax for an additional
year before processing and CFA measurements. The last 2.6 m of the main core just above
bedrock (762 to 764.6 m) was reserved for analysis of DNA in entrained basal sediments.
Primary adaptations to the Copenhagen CFA system involved: 1) Depth assignment via a
digital decoder using 1 s time markers; 2) Inclusion of three fraction collectors for discrete
sample analyses by, respectively, ion chromatography (IC), Inductively-Coupled Plasma
Mass Spectrometry (ICP-MS), including measurements of $^{239}Pu$ using an ICP-SFMS
technique (Gabrieli et al., 2011), and a Los Gatos Research (LGR) water isotope analyzer
instrument; 3) Continuous analysis of stable water isotopes ($\delta^{18}O$, $\delta D$) using a LGR analyzer
(Emanuelsson et al., 2015); 4) Black carbon analysis by a Single Particle Soot Photometer
(Droplet Measurement Technologies, Boulder, CO; DMT SP2) using the method from
McConnell et al. (2007); 5) Acidity measurements using an optical dye method (Kjær et al.,
2016); and 6) Continuous methane concentration analysis using a Picarro Cavity Ring-Down
Spectroscopy (CRDS) instrument following the method reported by Stowasser et al. (2012).
Discrete samples of meltwater from the melthead outer section were collected for quality
assurance of the continuous stable water isotope measurements. See Figure 3 for a diagram of
the CFA system set-up.
The ice was melted at a rate of 3 cm $min^{-1}$, producing approximately 16.8 mL contamination-
free water and gas mixture per minute of melting. Air bubbles were separated in a debubbler
and sent to the Picarro CRDS instrument for methane analysis. Each minute, 5 mL meltwater
was directed to each of two fraction collectors (IC and ICP-MS aliquots) and 1.1 mL was
used for continuous measurements of water isotopes (0.05 mL) and black carbon (1.05 mL)
by the LGR and DMT SP2 instruments. The remaining 1.8 mL was sent to flow-through
liquid conductivity and insoluble particle analyzers (Bigler et al., 2011), and then split for
continuous analysis of soluble calcium (Traversi et al., 2007) and acidity (Kjær et al., 2016).
A third fraction collector was used to collect discrete samples for water isotopes from the
melthead overflow lines. On average, 20 meters of ice were melted during a 24-hour period,
including measurements, calibrations and routine maintenance. Calibrations for water
isotopes, calcium, acidity and black carbon was carried out before and after each melting run,
which consisted of the continuous analysis of 3x1m long ice rods. Calibrations for methane,
based on standard gases with methane concentrations corresponding to glacial and
preindustrial Holocene levels, were carried out twice daily.
The resulting CFA chemistry records are very densely sampled (1 data point per mm),
however mixing in the tubing as the meltwater sample travelled from melthead to the
analytical systems caused the true depth resolution of the system to be significantly less than



the sampling resolution. This was especially the case for the RICE CFA set-up due to the
relatively small fraction of total meltwater directed to the continuous measurement systems.
Following the technique used in Bigler et al. (2011), actual depth resolution for the CFA
measurements range from 0.8 cm (for conductivity) to 2.4 cm (for calcium) (see
supplementary Table S1).

## 3.2. Constructing the Roosevelt Island Ice Core Chronology, RICE17, for the last 2700 years

The Roosevelt Island Ice Core Chronology 2017, RICE17, is constructed using multiple
approaches, as necessitated by changing properties and availability of data with depth. This
section describes the methodology used to construct the most recent 2700 years of the
RICE17 chronology, the period for which annual layer counting is feasible. For older sections
of the core, prior to the annual-layer counted interval, RICE17 is constructed by gas matching
to the WD2014 chronology for the WAIS Divide ice core, as reported in Lee et al. (2017).

### 3.2.1. Overview of the annual-layer counting strategy
For the uppermost section (0-42.5 m) of the RICE core, manual identification and counting of
annual layers was applied to records of water isotopes and ice impurities from the RICE main
core as well as the RICE-12/13B shallow core, where available. In this section, nuclear fall-
out and other distinctly identifiable marker horizons (Table 2, Sect. 3.2.3) in the ice-core
records were used to constrain the timescale. Below 42.5 m (1884 CE), development of the
timescale was augmented using the *StratiCounter* layer-counting algorithm (Winstrup et al.,
2012) using multiple CFA impurity records from the RICE main core. Unequivocal
identification of volcanoes in the RICE records was difficult, and consequently well
established volcanic marker horizons prior to 1884 CE (Sigl et al., 2015) were not used to
constrain the timescale. The only exception, in which an unequivocal volcanic event was
identified below 42.5 m depth, was a tephra layer at 165 m depth that is geochemically linked
to the Pleiades volcanic group. This tephra has been unequivocally identified in other West
Antarctic ice core records and dated to 1252±2 CE, and we used it to help constrain the
*StratiCounter* algorithm. Older than 1252 CE, RICE17 is a fully independent layer-counted
timescale. The chronology was validated by synchronization to the WAIS Divide WD2014
chronology by volcanic markers and methane matching.
The layer-counted part of RICE17 stops at 343.72 m (700 BCE). At this depth, the annual
layers are too thin (<6 cm) for reliable layer identification in data produced by the RICE CFA
set-up. Excellent agreement between the layer-counted timescale and the independent gas-
derived age at this depth allows us to produce the combined Roosevelt Island Ice Core
Chronology 2017, RICE17, by stitching the two together without any further adjustments.

### 3.2.2. Manual layer counting with constraints (0 - 42.5 m; 2012 - 1884 CE)
The top 42.5 m of the RICE17 chronology was obtained by manually counting annual layers
in the combined set of discretely-measured IC and ICP-MS data, where available, as well as
the continuous water isotope and chemistry records produced by the RICE CFA system. The
RICE main core starts at 8.57 m depth, so the top part of the timescale is based exclusively on
the RICE-12/13B shallow core. At 12.3 m, both cores display a distinct peak in their isotope
profiles, showing that they can be spliced directly without need for any depth adjustments.
Layer marks for the top 12.3 m were placed according to their depths in the RICE-12/13B
shallow core; below 12.3 m, layer marks are in reference to depth in the main core. In the
overlap section (8.57-19.41 m), we used the combined data set from both cores, thereby
reducing the risk of timescale errors caused by core breaks or bad data sections.



Layer counting in this section of the core relied predominantly on annual signals in non-sea-
salt sulfate (nss-SO$_4^{2-}$), acidity (H$^+$) and iodine, as these records displayed the most consistent
annual signals (Fig. 4). For the top 20 m, water isotope records contributed to annual layer
interpretations but smoothing through diffusion of water molecules in the firn causes the
annual signal to diminish with depth, such that annual layers were no longer visible below 20
m. Summers were identified as periods with high stable isotope ratios, high concentrations of
nss-SO$_4^{2-}$ and associated acidity [originating from phytoplankton activity in the surrounding
ocean during summer (Legrand et al., 1991; Udisti et al., 1998)], and low iodine
concentrations [due to summertime photolysis of iodine in the snowpack (Frieß et al., 2010)].
Several other records also displayed annual variability, but much less reliably.
The detailed annual structure of the accumulation record derived from the layer counts
depends on the approach taken to place the layer marks. For this most recent period, layer
marks were placed as close as possible to the peak value in all data series with clear summer
peaks (isotope, sulfur/sulfate, acidity). The summer peaks are assigned to the beginning of the
calendar year (January 1$^{st}$) as supported by ERAi near-surface air temperature and Amundsen
Sea Low (ASL) 500-hPa geopotential height (Emanuelsson et al., 2017b).
A confidence interval was assigned to the timescale by classifying layers as certain or
uncertain. Some uncertain layers were counted as a layer in the timescale, while others were
not, with the upper/lower bound of the age confidence interval being increased corresponding
to this choice. Classification of certain and uncertain layers in the timescale was achieved
while accounting for constraints imposed by timescale tie points (section 3.2.3). We constrain
the timescale to fit these age constraints, but conservatively estimate the uncertainty of the tie-
point ages to be ±1 yr, thereby allowing for small uncertainties in e.g. precise deposition time
of volcanic material in the ice-core record. In this way, a most likely timescale was
constructed along with an uncertainty estimate, which we interpret as the 95% confidence
interval of the age at a given depth, similar to that obtained from automated counting deeper
in the core (see section 3.2.4). Figure 4 demonstrates the approach used to select certain and
uncertain annual layers in the RICE core records.

### 29   3.2.3.  Age constraints and timescale validation (0 - 42.5 m; 2012 - 1884 CE)

**30   3.2.3.1 The 1974/75 snow surface**
The upper 42.5 m of the RICE17 chronology was tied to several unambiguous marker
horizons found in the ice-core records. The uppermost tie-point was established by
successfully matching the top of the previously-drilled RID-75 firn core with the RICE main
core by comparison of their respective water isotope profiles (Fig. 5). The first tie-point in the
RICE17 chronology is January 1975, at a depth of 14.62 m which corresponds to the snow
surface in the RID-75 firn core (Table 2).

**37   3.2.3.2 Nuclear bomb peaks**
The RID-75 core was previously annually dated using the water isotope profile along with
constraints from peaks in total specific β-activity from atmospheric nuclear bomb tests in the
mid-1950s and 1960s (Clausen et al., 1979). Isotope matching of the two cores is consistent
with high-resolution $^{239}$Pu measurements on the RICE main core (Fig. 5).
Fallout from atmospheric nuclear bomb testing causes a significant increase in plutonium
levels (Table 2), starting from very low background levels at 22 m and reaching peak values
at 21.6 m in the RICE main core. This peak can be attributed to the Castle Bravo Operation,
when the US military detonated a very large hydrogen bomb at Bikini Atoll, Marshall Islands.
The operation took place in March 1954 and caused unexpectedly large amounts of nuclear
fall-out globally during the following year (see e.g. Arienzo et al. (2016)). The abrupt increase





in [239]Pu-fallout makes this horizon a reliable age marker, and it was used as an age constraint
during development of the RICE17 timescale.
Several subsequent peaks in the [239]Pu and β-activity records can be attributed to successive
nuclear tests and subsequent test ban treaties (Table 2). However, these peaks are broader and
less distinct, and hence were only used for timescale validation. Devastating effects of the
Castle Bravo operation led to a moratorium agreement between Britain, USA, and the Soviet
Union banning atmospheric nuclear testing, in effect by November 1958. The effect is slowly
decreasing values of [239]Pu in the RICE core, reaching minimum values in the early 1960s.
Following an uptake of Soviet atmospheric nuclear testing in 1961-62, the [239]Pu values
increased in 1962 (19.8 m), and increased levels of β-activity and [239]Pu concentrations
continued until 1966/67. A second broad peak starts summer 1970/71. This period of
enhanced radioactivity has been attributed to French bomb tests at Mururoa (21ºS, 137ºW)
taking place from 1969-1971 (Clausen et al., 1979); the 1971 peak in [239]Pu is the last episode
of enhanced [239]Pu levels in the RICE core.
**3.2.3.3 Recent volcanic eruptions**
Clearly-identifiable volcanic horizons formed the basis for establishing the recent part of the
RICE17 chronology. A tephra layer located at 18.1-18.2 m has a geochemical composition
similar to tephra from Raoul Island, New Zealand, which erupted from November 1964 to
April 1965 (Table 2). Tephra of similar composition has also been found in the WAIS Divide
core corresponding to late 1964. This tephra layer is located in early 1965 CE according to the
RICE17 chronology. Additionally, sulfate deposition from the eruptions of Santa Maria (1902
CE, RICE depth 37.45 m) and Krakatau (1883 CE, RICE depth 42.34 m) were used to
constrain the lower part of the manually-counted timescale, with ages in accordance with
WD2014.
We were unable to confidently pinpoint acid deposition from several other large volcanic
eruptions during recent times. This was the case for e.g. the Pinatubo eruption (1991 CE),
although a small peak in the conductivity and acidity profiles does fit the WD2014 age of
acidic fallout from this eruption. Similarly, we find potential volcanic acid deposition in the
1963 CE stratum, which may be affiliated with the Agung eruption (1963 CE).
Manual layer counting stopped at 42.3 m depth, where an acidic horizon indicates significant
sulfate deposition from the Krakatau eruption, Indonesia (August 1883 CE). According to the
WD2014 chronology, the sulfate horizon corresponding to this eruption was deposited
starting early 1884 CE, and continuing for several years thereafter. Accordingly, an age of
1884 CE was attributed to this horizon. This strata was used as starting point for the
automated counting routine, which generated the remaining part of the annual-layer counted
RICE17 timescale.
**3.2.4.  Automated annual layer counting (42.5 - 343.7m; 1884 CE – 700 BCE)**
For the section 42.5-343.7 m (1884 CE – 700 BCE), the RICE17 annual layer-counted
timescale was established using the *StratiCounter* algorithm (Winstrup et al., 2012), extended
to interpret the annual signal based on multiple chemistry series in parallel (Winstrup, 2016).
*StratiCounter* is a Bayesian technique built on machine-learning methods for pattern
recognition, and it uses a Hidden Markov Model (HMM) framework (Rabiner, 1989).
*StratiCounter* infers the most likely timescale by counting annual layers in overlapping
batches of data stepwise down the ice core, with the length of each batch corresponding
approximately to a 50-year section. For each batch, the most likely annual layering is
calculated by combining *a priori* information on layer appearance – initially based on a small
section of the data used as a "training set" – with the observed annually-resolved data series.



*StratiCounter* tworks in an iterative manner, updating the *a priori* information (the annual
layer parameters) with each iteration until convergence is reached using optimal layering
parameters. The algorithm then proceeds to the next 50-year section of data. In this way, the
annual layers are allowed to slightly change characteristics with depth, as the layer
thicknesses decrease and the annual signals become smoother. Finally, a second source of
prior information (a set of generalized annual layer templates, which carries information on
e.g. relative peak phasing) is updated based on the retrieved layering, and *StratiCounter* re-
evaluates the entire timescale using this new set of templates (Winstrup, 2016). The output of
*StratiCounter* is a probability distribution of the age as function of depth, based on which the
most likely age can be estimated, as well as a 95% confidence bound on the age estimate. The
confidence interval assumes the timescale errors to be non-biased, so that uncertainties in
layer assignment partly cancel out over longer distances, giving rise to a non-linear increase
in uncertainty with depth.
For this deeper part of the RICE core, annual cycles were most reliably observed in the high-
resolution CFA measurements of black carbon (BC). The *StratiCounter* algorithm was
therefore tuned to select peaks in black carbon as the annual marker, while noting that it
generally peaks a little earlier in the year than the water isotope and acidity signals (see
Supplementary section S4), taken to indicate January $1^{st}$. Other CFA records also displayed
annual variations, particularly the acidity measurement, but tended to be less consistent. From
0 to 129 m, an annual signal was observed in the insoluble dust record (determined by laser
scattering) but the data below 129 m were corrupted by the presence of drill fluid in the CFA
system. Calcium and conductivity measurements sometimes displayed annual variations, but
were limited in their contribution to annual layer interpretations because multiple peaks per
year were frequently observed in these signals. The discretely-sampled ICP-MS data records
did not have sufficient resolution to resolve annual layers. *StratiCounter* was run based on the
full suite of CFA records: black carbon, acidity, dust, calcium, and conductivity, except that
the dust record was excluded in the contaminated lower part.
Due to the relatively shallow ice depth, layer thicknesses rapidly decrease with depth in the
Roosevelt Island ice core, thus it was necessary to make slight changes to the *StratiCounter*
algorithm settings through the depth range considered. We hence divided the ice-core records
into four sections, with section delimitations selected to span a given range of layer
thicknesses, as based on estimates obtained from methane matching to WAIS Divide (section
3.3.2). The algorithm was initialized based on an initial set of manual layer counts produced
for the interval 40-150 m. The *StratiCounter* procedure is described in more detail in
supplementary section S2. For the uppermost section, performance of the algorithm was
tested using a variety of smaller changes in algorithm settings, which all resulted in very
similar timescales (±10 years at 165 m). The final settings were chosen as those for which the
timescale was in best agreement with the WD2014 age of a tephra layer located at 165 m
depth (see Section 3.3.1). This tephra has a geochemistry consistent with an eruption of the
Pleiades, West Antarctica (Kurbatov et al., 2015), and tephra of similar geochemistry has
been found in the WAIS Divide core with a corresponding age of 1252±2 CE. This is
presently the only certain marker horizon in the RICE core below 42.5 m with a precise age
estimate, and the only horizon used to constrain the deeper part of the layer-counted section of
the RICE17 timescale. Proceeding to the deeper sections, the algorithm settings were kept as
similar as possible to those used in this upper part (Table S2).



## 3.3. Validation of the RICE17 chronology

The layer-counted RICE17 chronology was validated by 1) volcanic matching to WAIS Divide on the WD2014 timescale, and 2) by matching multi-decadal variations in the RICE methane record to a similar record from WAIS Divide.

Volcanic synchronization offers more precise validation between the age scales of the two cores but suffers the risk of possible event misattribution. Distinct volcanic marker horizons observed in the ice matrix are attributed in both cores, thereby offering very precise relative and absolute age constraints. However, there is a risk that some of the volcanic match-points are incorrectly identified. This risk is always present with volcanic matching, but here, the risk is greater than usual since volcanic matching of RICE to other Antarctic ice cores was not trivial. The extreme maritime environment of Roosevelt Island overwhelms the volcanic signatures due to marine biogenic sulfate emissions, and traditional approaches failed for reliable volcanic detection in the ice-core record. Adding to the challenge, many active volcanoes exist in the vicinity of Roosevelt Island. Regional eruptions produce some of the most discernable volcanic acidity peaks in the RICE ice-core records, and these complicate the task of identifying large tropical eruptions useful for synchronizing to other Antarctic ice cores. For more reliable identification of volcanic peaks in the RICE core, we use two new volcanic proxies as described in section 3.3.1.1.

Methane gas synchronization points are less likely to be misattributed but do not allow for a similarly precise matching due to the multi-decadal nature of methane variations as recorded in the ice cores, and given the need to account for the gas-age-ice-age difference ($\Delta$age). Methane matching is thus less valuable for resolving small-scale and relative age uncertainties of the layer-counted timescale. Absolute age control on the gas synchronization, however, is better than that of volcanic matching. We can therefore use methane gas synchronization to validate the absolute ages of the timescale and the volcanic match points.

Combining the two lines of evidence thus allows us to validate the RICE17 timescale with high precision (from volcanic matching based on new volcanic tracers) as well as high absolute age accuracy (from methane matching) (Fig. 6). Using the annual-layer counted timescale as base for comparison, we successfully established a robust series of volcanic match points between RICE and the WAIS Divide core (Table 2), which were subsequently confirmed by methane matching of the two cores.

### 3.3.1. Timescale validation by volcanic matching

#### 3.3.1.1 Detection of volcanic eruptions using new proxies

The low altitude and coastal location of the RICE ice-core site results in a large seasonal influx of sulfate with biogenic emissions from the surrounding ocean, tending to obscure most of the sulfur contributed by volcanic eruptions. The seasonal signal varies from almost no sulfate during winter to summer peak values of up to 200 ppb non-sea-salt sulfate. For comparison, in the WAIS Divide core, with accumulation rates of similar magnitude, sulfur deposition from the large Tambora eruption (Indonesia, 1815 CE) gives rise to maximum non-sea-salt sulfur concentrations of 80 ppb (Sigl et al., 2015). Consequently, in order to observe volcanic signatures in the RICE core, very high-resolution records are required to disentangle the sporadic influx of volcanic sulfur from the annual variability. Below 40 m, the resolution of the discretely-sampled S record is too low (9.5 cm) for this purpose.

The RICE ice core acidity has been employed as a tracer of volcanic eruptions, rather than the traditional approach of evaluating total sulfur content. In the atmosphere, volcanic emissions of gaseous $SO_2$ are rapidly oxidized to $H_2SO_4$, a strong acid. Only a fraction of biogenic



sulfur emissions, however, are oxidized to $H_2SO_4$. Consequently, the relative magnitude of
volcanic versus biogenic influx is expected to be more significant in terms of acidity than
sulfur concentrations. This is especially the case for regional volcanic eruptions. The
prevalence of quiescent regional volcanism with relatively high halogen content in West
Antarctica (Zreda-Gostynka et al., 1997) implies that these halogens may contribute strongly
to the measured ice acidity. This is not the case for larger and more distant volcanoes, since
halogens are relatively quickly removed from the atmosphere during transport. Regional
eruptions will therefore generally have the clearest expression in the acidity records, which
causes additional challenges in volcanic synchronization of RICE to other ice cores.
Acids increase the conductivity of the ice, and two commonly used tracers for volcanic
activity are measurements of liquid conductivity and ECM. ECM measures the solid ice
conductivity, which is determined by the $H^+$ content in the ice, hence providing a direct
measure for acidity. Measurements of ECM and liquid conductivity on the RICE ice-core
were both strongly influenced by the high sea salt concentrations found in the ice. As
previously noted by Kjær et al. (2016), the RICE conductivity record is almost identical to the
mostly sea-salt-derived RICE calcium record (Figs. 4 and 7), suggesting sea spray to be the
main source for peaks in liquid conductivity. The high influx of sea salts at Roosevelt Island
hampers a volcanic attribution directly from the RICE conductivity record. By comparing to
the sea-salt-derived calcium concentrations, however, we were able to extract a proxy for
non-sea-salt conductivity as the conductivity excess. Being a secondary product, this tracer is
prone to measurement errors, calibration and co-registration uncertainties, as well as affected
by the different smoothing of the records during measurements. Thus, only very distinct
signals in non-sea-salt conductivity were used for identification of volcanic eruptions, and
only when confirmed by other lines of evidence.
A second new tracer for volcanic eruptions was based on the high-resolution CFA acidity
record, which directly registers the concentration of $H^+$ ions in meltwater from the core. Also
this record is influenced by the annual variability in acidic influx from oceanic biogenic
emissions, a high peak in which potentially may be mistaken for the imprint from a small
volcanic eruption. Large eruptions with high acidic influx were, however, distinctly visible.
Figure 7a displays a sequence of volcanic signals in the RICE acidity and non-sea-salt
conductivity (conductivity-to-calcium residual) records. Substantial evidence for high acidity
influx was required in both volcanic tracers for the positive attribution of a volcanic horizon.

### 3.3.1.2 The Pleiades tephra horizon

A visible tephra layer deposited at 165.01-165.02 m has been geochemically attributed to the
Pleiades volcanic group (Kurbatov et al., 2006; Narcisi et al., 2001) in the Victoria Land
region of Antarctica (Fig. 1). This tephra has previously been found in the Talos Dome
(TALDICE) and Siple Dome ice cores (Dunbar et al., 2003). It was also matched to a
contemporary (1252±2 CE) tephra horizon in the WAIS divide ice core (pers. comm. Nelia
Dunbar, 2014) on the WD2014 timescale, which allowed a firm synchronization of the two
cores at this depth. During development of RICE17, this horizon was used to select the
optimal version of the layer-counted timescale (see section 3.2.4), resulting in a RICE17 age
of 1252±13 CE for this horizon.

### 3.3.1.3 Volcanic synchronization to WAIS Divide

With the Pleiades tephra horizon as chronostratigraphic marker, and using the RICE17 layer-
counted timescale as guideline, the sequence of volcanic horizons identified in RICE could be
linked to a volcanic record from the WAIS Divide ice core on the WD2014 chronology (Sigl
et al., 2015) (Fig. 7b). The frequency of volcanic synchronization markers was quite variable,
with unambiguous attribution of synchronization points only possible where a sequence of



relatively large volcanic eruptions could be identified in both records. Consequently, some
time periods suffer from sparse coverage of reliable volcanic links (e.g., 125-150 m, 215-250
m, 270-280 m, 335-340 m).
RICE17 volcanic eruption ages are given in Table 2. This table also includes the
corresponding eruption ages based on the WD2014 chronology, which has a maximum
uncertainty of ±7 years over the last 2700 years (Sigl et al., 2016). To facilitate depth
designation, we used the peak of the volcanic signatures in both cores as depth-age validation
points for the annual-layer-counted RICE17 timescale.
We note that several large and well-known volcanoes, routinely found in Antarctic ice cores,
are difficult to locate in the RICE acidity and conductivity records. This has a variety of
reasons. Some large volcanoes were located in broken core sections removed before CFA
measurements (e.g. around Tambora, 1815 CE) or within sections of bad data quality due to
the introduction of air or drill liquid in the CFA system. For sections with no CFA data, a
tentative attribution of these large volcanoes was made based on the limited evidence from the
non-sea-salt sulfur and ECM records, where possible (Table 2). At other times, volcanic
matching was impossible due to ambiguity caused by several closely-spaced acidic peaks in
RICE, and/or due to insufficient data to discriminate volcanically-produced acids from the
high acidic background level.
All validation points were found to be in agreement with the RICE17 chronology within the
assigned confidence interval. In the absence of additional constraints, which may become
available in future, we refrain from constraining the RICE17 chronology to exactly fit the
marker horizons, and retain its nature as an independently layer-counted timescale.

### 3.3.2. Timescale validation using decadal variability in methane records

Given the challenges in volcanic synchronization, a second validation of the RICE17
chronology was performed by matching variations in RICE gas records to similar records
from the WAIS Divide ice core (Lee et al., 2017). Records of atmospheric methane ($CH_4$) and
isotopic composition of molecular oxygen ($\delta^{18}O_{atm}$) reflect global changes in atmospheric
composition (Ferretti et al., 2005; Mitchell et al., 2013; Mitchell et al., 2011; Severinghaus et
al., 2009). Over recent millennia, the period in focus here, atmospheric $\delta^{18}O_{atm}$ concentrations
have remained stable, whereas the methane record displays multi-decadal fluctuations.
Corresponding fluctuations are found in gas records from WAIS Divide (Wais Divide Project
Members, 2013, 2015). Matching up the methane records from the two cores provided an
evaluation of the absolute accuracy of the annual-layer-counted RICE17 chronology.
The gas records from RICE and WAIS Divide were synchronized using a Monte Carlo
technique adapted from (Huybers, 2002; Lee et al., 2017). Subsequent manual adjustment to
the match-points (average adjustment: 9 years, and all smaller than 23 years) in this top part
of the two cores provided a slightly improved fit between the records. Given the stability of
the $\delta^{18}O_{atm}$ record over the last millennia, the synchronization was solely constrained by the
observed variability in the methane records. Synchronization was based on discretely-
measured gas records due to significant data gaps in the high-resolution CFA methane record.
Average sample resolution of the RICE $CH_4$ and $\delta^{18}O_{atm}$ records is 26 years, which
contributes to the matching uncertainty.
Matching records of past atmospheric composition provides the ice-core gas-age, i.e. the age
of the gas at a given depth. By modelling the densification process, the gas-age-to-ice-age
difference (Δage) can be calculated, and this correction factor can be applied to the ice-core
gas-ages to obtain the relevant ice-core ice-ages at a given depth. We estimated Δage by
applying a dynamic Herron-Langway firn densification model (Herron and Langway, 1980)



following Buizert et al. (2015). The densification model is described in detail in Lee et al. (2017), and will only briefly be described here: It is forced using a site temperature history derived from the RICE stable water isotopes, and the firn column thickness is constrained using the isotopic composition of molecular nitrogen ($\delta^{15}$N of $N_2$), while assuming a convective zone thickness of 2 m. Ice flow thinning are based upon the vertical velocity of layers measured using phase-sensitive radio echo sounding (pRES) (Kingslake et al., 2014), also used for deriving accumulation rates from the observed annual layer thicknesses in the RICE core (section 3.4).

The relatively high accumulation rates at RICE, and the relatively warm surface temperatures, give rise to small values of Δage compared to most other Antarctic sites. More importantly, the uncertainties associated with these Δage corrections are small: Estimated 1σ-uncertainties on the Δage corrections over the last 2700 years are in the order of 30 years. Hence the gas-based age markers, tightly constrained by the observed multi-decadal variability in atmospheric $CH_4$, provide a chronology that is independent of the annual-layer counting.

### 3.4. Accumulation reconstruction

The accumulation rate history is inferred from depth profiles of the annual-layer thickness, with corrections for densification of the firn and thinning of layers due to ice flow. Below we develop an accumulation history at Roosevelt Island for the past 2700 years.

#### 3.4.1. Changes in density with depth

Bag-mean densities were measured on the main RICE core for the interval 8-130 m, at which depth ice densities were reached (Bertler et al., 2017; Lee et al., 2017). In steady state, densification of the upper 8 m of the firn is parameterized using the initial snow density and the surface temperature, and further densification to ice depends on the stress from the overburden (Herron and Langway, 1980). With an initial density of 0.41 g cm$^{-3}$, a surface temperature of -22° C, and an accumulation rate of 0.22 m w.e yr$^{-1}$ the modelled density profile provides a good fit to the observed values (Fig. S1). The modelled densities for the upper 8 m provide a smooth transition to the measured density profile, and this profile was used to establish annual layer thicknesses. Note that the surface temperature derived from ERAi (-27.4±2.4° C) is significantly lower than that used in the model.

The observed density profile is well approximated using the Herron-Langway model, although it is slightly skewed towards less dense values due to strain in the firn, suggesting that effects of wind-remobilization or surface melt have been minimal.

#### 3.4.2. Thinning of annual layers due to ice flow

An estimate of vertical strain from ice flow is needed to extract information about the accumulation-rate history from the annual layer thickness profile. Kingslake et al. (2014) conducted repeat phase-sensitive radio-echo soundings (pRES) across Roosevelt Island to measure the spatial pattern of modern vertical velocities. Near-surface vertical strain rates are more compressive near the divide than on the flanks, which results in the distinctive stack of Raymond arches visible in the radar-detected stratigraphy beneath the divide in Fig. 8b (Raymond, 1983). Of particular interest is that the position of the modern divide is offset by ~500 m from the peaks of the deeper arches, suggesting recent divide migration. Here we use an approximation of the vertical velocity profile and constrain model parameters to (i) account for changes in vertical velocity caused by divide migration; and (ii) match the observed surface accumulation rate and estimate of ice sheet thinning.

The vertical velocity profile used here is found by fitting a model to measurements of englacial vertical velocities [Fig. 8a, Kingslake et al. (2014)]. Following Kingslake et al.





(2014), we parameterize the vertical velocity profile at normalized height above the bed $w(\zeta)$
using a shape factor $p$ and the vertical velocity at the surface $w_s$ (Lliboutry, 1979):

$$w(\zeta) = w_s \left( 1 - \frac{p+2}{p+1}\zeta + \frac{1}{p+1}\zeta^{p+2} \right)$$

Using $p$=-1.22 minimized the overall misfit to the measurements. Since no measurements
exist in the upper portion of the ice sheet (Fig. 8a), we choose to extend the velocity profile
linearly to the surface, starting at 155 m depth, in order to match $w_s$ of 0.26 m i.e yr$^{-1}$. This
value of $w_s$ was selected because it is the sum of the modern accumulation rate (0.24 m i.e yr$^{-1}$
over the past 50 years) and ice-sheet thinning for the past 2700 years (0.02 m i.e yr$^{-1}$). The
rate of ice-sheet thinning was estimated using an ice-flow model to match the dated
architecture of the Raymond stack (Fig. 8b).
The ice core was drilled on the modern divide but the Raymond arches at mid-depth are offset
by 500 m (Fig. 8b). The divide has likely migrated within the past several hundreds of years.
Observations from Roosevelt Island indicate the transition from divide- to flank-type flow at
Roosevelt Island occurs over distances of ~900 m (Kingslake et al., 2014). That is, ice
recovered in the ice core has experienced different vertical velocity regimes as the divide
position changed. Here we infer the evolution of the vertical velocity regime at the modern
divide using the architecture of the Raymond stack. For the core site, we construct a vertical
velocity profile appropriate for transitional-type flow between 344 m (700 BCE) until 1450
CE. We assume divide-flow after the divide reached its modern position 1450 CE. Figure 9
shows our inferred thinning function at the core site.
At the surface, uncertainty in the thinning function is zero (thinning has not started), but
uncertainties increase with depth. Uncertainty comes from: (i) the lack of pRES
measurements in the upper 90 m of the ice sheet to help constrain the near-surface vertical
velocity; and (ii) the vertical velocity profile may have varied over time in ways not
accounted for in this analysis. The second source of uncertainty is partly mitigated because
the Raymond Bump amplitude constrains the onset of divide flow to some time prior to 3000
yr ago (Martín et al., 2006). Hence most of the uncertainty in the thinning function for the
period of interest arises from the lack of measurements in the upper 90 m. To assess the
magnitude of the uncertainty we compare results using two alternate thinning functions. First,
we used the method described above to find the best fit when using a vertical surface velocity
of 0.24 m i.e yr$^{-1}$ (0.22 m w.e yr$^{-1}$), thereby neglecting the contribution from surface lowering.
Second, we used the best fit derived by Kingslake et al. (2014). The difference between the
thinning functions is a maximum at ~100 m depth. The mean difference between the thinning
functions was ~5%, with a maximum difference of 9%. Because other factors not accounted
for in this analysis may contribute additional uncertainty, we assume that the uncertainty
increases from zero at the surface, to 10% at 100 m depth (1630 CE), and then constant
uncertainty (10%) down to 344 m depth.

## 39 4. Results

### 40 4.1. The layer-counted RICE17 chronology

The RICE17 chronology covers the period back to 700 BCE (0-343.72 m), the past 2700
years. The majority of the timescale relies on CFA multi-parameter chemistry measurements,



with annual variations in black carbon as the most reliable annual tracer. Below 42.5 m (1884
CE), the timescale was produced using the *StratiCounter* annual-layer-counting algorithm,
fine-tuned to be in accordance with the WD2014 age of the Pleiades tephra layer at 165 m
depth - but otherwise it is an independent layer-counted timescale. This is also reflected in the
inferred confidence interval of the timescale, as the age uncertainty (±13 years) at the Pleiades
tephra horizon is passed on to the deeper part of the timescale. Associated 95% confidence
intervals on the age-scale shows an approximately linear increase in age uncertainty with age,
corresponding to a faster-than-linear increase with depth (Fig. 6b). The uncertainty reaches a
maximum value of ±45 years at 344 m, the end of the layer-counted section. The timescale
was validated by volcanic and gas synchronization to the WAIS Divide ice core on the
annually-layer counted WD2014 chronology (Sigl et al., 2015; Sigl et al., 2016), which has a
counting uncertainty of merely 7 years over the last 2700 years.
We observe that the layer-counted RICE17 timescale is consistent with the independent age
control points obtained by matching methane and atmospheric oxygen isotope ($\delta^{18}O_{atm}$)
records to WAIS Divide (Fig. 6b). Over the past 2700 years, there are 18 gas-age control
points. This results in an average spacing of 150 years, gradually increasing with depth.
Uncertainties in feature matching and calculation of Δage result in combined 2σ-uncertainties
for these control points of ~60 years. This uncertainty is primarily caused by synchronization
uncertainties (~48 years), and less so caused by uncertainties in Δage correction (~36 years).
Based on the automatic matching routine, agreement of RICE17 to the gas-matched ages is
better than 33 years for all control points, with a root-mean-square (RMS) difference between
the RICE17 and WD2014 timescales of 14 years. The average age difference is -1 years; as
the gas-age derived chronology and the annual layer counts are independent, the good
agreement indicates that there is no systematic bias in either the layer-counted chronology or
the methane age control points. Subsequent visual comparison of the two methane profiles
allowed slight manual adjustments to be made to the gas-age control points, which further
improved the matching. Using the modified age control points, agreement between WD2014
and RICE17 is better than 18 years with a RMS difference of 8.6 years.
Good agreement of the obtained volcanic marker horizons with both 1) absolute age markers
from methane matching, and 2) relative ages between volcanic markers as based on the
RICE17 chronology, strengthens our trust in the volcanic synchronization. Also from the
volcanic match points, we observe very good agreement between the RICE17 and WD2014
timescales. The agreement is especially remarkable down to 280 m (~200 CE), where the
observed discrepancies are less than ±4 years at all marker horizons, with a RMS difference
of 14 years. This is much less than the inferred RICE17 age uncertainty at this depth (±32
years), indicating that the inferred confidence bounds on the age-scale are reliable, albeit
somewhat conservative.
The decrease in layer thicknesses with depth causes the annual signal in the impurity records
to become increasingly difficult to reliably identify. At 280 m (200 CE), the depth resolution
of the CFA chemistry series (1-2 cm) is starting to become marginal compared to the annual
layer thicknesses (8 cm at 280 m depth). Nevertheless, the RICE17 chronology continues to
be in good accordance with the WD2014 timescale, although the volcanic matching indicates
that RICE17 has a slight bias (~2%) towards younger ages, meaning that some annual layers
are missed in the RICE records below 280 m. Consequently, the RICE17 chronology slowly
diverges from WD2014, reaching a maximum age difference of 30 years at the lowermost
volcanic age marker at 343.3 m (691.4 BCE, Table 2). This offset is within the derived
uncertainty on the timescale (45 years), and of similar magnitude as the uncertainty of the
methane-derived ages. As this age difference approaches the uncertainty of the methane-



derived RICE timescale developed in Lee et al. (2017), we decide to stop the layer-counts at the gas-age control point at 344 m.

To extend the timescale for the RICE core further down, the high-resolution annual-layer counted timescale is combined with the gas-matched timescale, which covers the entire core with lower resolution. At the lowermost gas-age control point at 343.7 m, the two independently derived RICE age-scales agree within 3 years. Hence, they can here be stitched together directly without need for any further adjustments to form a continuous timescale for the entire RICE ice core.

## 4.2. Layer thicknesses

RICE annual layer thicknesses decrease from more than 40 cm at the ice surface to ~6 cm at 344 m. Overall, the accuracy of the RICE17 timescale is supported by the smooth variations of the annual layer thickness profile (Fig. 6a).

High inter-annual variability in RICE layer thicknesses adds to the challenge of correctly identifying layers in the ice-core record. Annual layer thicknesses are distributed according to a log-normal distribution with an average standard deviation of $\sigma = 0.28$. This implies that 1.3% of the layers will have a thickness that is either twice as large or less than half the most common layer thickness. In the deeper part of the core, this corresponds to very thin layers (<3 cm) that are unlikely to be resolved by any of the available records, likely contributing to the small bias (~3%) towards undercounting of annual layers in the oldest part of the layer-counted RICE17 timescale.

The high inter-annual variability in Roosevelt Island precipitation is likely caused by the majority of the precipitation coming from individual storms bringing moisture to the location, the number of which passing by Roosevelt Island may be highly variable from year to year. In the top part of the RICE17 timescale, we observe 2010 CE to be a very thin layer, corresponding to a year of very low accumulation (7.4 cm w.e). Hardly recognizable as an annual layer from the ice core records themselves, this year has previously been identified as an extremely dry and cold winter over large parts of Antarctica (Schlosser et al., 2016), and ERA-interim data shows almost no accumulation at Roosevelt Island this year (Bertler et al., 2017).

## 4.3. Roosevelt Island accumulation history

The RICE17 layer thicknesses were converted into past accumulation rates, by correcting for density changes and ice flow thinning of annual layers with depth. The resulting accumulation history over the last 2700 years is shown in Fig. 9, along with the uncertainty from ice flow thinning when extending back in time.

### 4.3.1. Spatial consistency in observed accumulation rates

A very high degree of replicability in the derived year-to-year profile of layer thicknesses is confirmed by comparing the accumulation rates obtained for the overlap sections of the three available cores: RICE main core, RICE-12/13B, and RID-75. All cores were corrected for density changes using the density profile from the RICE main core.

The three accumulation records are very strongly correlated (correlation coefficients ranging between 0.85 and 0.87), indicating that the RICE annual layer thicknesses are representative of local snow accumulation, and that depositional noise to a high degree can be disregarded when converting annual layer thicknesses into estimates of past accumulation. The good agreement between measured water isotope records from the cores (RICE main and RID-75, Fig. 5) further confirms the consistency and stability of snow deposition at Roosevelt Island.



### 4.3.2. Long-term accumulation trends

From 700 BCE to around 1260 CE, the obtained RICE accumulation rates are slightly increasing (Fig. 9), although the uncertainty from ice flow thinning prevents any definitive conclusion on the trend during this period. At 1260 CE, the average accumulation rate reaches 0.28 m w.e yr$^{-1}$, the maximum observed value over the last 2700 years. Since then, the accumulation rate has decreased. An inflection point exists around 1700 CE, after which the decrease in accumulation rates occurs more rapidly (0.9 cm per century, Table 3). The current accumulation rate, as averaged over the last 100 years, is 0.21 m w.e yr$^{-1}$, which is at the low extreme of values observed in the record.

Uncertainties in the accumulation record arise from three factors: 1) the annual layer count, 2) the measured density profile, and 3) the applied thinning function. The uncertainty on the thinning function increases with depth and age (Fig. 9a; dotted lines). Except for in the uppermost part of the record, uncertainties in the thinning function are by far the most important, exceeding all other sources of uncertainty: At 500 BCE, close to the end of the layer-counted section of RICE17, the possible range of accumulation rates spans from 0.17 m w.e yr$^{-1}$. (i.e. significantly lower than today) to 0.4 m w.e yr$^{-1}$ (almost twice as large as today). In our estimates for the 95% uncertainty bounds on past accumulation rates, we therefore disregard the contribution from other sources than ice flow thinning of annual layers with depth. However, we note that given the applied thinning function results in almost constant accumulation levels in the earlier part of the RICE record, which may indicate that the uncertainty of the thinning function is less than our conservative estimate of 10%.

The uncertainty in the thinning function implies a high uncertainty in inferring slow trends in RICE accumulation rates over time, especially in the deeper part of the record. However, it cannot explain the rapid decrease in accumulation observed in recent time.

### 4.3.3. Large changes in recent accumulation rates

For the most recent period, uncertainties associated with our derivation of accumulation rates are small, and we can infer accumulation trends with greater confidence. This reveals a distinct decrease in recent accumulation rates (Fig. 10). The decade from 1990-2000 CE was the decade with the lowest accumulation rates since 1700 CE, and the most recent decade from 2000-2010 CE also stands out as a decade with very low accumulation. Only two previous decades (1850-1860 CE, 1950-1960 CE) have an average accumulation rate less than those observed over the last two decades. The observed mean accumulation rate of the last 10 years is 0.195 m w.e yr$^{-1}$, which is a 30% decrease in accumulation rates compared to a 100-year period around 1250 CE, when observed accumulation rates were the highest in the record.

Positive RICE accumulation anomalies have been linked to increased occurrence of eastern Ross Sea/Amundsen Sea blocking events. These blocking events impede the prevailing westerly winds, and direct on-shore winds towards the eastern Ross Sea, which in turn affects the frequency of marine air intrusion, as well as sea-ice distribution and precipitation at RICE and the western Marie Byrd Land regions (Emanuelsson et al., 2017a; Küttel et al., 2012). As eastern Ross Sea/Amundsen blocking is so closely linked to positive RICE accumulation anomalies, a decline in Ross Sea/Amundsen Sea blocking is a likely cause for the recent decline in RICE accumulation. Indeed, there has been several reports of recent strengthening of SAM [e.g., Schneider et al. (2015)] and deepening of the ASL [negative geopotential height trend; e.g. Raphael et al. (2015)], accompanied by reduction of Pacific sector southern hemisphere high-latitude blocking (Oliveira and Ambrizzi, 2017).

Linear regression on the derived annual accumulation history shows that accumulation rates at
Roosevelt Island since 1950 CE have decreased by an average of 6.6 cm yr$^{-1}$ per century,
which is almost 7 times faster than the long-term decreasing trend taking place during the
period from 1700 CE to current day (Table 3).

## 5. Discussion

### 5.1. Volcanic detection in low-elevation coastal ice cores

The identification of large tropical volcanic events in the RICE ice core is complicated by the
seasonal signal of marine biogenic sulfate, thereby requiring non-traditional methods for the
identification and assignation of these events. Whereas most high-elevation (i.e. >2000 m
altitude) ice cores have a sulfate background on the order of 50 ng g$^{-1}$ or less, the equivalent
baseline at RICE is closer to 200 ng g$^{-1}$. Identification of volcanoes therefore relied on non-
traditional techniques such as CFA-acidity measurements, along with ECM and calcium-
conductivity residuals. Each of these detection methods is effective because they focus on
total acidity (H$^{+}$ ion concentration) rather than sulfate ion concentration. Emissions from local
volcanoes includes relatively short-lived halides such as bromine, chlorine and fluoride,
which add to the acidity signal, whereas only sulfate is deposited from distant volcanic
eruptions. As a result of focusing on the total acidity, the RICE volcanic record is biased
toward local volcanism.
An implication of the geographical bias in the RICE volcanic record, combined with the high
sulfate background level, is that "typical" large distant volcanic eruptions such as Tambora
(1815 CE), Unknown (1809 CE) and Huaynaputina (1600 CE), are all difficult or impossible
to identify in the RICE records. The absence of these marker horizons contribute to
uncertainty between chronological markers, and it complicates the synchronization of RICE
to the many Antarctic ice cores, that have been dated using these eruptions.
Volcanic acid deposition markers in the RICE core include Kuwae (131.2 m, 1453 CE) and
Samalas (164.06 m, 1257 CE), among others (Table 2). In addition to the volcanic horizons
present in other Antarctic cores, we noted several strong volcanic imprints that seemingly
have no counterpart in the WAIS Divide ice core data, and thus most likely originate from
local West Antarctic volcanoes. An extended list of volcanic eruptions in the RICE core is
provided in Table 2.
The dipole effect of the Amundsen Sea Low (ASL) may additionally influence the
effectiveness by which individual volcanic events are recorded in RICE ice core (Yuan and
Martinson, 2000). The ASL dipole acts to direct storm systems either toward the Antarctic
Peninsula/Ellsworth Land region, or toward the western Marie Byrd Land/Ross Ice Shelf
region (Genthon et al., 2005). Consequently, storm tracks and associated accumulation and
wet deposition of ions is strongly controlled by the location of the ASL. This is likely to favor
deposition and preservation of volcanic signals in one location (e.g. Antarctic Peninsula) at
the expense of the other (RICE, Siple Dome). Absence of sulfate in the RICE core from some
of the larger distant volcanic eruptions may be due to a particularly strong ASL state at the
time, directing sulfate ions preferentially toward Ellsworth Land and away from Roosevelt
Island. This effect is also suggested by synchronization of volcanic peaks between WAIS
Divide and RICE ice cores, with regular periods in which synchronization of volcanic
indicators is relatively straightforward, and other periods in which synchronization between
ice cores is difficult or impossible.



Another unusual feature of the RICE volcanic record is that the core contains relatively few
visible tephra layers; Only 7 tephra horizons have been identified, of which 6 are within the
Holocene section of the core, and only one exists within the last 2700 years (Table 2). This
tephra layer (165.02 m depth) corresponds to the 1252 AD eruption of Pleiades in Northern
Victoria Land, and was also found in Siple Dome and Taylor Dome ice cores (Dunbar et al.,
2003). For comparison, Narcisi et al. (2010) found an average frequency of 1 visible tephra
layer per 1000 years in the TALDICE ice core.
These challenges of identification and age assignment of volcanoes in the RICE ice core are
analogous to the situation at other coastal locations in Antarctica with high background levels
of marine sulfate, which can effectively mask the presence of sulfate from distant volcanic
eruptions (Philippe et al., 2016; Steig et al., 2005). The RICE volcanic record demonstrates
the importance of building an Antarctic-wide network of volcanic reference horizons and the
development of non-traditional volcanic detection methods based on acidity. Particular
emphasis should be placed on the production of annually-counted timescales, especially as
CFA systems become widespread among glaciology laboratories and methods of sufficient
resolution are becoming available for relatively high-accumulation Antarctic sites, such as
RICE. The importance of a publicly available framework of ice core tephro-stratigraphy and
reference datasets for volcanic source provinces is also reaffirmed in this work.

### 5.2.  Diverging regional accumulation trends within West Antarctica

Large regional differences in accumulation trends exist within West Antarctica. In
comparison to WAIS Divide (Fig. 9), the RICE accumulation history is much more variable
on the centennial scale. Further, the longer-term trends are significantly different between the
two locations. At WAIS Divide, accumulation rates were approximately 25% higher 2500
years ago and have been declining since, with an acceleration in the rate of decline in the past
1000 years. The start of this decline takes place within roughly the same period that also
RICE accumulation rates start to decrease. Over the most recent decade (2000-2010),
however, WAIS Divide accumulation displays a marked increase in accumulation rates
(Thomas et al., 2017), whereas the accumulation at RICE has decreased.
The different trends in accumulation rates across West Antarctica may be explained by
changes in the strength of the Amundsen Sea Low. The ASL influences accumulation rates in
a dipole pattern: negative geopotential height anomalies in the ASL region reduces Amundsen
sea blocking, leading to less accumulation over the Ross Ice Shelf area and conversely,
greater accumulation over Ellsworth Land (Raphael et al., 2015). The fulcrum of this bipole is
located in the vicinity of the West Antarctic ice divide, hence the WAIS Divide ice core
should be minimally influenced by the strength of the ASL.
Most other coastal Antarctic sites have experienced a significant increase in accumulation
rates (10%) since the 1960s, due to a higher frequency in blocking systems that have
increased the precipitation in these areas (Frezzotti et al., 2013). The broad similarities and
differences noted here raise the question of whether West Antarctic accumulation, as a whole,
has been decreasing, or whether the trends represent a redistribution of precipitation. It
highlights the problem that only with large spatial coverage, will the current trend in total
Antarctic mass balance be elucidated.





### 5.3. Current mass balance of Roosevelt Island and implications for Ross Embayment mass balance

Clausen et al. (1979) estimated the current (1954-1975) accumulation rate at the summit of Roosevelt Island to be 0.20 m w.e $yr^{-1}$, whereas we here find the current accumulation rate (average of the last 50 years) to be 0.22±0.06 m w.e $yr^{-1}$. .While these estimates are consistent, we note that the discrepancy between values may be due to the observed high inter-annual variability in accumulation rates. That is, the averaging period of accumulation rates may influence the result. Adding to the challenge is the large spatial gradient in accumulation rates across Roosevelt Island, which means that the location of the measurements is important for estimating the accumulation rate. However, the correlation between annual accumulation rates obtained from the three Roosevelt Island ice cores (RID-75, RICE-12/13-B, RICE main) is high, consistent with the less than 1 km difference between their reported sampling locations.

An early mass balance study of the Ross Ice Shelf (Shabtaie and Bentley, 1987) concluded that the ice shelf is in a positive state of mass balance, gaining 38±12 $km^3$ $yr^{-1}$, while the upstream inland ice is in a state of negative mass balance: -23±15 $km^3$ $yr^{-1}$. The Ross Ice Shelf primarily gains mass from the inland ice (111±4 $km^3$ $yr^{-1}$) and snow accumulation (42±3 $km^3$ $yr^{-1}$), with snow accumulation therefore accounting for 29% of the total mass gain. With respect to our finding of a 30% decrease in accumulation rates since 1700 CE, we note that sub-shelf processes will be more significant to the overall mass balance of the Ross Ice Shelf in future. The stability of the Ross Ice Shelf may be further threatened by anticipated ocean warming and sea level rise, respectively enhancing subsurface melting and buoyant force on the ice shelf (Joughin and Alley, 2011).

### Conclusions

The RICE ice core from Roosevelt Island, Ross Ice Shelf, West Antarctica, was successfully dated back to 700 BCE by annual layer counting based on multiple high-resolution impurity records. The timescale therefore begins after the establishment of an ice divide at Roosevelt Island, and covers most of the time that stable ice divide flow has taken place here. The resulting timescale was validated by volcanic and methane synchronization to WAIS Divide ice core WD2014 chronology, and the two timescales are in excellent agreement. Reliable volcanic match points were difficult to establish, and required the use of new techniques and data sets. This indicates a general challenge for low-altitude ice core sites located close to the open ocean.

Correcting for ice flow thinning of annual layers with depth produced an annual accumulation record for Roosevelt Island for the past 2700 years. Accumulation rates were reasonably constant until 1260 CE, after which accumulation rates have consistently decreased. Current accumulation trends at Roosevelt Island indicate a rapid decline of 6.6 cm w.e per century, with a modern accumulation rate of 0.22 m w.e $yr^{-1}$ (50 year average). This recent trend is similar to that observed at WAIS Divide, although the time of change is earlier at RICE than at WAIS Divide.

Layer thickness profiles in the three Roosevelt Island ice cores analyzed here show a high degree of correlation between cores, giving confidence that the RICE core is a reliable climate archive suitable for further understanding of climate and geophysical variability across West Antarctica.



## Data availability:

The following data will be made available on the Centre for Ice and Climate website (http://www.iceandclimate.nbi.ku.dk/data/) as well as public archives PANGAEA and NOAA paleodatabase: RID-75 isotope and beta-activity records; RICE17 timescale; and RICE accumulation rates.

## Acknowledgments

This work is a contribution to the Roosevelt Island Climate Evolution (RICE) Program, funded by national contributions from New Zealand, Australia, Denmark, Germany, Italy, China, Sweden, UK and USA. The main logistic support was provided by Antarctica New Zealand (K049) and the US Antarctic Program. We thank all the people involved in the RICE logistics, fieldwork, sampling and analytical programs. The Danish contribution to RICE was funded by the Carlsberg Foundation's North-South Climate Connections project grant. The research also received funding from the European Research Council under the European Community's Seventh Framework Programme (FP7/2007-2013) ERC grant agreement 610055 as part of the Ice2Ice project. The RICE Program was supported by funding from NSF grants (PLR-1042883, ANT-0837883, ANT-0944021, ANT-0944307 and ANT-1643394) and New Zealand Ministry of Business, Innovation, and Employment grants issued through Victoria University of Wellington (RDF-VUW-1103, 15-VUW-131), GNS Science (540GCT32, 540GCT12) and Antarctica New Zealand (K049). Figure 1 was made using Quantarctica2 (Norwegian Polar Insitute) basemaps and QGIS software.

We present here ice core data collected and analyzed by Henrik Clausen, Willi Dansgaard, Steffen Bo Hansen and Jan Nielsen under the Ross Ice Shelf Project (RISP) carried out between 1973 and 1978. We acknowledge the pioneering work conducted by these researchers and the ongoing international collaborations they established.

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

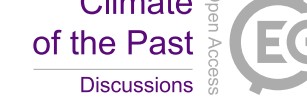

**Figures**

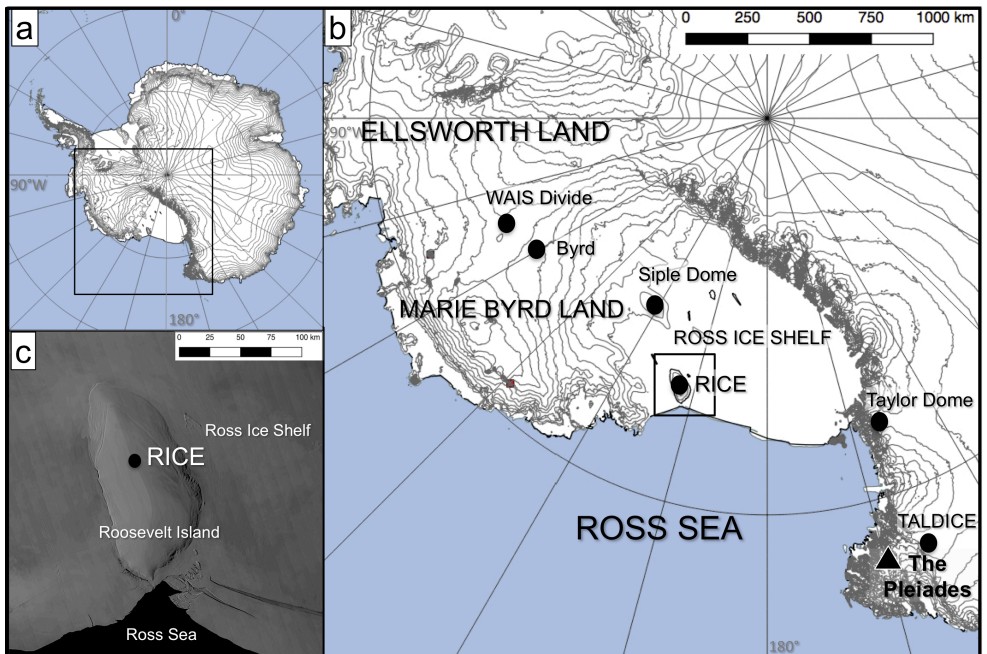


**Figure 1:** a, b): Roosevelt Island is located in the eastern margin of the Ross Ice Shelf
embayment. Locations discussed in the text are represented by triangles (volcanoes) or cirlces
(ice-cores). c) MODIS image of Roosevelt Island (Haran et al. 2013), which protrudes as an
ice dome from the surrounding Ross Ice Shelf. The RICE ice core is drilled on the ice divide
of Roosevelt Island.

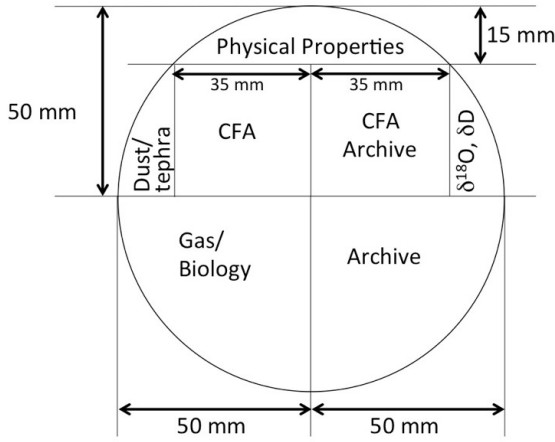

**Figure 2:** The RICE main core cutting plan included 2 CFA sticks of size 35x35mm.





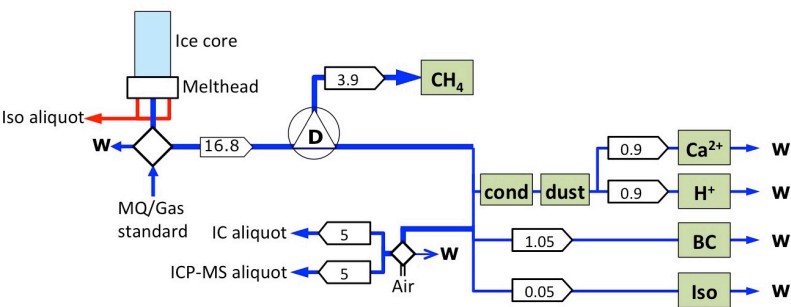

**Figure 3:** CFA set-up for analysis of the RICE core. A 1-m long ice-core rod (light blue) is
placed on a melt head, which separates melt water from the pristine inner part of the core
from that of the more contaminated outer rim. Meltwater from the outer stream (red) is used
for discrete measurements of water isotopes, while the melt water stream from the inner core
section (dark blue) passes through a debubbler (D), which separates air from the melt water.
The air composition is analyzed for methane concentration, while the meltwater stream is
channeled to various analytical instruments for continuous impurity analysis of dust,
conductivity (cond), calcium ($Ca^{2+}$), acidity ($H^+$), black carbon (BC), and water isotopes (Iso),
as well as collected in vials for discrete aliquot sampling by IC and ICP-MS. W denotes waste
water. Diamonds represent injection valves used for introduction of air or water standards
when the melter system is not in use. Arrow boxes indicate liquid flow rates in mL min$^{-1}$.
Green boxes represent analytical instruments.



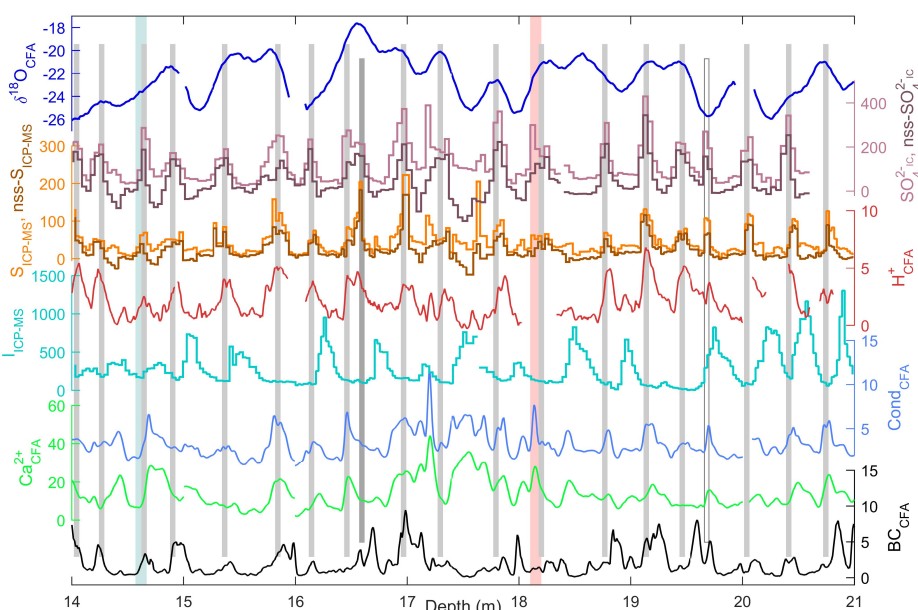

**Figure 4:** Assignment of annual layers in an upper section of the RICE core. All units are in
ppb, except for $\delta^{18}O$ (in ‰), $H^+$ (in µeq $L^{-1}$), and conductivity (in µS $cm^{-1}$). The CFA
chemistry records are smoothed with a 3-cm moving average filter. Two uncertain layers exist
within the displayed section: At 16.6 m, an uncertain layer is being counted as part of the
timescale, in order to match the tiepoint ages corresponding to the isotope match to RID-75
(cyan bar; 14.6 m) and the Raoul tephra horizon (red bar; 18.1 m). A second uncertain layer is
located at 19.7 m; the sulfate record suggest that it is an annual layer, but this is not supported
by iodine and $\delta^{18}O$. This layer is not counted in the RICE17 chronology, in order to match the
age of the next tie-point located at 22 m.





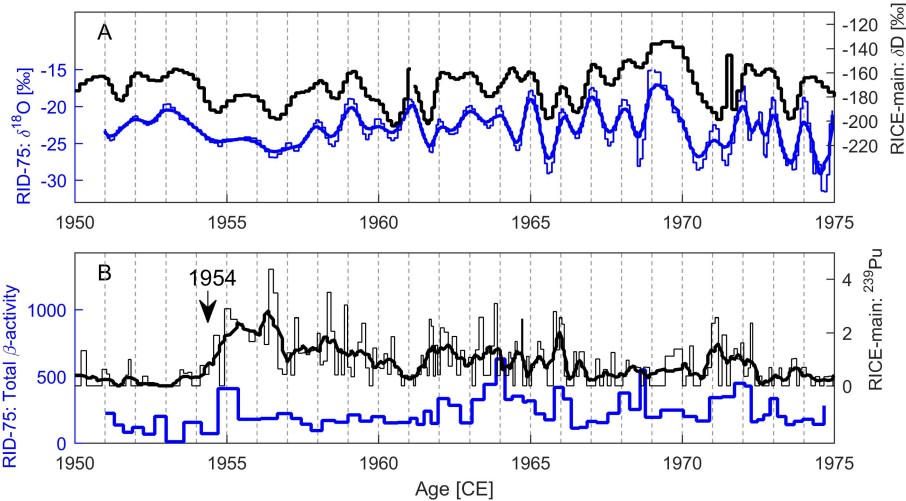

**Figure 5:** A) RICE water isotope profile (δD) compared to isotope data (δ$^{18}$O) from the old
RID-75 core for the period 1950-1975. Diffusion causes the isotope record to smooth over
time, and a smoothed version of the RID-75 isotope profile (thick blue) highlights its
similarities to the RICE isotope record. B) Total specific β-activity (in disintegrations per
hour, dph) for the RID-75 core compared to $^{239}$Pu measurements (normalized intensities) from
the RICE main core. Both cores show a sharp increase in nuclear waste deposition starting in
1954CE, and several broader peaks hereafter.





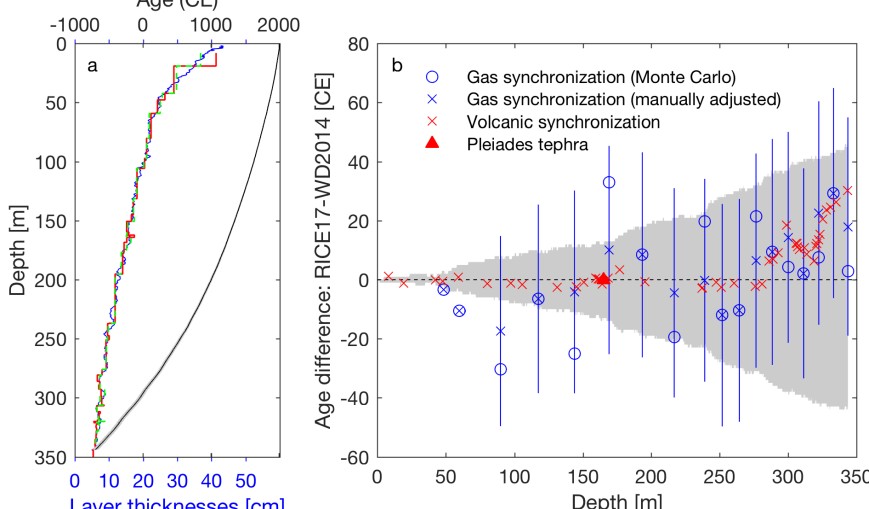

**Figure 6:** a) The RICE17 timescale with depth, including the associated 95% confidence interval (grey area, almost invisible due to scale), and the depth evolution in mean layer thicknesses (50 year running mean; blue). Red stepped line shows the annual layer thicknesses derived from volcanic match points. b) RICE17 and its corresponding 95% age confidence interval (grey area) compared to WD2014 from volcanic (red) and methane (blue) synchronization to the WAIS Divide core. The solid red triangle indicates the Pleiades tephra layer at 165m depth. A positive age difference implies fewer layers in the RICE17 timescale than in WD2014.





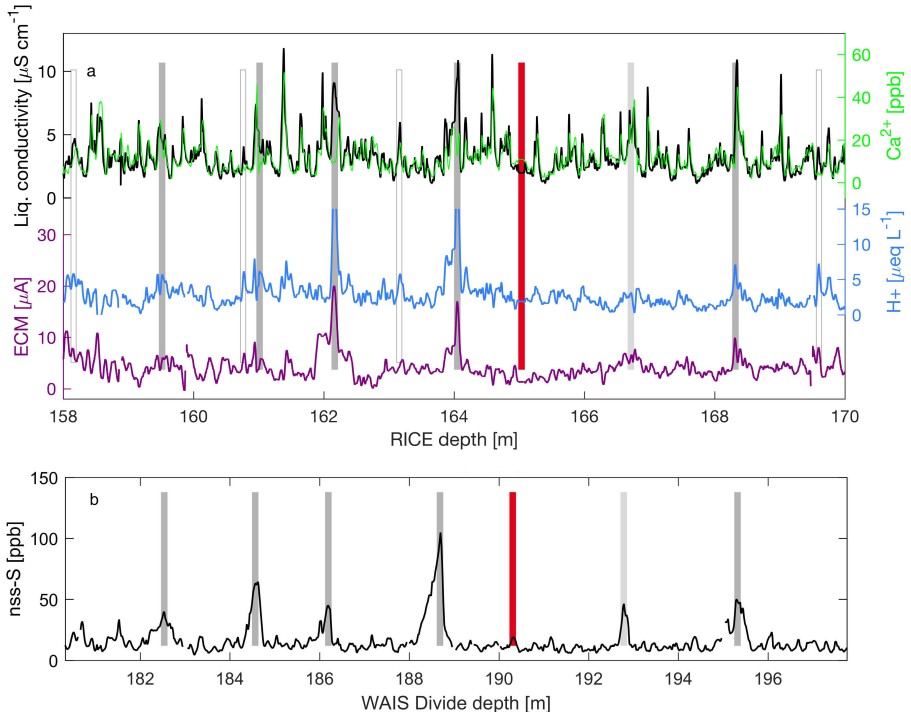

**Figure 7:** a) The RICE volcanic records: ECM (purple), acidity (blue), and calcium-to-
conductivity relation (black and green), and their synchronization to b) the WAIS Divide non-
sea-salt sulfur record (Sigl et al., 2015). Solid vertical bars indicate common volcanic match
points (Table 2). The red bar signifies the Pleiades tephra horizon (1252 CE), found in both
cores. Dark grey bars are volcanic match points that are clearly expressed in both cores. In the
RICE core, these can be found as a peak in the acidity record as well as significantly
enhanced conductivity levels compared to the sea-salt influx recorded via the calcium record.
The light grey bar is a potential volcanic match between the two cores that was not used for
validation of the timescale. RICE occasionally shows indications of volcanic activity (white
bars), for which no counterpart exist in the WAIS Divide core.





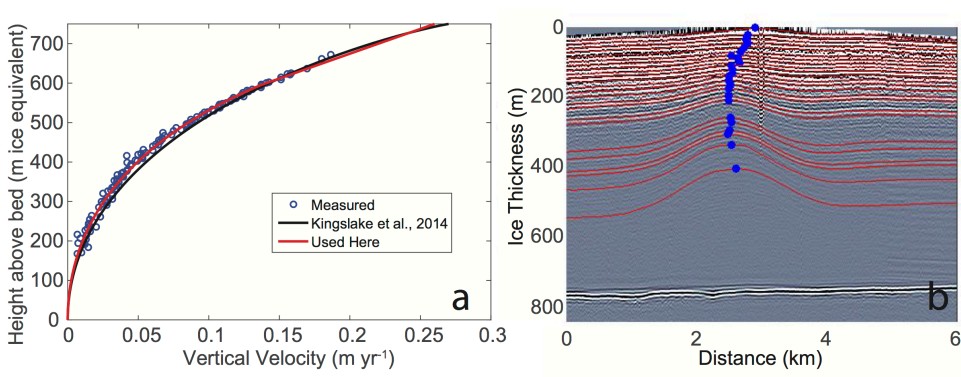

**Figure 8:** a) Vertical velocity measurements from pRES (Kingslake et al., 2014) and the
associated fitted functions. Fit used here improves overall misfit and does not have a bias at
mid-depth. b) Radar echogram with traced layers (red) and location of maximum amplitudes
of the stack of Raymond arches (blue circles). The location of the modern ice divide is
marked by the returns from a pole to the right (west) of the maximum bump amplitudes at
depth.



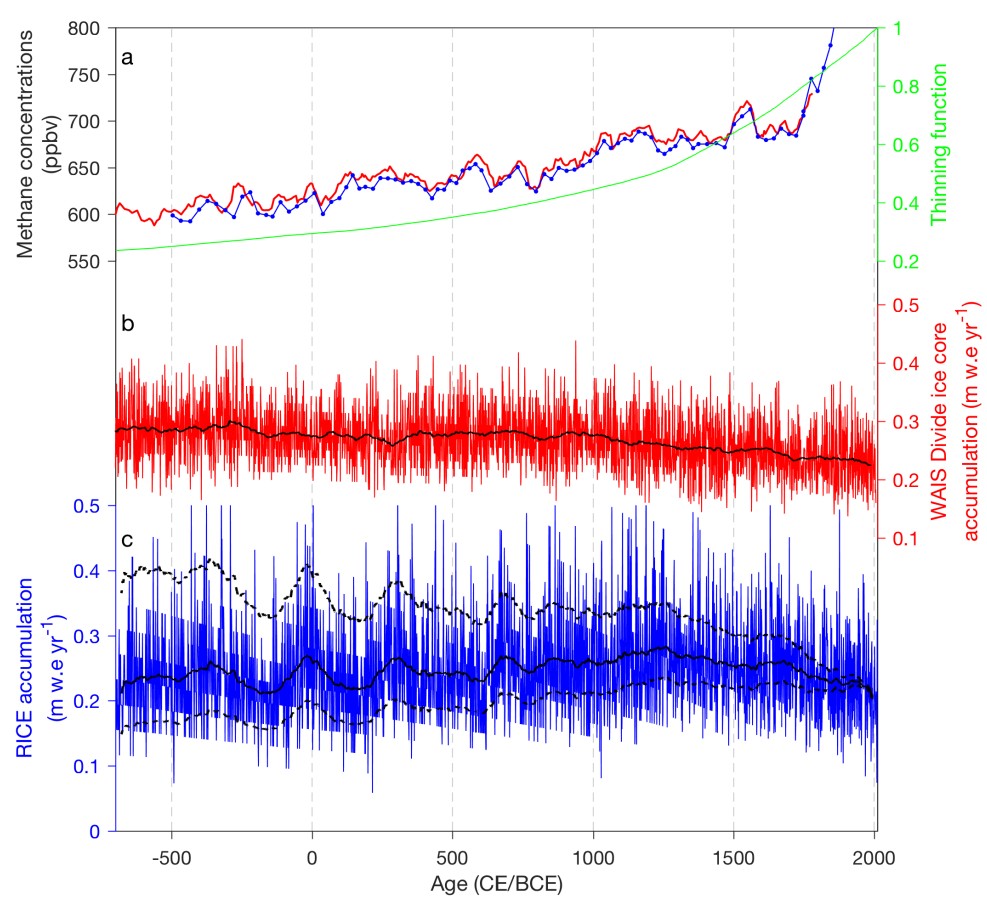

**Figure 9: a)** Measured methane concentrations from RICE (red, on the RICE17 timescale)
and from WAIS Divide (blue, on the WD2014 timescale). The derived thinning function
(green) for the RICE core is also displayed. **b)** WAIS Divide (red) and **c)** RICE (blue)
accumulation histories over the past 2700 years. A 100-year smoothed version is shown in
black. For the RICE accumulation history, the 95% confidence interval of the derived
accumulation rates is shown as dotted black lines.





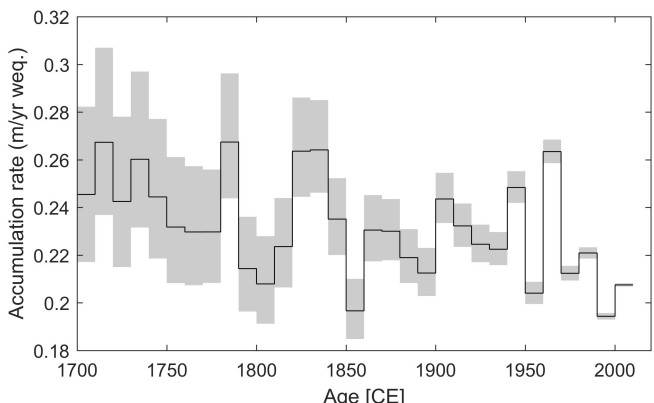

2  **Figure 10:** Decadal accumulation rates at Roosevelt Island since 1700 CE. Grey shadows
3  indicate the 95% uncertainty bounds due to uncertainties in the thinning function.





## 1 Tables

| Ice core | RID-75 | RICE | RICE-12/13-B |
|---|---|---|---|
| **Drilled** | 1974/75 | 2011/12 (0-130 m) | Jan 2013 |
| | | 2012/13 (130-764.6 m) | |
| **Depth** | 0-10.68 m | 8.57-764.60 m | 0-19.41 m |
| **Location** | 79°22' S, 161°40' W | 79°21.840' S, 161°42.360' W | 79°21.726' S, 161°42.000' W |
| **Analysed** | 1977-78 | 2011-13 | 2013 |
| **β-activity** | 16 cm resolution, Clausen et al. (1979) | - | - |
| **$\delta^{18}O$, $\delta D$** | Only $\delta^{18}O$. 4 cm resolution, Clausen et al. (1979) | Continuous and 2 cm resolution, Bertler et al. (2017) | Continuous and 2 cm resolution, Bertler et al. (2017) |
| **CFA** | - | $H^+$, $Ca^{2+}$, conductivity, dust, BC (8.57-344 m; continuous, this paper) | $H^+$, $Ca^{2+}$, conductivity, dust, BC (continuous; this paper) |
| **ECM** | - | (49-344 m; continuous, this paper) | - |
| **IC** | - | $Na^+$, $Ca^{2+}$, $Mg^{2+}$, $SO_4^{2-}$ (8.57-20.6 m; 4 cm resolution, unpublished). | $Na^+$, $Ca^{2+}$, $Mg^{2+}$, $SO_4^{2-}$ (4.5 cm resolution, unpublished) |
| **ICP-MS** | - | S, Na, I, $Pu^{239}$ (8.57-40 m; 2.5 cm resolution, unpublished). | S, Na, I (9.5 cm resolution, unpublished) |
| **$CH_4$ $\delta^{18}O_{atm}$** | - | discrete samples; Lee et al. (2017) | - |

5 **Table 1:** The Roosevelt Island ice and firn core records used in this study.





| Depth (m) | RICE17 age (CE) | Age confidence interval (95%) | Event | WD2014 age (CE) |
|---|---|---|---|---|
| **0**[*] | **2013.0** | **[2013, 2013]** | **Snow surface (January 2013)** | - |
| 8.81[*]? | 1991.6 | [1990.6, 1992.6] | Eruption: Pinatubo (Philippines, May 1991) | 1991.7 (1991.5-1993.5[2]) |
| **14.62** | **1975.1** | **[1974.1, 1976.1]** | **Isotope match to RID-75 snow surface (winter 74/75)** | - |
| 16.19 | 1970.9 | [1969.9, 1971.9] | Radioactivity peak (previously-determined age[1]: winter 1970/71) | - |
| **18.10-18.20** | **1965.2-1965** | **[1964.0, 1966.2]** | **Tephra: Raoul Island (New Zealand, Nov 1964)** | WD ages |
| 19.16? | 1963.0 | [1962.0, 1964.0] | Eruption: Agung (Indonesia, Feb 1963)? | 1965.0 (1963.6-1965.6[2]) |
| **21.98** | **1954.7** | **[1953.7, 1955.7]** | **Onset of high radioactivity levels: Castle Bravo, Marshall Islands (March 1954)** | - |
| **37. 45** | **1903.8** | **[1902.8, 1904.8]** | **Eruption: Santa Maria, Guatemala (Oct 1902)** | **1903.6 (1902.9-1904.3[2])** |
| **42.50** | **1884.5** | **[1883.5, 1886.0]** | **Eruption: Krakatau (Indonesia, Aug 1883)** | **1885.0 (1884.0-1886.4[2])** |
| 47.90 | 1863.3 | [1860.3, 1865.3] | Eruption: Makian (Indonesia, Sept. 1860/ 61) | 1863.9 (1862.6-1864.6[2]) |
| 59.38[#] | 1817.4 | [1812.4, 1821.4] | Eruption: Tambora (Indonesia, April 1815) | 1816.4 (1815.4-1818.4[2]) |
| 69.63 | 1770.2 | [1764.1, 1775.1] | Unknown | - |
| 80.08[#] | 1722.4 | [1715.3, 1728.3] | Unknown, also found in WAIS Divide, but not previously reported. | 1724.0 |
| 82.47 | 1711.4 | [1704.4, 1717.4] | Unknown | - |
| 83.78 | 1705.2 | [1698.2, 1711.2] | Unknown | - |
| 84.95 | 1699.1 | [1692.1, 1706.0] | Unknown | - |
| 85.96 | 1695.1 | [1687.1, 1701.1] | Unknown, also found in WAIS Divide | 1696.0 (1694.6-1697.4[2]) |
| 88.87 | 1680.3 | [1673.2, 1687.2] | Unknown | - |
| 97.13 | 1641.2 | [1634.2, 1649.2] | Eruption: Parker Peak (Philippines) | 1642.3 (1641.6-1643.7[2]) |
| 101.04 | 1624.0 | [1615.0, 1632.0] | Unknown | - |
| 103.80 | 1609.2 | [1600.1, 1617.1] | Unknown | - |





| 105.58[#] | 1599.3 | [1590.5, 1607.3] | Eruption: Huaynaputina (Peru) | 1600.9 (1600.4-1603.3[2]) |
|---|---|---|---|---|
| 106.56[#] | 1594.3 | [1594.2, 1585.2] | Eruption: Ruiz (Columbia)? | (1594.7-1597.5[2]) |
| 107.39[#] | 1589.9 | [1580.9, 1598.9] | Unknown, also found in WAIS Divide | (1590.8-1592.3[2])? |
| 108.01 | 1587.3 | [1578.3, 1595.3] | Eruption: Colima (Mexico)? | (1585.8-1587.0[2]) |
| 110.86 | 1571.6 | [1561.6, 1579.6] | Unknown | - |
| 111.75 | 1567.3 | 557.3, 1575.3] | Unknown | - |
| 113.47 | 1558.6 | [1548.6, 1566.6] | Unknown | - |
| 114.05 | 1555.2 | [1545.2, 1564.0] | Unknown | - |
| 114.19 | 1554.4 | [1544.4, 1563.4] | Unknown | - |
| 115.82 | 1546.3 | [1536.3, 1554.3] | Unknown | - |
| 115.99 | 1545.4 | [1535.4, 1553.4] | Unknown | - |
| 116.25 | 1543.4 | [1533.4 ,1552.4] | Unknown | - |
| 119.59 | 1524.1 | [1514.1 ,1533.1] | Unknown | - |
| 120.16 | 1520.5 | [1510.5, 1530.1] | Unknown | - |
| 120.83 | 1516.2 | [1507.2, 1526.2] | Unknown | - |
| 121.15 | 1514.3 | [1505.3, 1524.3] | Unknown | - |
| 121.85 | 1511.1 | [1502.1 ,1521.1] | Unknown | - |
| 122.67 | 1507.0 | [1497.0, 1517.0] | Unknown, also found in WAIS Divide | (1504.9-1508.1[2]) |
| 123.80 | 1500.2 | [1491.2, 1511.2] | Unknown | - |
| 124.22 | 1498.1 | [1489.1, 1508.1] | Unknown | - |
| 124.91 | 1495.0 | [1485.0, 1505.0] | Unknown | - |
| 131.24 | 1457.2 | [1447.2, 1468.2] | Eruption: "Kuwae" (Vanuatu) | 1459.8 (1458.4-1461.4[2]) |
| 132.02 | 1453.6 | [1442.6, 1464.6] | Unknown, also found in WAIS Divide | (1453.4-1454.9[2]) |
| 134.11 | 1441.1 | [1430.1, 1451.1] | Unknown | - |
| 145.11 | 1376.4 | [1365.4, 1387.4] | Unknown, also found in WAIS Divide | 1378.8 (1377.8-1379.7[2]) |
| 145.93 | 1372.3 | ]1361.3, 1383.7] | Unknown | - |
| 149.57 | 1350.3 | [1338.3, 1362.0] | Unknown | - |
| 150.53 | 1345.2 | [1333.2, 1356.2] | Unknown, also found in WAIS Divide | 1346 (1344.5-1348.1[2]) |
| 151.69 | 1337.1 | [1326.1, 1349.1] | Unknown, also found in WAIS Divide | (1336.3-1337.6[2]) |
| 152.81 | 1329.9 | [1318.9, 1341.9] | Unknown | - |
| 153.19 | 1327.5 | [1316.5, 1339.5] | Unknown | - |





| | | | | |
|---|---|---|---|---|
| 153.68 | 1324.5 | [1313.5, 1336.5] | Unknown | - |
| 154.72 | 1316.9 | [1305.9, 1329.9] | Unknown | - |
| 155.48 | 1311.6 | [1300.6, 1324.6] | Unknown | - |
| 157.13 | 1302.0 | [1290.0, 1315.0] | Unknown | - |
| 157.61 | 1299.0 | [1287.0, 1312.0] | Unknown | - |
| 158.15 | 1295.7 | [1283.7, 1308.7] | Unknown | - |
| 159.52 | 1287.2 | [1274.5, 1299.5] | Unknown, also found in WAIS Divide | (1285.6-1288.2[2]) |
| 160.77 | 1279.2 | [1266.2, 1291.2] | Unknown | - |
| 161.01 | 1277.5 | [1264.5, 1289.5] | Unknown, also found in WAIS Divide | (1276.5-1278.7[2]) |
| 162.17 | 1269.9 | [1257.0, 1282.9] | Unknown, also found in WAIS Divide | (1269.5-1271.4[2]) |
| 164.05 | 1257.4 | [1245.4, 1270.4] | Eruption: Samalas (Indonesia) | (1257.9-1261.1[2]) |
| **165.04** | **1251.4** | **[1239.4, 1264.4]** | **Tephra: Eruption: Pleiades (West Antarctica)** | **1252 ± 2** |
| 166.71 | 1242.1 | [1229.1, 1255.1] | Unknown, also found in WAIS Divide | 1242.0±2.0 |
| 168.32 | 1231.4 | [1218.4, 1245.4] | Unknown, also found in WAIS Divide | 1231.0±2.0 |
| 169.60 | 1223.1 | [1210.1, 1237.1] | Unknown | - |
| 172.94 | 1201.4 | [1188.4, 1217.4] | Unknown | - |
| 173.91 | 1194.6 | [1180.6, 1210.6] | Unknown | - |
| 174.95 | 1188.0 | [1174.0, 1204.9] | Unknown | - |
| 175.56 | 1184.2 | [1170.2, 1201.2] | Unknown | - |
| 175.99 | 1181.2 | [1166.2, 1198.2] | Unknown | - |
| 176.76 | 1175.5 | [1160.5, 1192.5] | Unknown, also found in WAIS Divide | 1173±2 |
| 177.84 | 1166.8 | [1151.8, 1183.8] | Unknown | - |
| 178.74 | 1160.2 | [1145.2, 1178.2] | Unknown | - |
| 182.69 | 1133.0 | [1116.0, 1150.0] | Unknown | - |
| 183.35 | 1129.3 | [1112.3, 1146.3] | Unknown | - |
| 184.10 | 1123.9 | [1106.9, 1140.9] | Unknown | - |
| 184.36 | 1122.7 | [1105.7, 1139.7] | Unknown | - |
| 185.01 | 1117.7 | [1100.7, 1134.7] | Unknown, also found in WAIS Divide | 1111±2 |
| 185.51 | 1114.3 | [1097.3, 1131.3] | Unknown | - |
| 185.93 | 1111.2 | [1094.2, 1128.2] | Unknown | - |





| 186.44 | 1107.7 | [1090.7, 1124.7] | Unknown | - |
|--------|--------|------------------|---------|---|
| 189.41 | 1086.5 | [1068.5, 1104.0] | Unknown | - |
| 195.23 | 1039.4 | [1021.4, 1058.4] | Unknown, also found in WAIS Divide | 1041±2 |
| 201.13 | 995.6 | [975.6, 1014.6] | Unknown | - |
| 203.47 | 974.3 | [954.6, 993.6] | Unknown | - |
| 204.47 | 966.0 | [946.0, 986.0] | Unknown | - |
| 206.03 | 954.1 | [934.1, 974.1] | Unknown, also found in WAIS Divide | 961±2 |
| 208.14 | 937.0 | [917.0, 957.0] | Unknown | - |
| 209.82 | 923.4 | [903.4, 943.4] | Unknown | - |
| 213.50 | 891.3 | [871.3, 912.3] | Unknown | - |
| 215.72 | 874.2 | [853.2, 894.4] | Unknown | - |
| 216.45 | 867.5 | [846.5, 888.0] | Unknown | - |
| 222.48 | 817.6 | [795.6, 837.6] | Unknown | - |
| 232.10 | 726.1 | [705.1, 749.1] | Unknown | - |
| 232.67 | 720.3 | [699.3, 743.3] | Unknown | - |
| 236.93 | 683.1 | [661.1, 706.1] | Unknown, also found in WAIS Divide | 686.0±2.2 |
| 237.24 | 680.3 | [658.3, 703.3] | Unknown, also found in WAIS Divide | 683.0±2.2 |
| 239.30 | 663.6 | [640.6, 685.6] | Unknown | - |
| 246.94 | 580.8 | [553.8, 607.8] | Unknown | - |
| 247.47 | 575.3 | [548.3, 602.3] | Unknown, also found in WAIS Divide | 577.0±2.4 |
| 248.76 | 561.3 | [534.3, 589.3] | Unknown | - |
| 250.94 | 539.2 | [511.2, 566.2] | Unknown, also found in WAIS Divide | 542.0±2.5 |
| 252.09 | 527.4 | [499.4, 555.4] | Unknown | - |
| 253.81 | 509.1 | [480.1, 537.1] | Unknown | - |
| 254.79 | 498.2 | [469.2, 527.2] | Unknown | - |
| 255.62 | 488.8 | [459.8, 517.8] | Unknown, also found in WAIS Divide | 495.0±2.6 |
| 256.24 | 481.4 | [452.4, 510.4] | Unknown, also found in WAIS Divide | 489.0±2.6 |
| 257.36 | 469.1 | [440.1, 498.1] | Unknown | - |
| 258.65 | 454.2 | [425.2, 483.2] | Unknown | - |
| 260.59 | 434.3 | [404.3, 463.3] | Unknown, also found in WAIS Divide | 436.0±2.8 |



| | | | | |
|---|---|---|---|---|
| 264.16 | 394.9 | [363.9, 424.9] | Unknown | - |
| 265.16 | 381.8 | [351.5, 411.8] | Unknown | - |
| 265.79 | 374.2 | [343.2, 404.2] | Unknown, also found in WAIS Divide | 380.0±2.9 |
| 267.00 | 360.8 | [329.8, 390.8] | Unknown | - |
| 267.41 | 357.0 | [325.9, 387.0] | Unknown | - |
| 267.61 | 355.3 | [323.3, 358.3] | Unknown | - |
| 268.11 | 348.8 | [317.8, 379.0] | Unknown, also found in WAIS Divide | 355.0±2.9 |
| 269.36 | 334.1 | [304.1, 366.1] | Unknown | - |
| 269.50 | 332.6 | [302.6, 364.6] | Unknown | - |
| 269.93 | 328.2 | [298.2, 360.2] | Unknown | - |
| 270.49 | 322.3 | [291.3, 354.3] | Unknown | - |
| 271.08 | 316.4 | [285.4, 347.4] | Unknown | - |
| 272.25 | 305.4 | [274.4, 337.4] | Unknown, also found in WAIS Divide | 305.0±3.1 |
| 272.75 | 299.8 | [268.8, 331.6] | Unknown | - |
| 276.05 | 264.4 | [232.4, 295.4] | Unknown, also found in WAIS Divide | 267.0±3.1 |
| 279.31 | 224.4 | [192.4, 256.4] | | |
| 280.81 | 205.5 | [172.5, 237.5] | Unknown, also found in WAIS Divide | 207.0±3.3 |
| 281.08 | 202.2 | [169.2, 234.2] | Unknown | - |
| 282.43 | 184.1 | [149.1, 217.1] | Unknown | - |
| 282.67 | 180.3 | [145.3, 213.3] | Unknown | - |
| 283.34 | 171.2 | [136.2, 204.2] | Unknown, also found in WAIS Divide | 171.0±3.4 |
| 283.65 | 166.3 | [131.3, 200.3] | Unknown | - |
| 284.79 | 150.4 | [115.4, 184.4] | Unknown | - |
| 284.98 | 148.1 | [113.1, 182.1] | Unknown, also found in WAIS Divide | 144.0±3.4 |
| 286.17 | 131.9 | [96.9, 166.9] | Unknown, also found in WAIS Divide | 126.0±3.5 |
| 286.40 | 128.3 | [93.3, 163.3] | Unknown, also found in WAIS Divide | 122.0±3.5 |
| 287.51 | 113.3 | [77.3, 148.3] | Unknown | - |
| 288.35 | 102.3 | [66.3, 137.3] | Unknown, also found in WAIS Divide | 96.0±3.5 |
| 289.18 | 90.9 | [54.9, 126.9] | Unknown, also found in WAIS Divide | 84.0±3.6 |



| | | | | |
|---|---|---|---|---|
| 291.11 | 64.3 | [27.3, 100.3] | Unknown | - |
| 292.83 | 40.9 | [3.9, 77.9] | Unknown, also found in WAIS Divide | 32.0±3.7 |
| 294.27 | 22.9 | [-14.1, 59.9] | Unknown | - |
| 294.68 | 18.7 | [-18.3, 55.7] | Unknown | - |
| 294.95 | 15.8 | [-21.2, 52.8] | Unknown | - |
| 295.46 | 10.3 | [-26.7, 47.3] | Unknown | - |
| 296.12 | 3.1 | [-33.8, 41.1] | Unknown | - |
| 296.41 | -1.6 | [-37.6, 36.4] | Unknown | - |
| 298.15 | -22.0 | [-59.0, 16.0] | Unknown | - |
| 298.39 | -24.3 | [-61.3, 13.7] | Unknown | - |
| 298.76 | -28.0 | [-65.0, 10.0] | Unknown, also found in WAIS Divide | -46.0±3.9 |
| 299.64 | -38.9 | [-75.9, -0.9] | Unknown | - |
| 300.25 | -47.6 | [-84.6, -8.6] | Unknown | - |
| 300.71 | -52.7 | [-89.7, -13.7] | Unknown, also found in WAIS Divide | -72.0±3.9 |
| 301.83 | -65.8 | [-104.8, -27.8] | Unknown | - |
| 302.75 | -77.8 | [-117.8, -40.8] | Unknown | - |
| 303.01 | -80.9 | [-120.9, -43.9] | Unknown, also found in WAIS Divide | -99.0±4.0 |
| 303.80 | -94.2 | [-133.2, -56.2] | Unknown | - |
| 304.13 | -97.9 | [-137.9, -59.9] | Unknown | - |
| 305.09 | -112.1 | [-151.1, -73.1] | Unknown | - |
| 305.98 | -125.8 | [-164.8, -86.8] | Unknown, also found in WAIS Divide | -138.0±4.1 |
| 306.41 | -131.0 | [-171.0,-92.0] | Unknown, also found in WAIS Divide | -143.0±4.1 |
| 306.81 | -136.7 | [-175.7, -97.7] | Unknown | - |
| 306.89 | -138.0 | [-177.0, -99.0] | Unknown, also found in WAIS Divide | -148.0±4.1 |
| 308.30 | -161.0 | [-201.0, -122.0] | Unknown, also found in WAIS Divide | -171.0±4.1 |
| 309.13 | -174.0 | [-214.0, -135.0] | Unknown | - |
| 310.08 | -188.7 | [-228.7, -148.7] | Unknown | - |
| 310.21 | -190.8 | [-230.8, -150.8] | Unknown | - |
| 310.89 | -202.9 | [-242.8, -162.8] | Unknown, also found in WAIS Divide | -211.0±4.2 |
| 311.92 | -216.8 | [-257.8, -177.8] | Unknown, also found in WAIS Divide | -227.0±4.3 |





| 313.40 | -239.0 | [-280.0, -199.0] | Unknown, also found in WAIS Divide | -247.0±4.3 |
| 314.02 | -247.7 | [-288.7, -207.7] | Unknown | - |
| 314.39 | -252.6 | [-293.6, -212.6] | Unknown | - |
| 317.31 | -296.1 | [-338.1, -255.1] | Unknown | - |
| 318.9 | -318.8 | [-359.8, -276.8] | Unknown, also found in WAIS Divide | -325.0±4.5 |
| 320.24 | -334.1 | [-376.1, -293.1] | Unknown, also found in WAIS Divide | -345.0±4.5 |
| 320.88 | -344.6 | [-385.6, -302.6] | Unknown, also found in WAIS Divide | -356.0±4.6 |
| 322.15 | -362.7 | [-403.7, -320.7] | Unknown, also found in WAIS Divide | -376.0±4.6 |
| 322.80 | -372.1 | [-413.1, -330.1] | Unknown | - |
| 323.14 | -376.7 | [-417.7, -333.7] | Unknown, also found in WAIS Divide | -392.0±4.6 |
| 325.25 | -405.5 | [-447.5, -363.5] | Unknown, also found in WAIS Divide | -426.0±4.7 |
| 327.09 | -430.7 | [-472.7, -388.7] | Unknown | - |
| 327.98 | -445.2 | [-487.2, -403.2] | Unknown, also found in WAIS Divide | -469.0±4.8 |
| 328.58 | -456.1 | [-498.1, -413.1] | Unknown | - |
| 329.53 | -469.1 | [-511.1, -426.1] | Unknown | - |
| 330.49 | -484.0 | [-526.0, -441.0] | Unknown | - |
| 330.64 | -486.4 | [-528.4, -442.4] | Unknown | - |
| 331.04 | -491.8 | [-534.8, -448.8] | Unknown | - |
| 331.15 | -495.1 | [-538.1, -452.1] | Unknown, also found in WAIS Divide | -519.0±4.9 |
| 332.14 | -510.9 | [-553.9, -467.9] | Unknown | - |
| 332.78 | -522.5 | [-565.5, -479.5] | Unknown | - |
| 334.93 | -554.6 | [-598.6, -511.6] | Unknown, also found in WAIS Divide | -580.0±5.2 |
| 335.85 | -567.5 | [-611.5, -524.5] | Unknown, also found in WAIS Divide | -596.0±5.3 |
| 340.90 | -651.0 | [-695.0, -607.0] | Unknown | - |
| 342.34 | -673.3 | [-719.3, -630.3] | Unknown | - |
| 342.75 | -682.1 | [-727.1, -638.1] | Unknown | - |
| 343.3 | -691.4 | [-745.0, -656.0] | Unknown, also found in WAIS Divide | -722.0±6.0 |

1  *: Depth in RICE-12/13-B shallow core. 1: Age from Clausen et al. 1979. #: CFA acidity record is missing for
2  relevant interval, depth attribution is based on ECM and cond-Ca-relation. 2: Age interval for volcanic influx at



WAIS Divide from (Sigl et al. 2013), below 1258CE updated according to the new WAIS Divide timescale,
WD2014 (M. Sigl et al. 2016). ¤Below 42.3m: decimal ages have been calculated assuming BC to peak Jan 1st.
**Table 2:** Volcanic marker horizons used for development and validation of the RICE17
chronology. Strata in bold were used for constraining the timescale. Synchronization to WAIS
Divide provides volcanic ages according to WD2014 (Sigl et al. 2015; Sigl et al. 2016). Age
of sulfur peak and age interval for enhanced sulfur influx is provided. RICE depths
correspond to volcanic peaks. Historical eruption ages (in column 4) are start date of the
eruptions. Unless otherwise mentioned, volcanic attribution is based on acidity, the calcium-
conductivity difference, ECM and/or sulfur records. Due to difficulty in RICE volcanic
attributions, the depth assignation corresponds to the peak values, which may occur
significantly later than start of the eruption. Peak values, as well as the age range for the
volcanic sulfur influx (Sigl et al. 2013) at the WAIS Divide site are provided.

| Time period | Accumulation rate trend [cm w.e per century] |
| --- | --- |
| 700 BCE – 1260 CE | +0.2 ±0.1 |
| 1260 CE – 1700 CE | -0.3±0.3 |
| 1700 CE – 2012 CE | -0.9±0.6 |
| 1950 CE – 2012 CE | -6.6±0.5 |

19   **Table 3:** RICE accumulation rate trends during various time periods.

