# Peer review of "A 2700-year annual timescale and accumulation history for an ice core from Roosevelt Island, West Antarctica"

_Climate of the Past, 2017_

## Referee Comment (RC1) · Anonymous Referee #1 · 11 Oct 2017

Review of Winstrup et al. 2017: A 2700-year annual timescale and accumulation history for an ice core from Roosevelt Island, West Antarctica

Overview: WINSTRUP et al. present a 2700 year annually resolved timescale for the Roosevelt Island Climate Evolution (RICE) ice core constructed by identifying annual layer in multiple impurity records and synchronized to WAIS Divide ice core using methane and volcanic signal. The work is motivated by the effort to provide the best accurate timescale for the upper 344 m about 2700 yr. This paper is used for the datation of the accompanied paper submitted to Climate of the past discussion by Bertler et al., CP2017-95.

[Figure]

An accurate timescale is prerequisite for any paleoclimatic reconstruction. The important effort to reconstruct the timescale of RICE ice core must be supported. While the main result is of interest this manuscript suffers from a number of flaws. The manuscript is poorly written, too long with several repetition redundant, several contradictions between the same paragraph or other paragraphs, the data and the result are inaccurate present (see example tephra layers). The methods chapter are not well structured with several information reported two three times and not in the appropriate chapter as result and discussion. Most of the information about the methods is reported in the supplementary material, where are more clearly presented. The manuscript must be completely revised and shortening significantly. Some references are uncorrected or mismatched. Five accompanied papers of RICE core are submitted or in preparation, but their result are used to validate or as source of the result of the manuscript (ex. Lee et al., in preparation). To make this manuscript a significant contribution to the literature, the authors need to better justify their time scale and snow accumulation records.

In particular:

Clarify the use of the WAIS volcanic signal and methane with RICE17 chronology, in the text look like that is use as synchronisation (see 3.3.1.3), but several point is stressed that the accuracy is low and it is use only at posteriori as validation. All the process of comparison between RICE and WAIS must be clarify, it is repeated several time in different way. If the two records are synchronised by volcanic the age error must be the same closer the tie points, between one tie point to other can increase. The process must be revised.

The tephra layers where used to fix the chronology, but it is not reported the analysis of tephra particles (Raboul 1964 CE and Pleaide 1252 CE) and the analysis on WAIS ice record (up to now never published on my knowledge), that can be permit an unequivocal attribution.

The explanation because nssSO4 signal or acidity peak of major eruption reconnaissance in WAIS (Tambora, Unknow etc.) are not recorded in the RICE records is questionable, but Authors have attributed as unknown more than hundreds chemistry signal to volcanic eruption (123 event Table 2) and those are not observed in WAIS or others ice core in Ross Sea (Siple Dome, Taylor Dome, Talos Dome). Why RICE records is able to record 193 volcanic event, with all the problems pointed out in paragraph 3.3?

Black Carbon, on the base of figure 4 does not appear the best proxies of seasonal signal, H+ appear more conservative and less misleading of BC.

Authors report strong gradient in snow accumulation spatially ranged from 0.09 to 0.30 m we/yr and migration of the dome from 500 to 900 m. Can the Authors exclude any impact on the snow accumulation history due to migration of the dome ? and/or on thinning function?

Paragraph 5.3 "Current mass balance. . .. "does not report any new valuable information for the mass balance of the RIS

In detail: Pag 3 line 44-47, How could explained stable ice divide flow with a migration of the ice divide position of around 500 -900 m?

Pag 4 Line 33-34 RICE could be representative of East Ross Sea, not of Victoria Land, see accompanied RICE paper (Bertler et al., submitted)

Chapter methods This part is too long and inappropriate as method chapter and most of the text must be moved to result chapter. The does not provide information about the sample resolution along the core and the analysis performed and at which resolution ice (cm) and sample per year. Percentage of missing record of CFA example must be reported and show. Most of the information about methods is relegate in the supplementary info.

Pag 6 Line 21-24 This paragraph present contraindiction in several points, along the entire text, correlation between RICE and WAIS "volcanic event" are used or not to tune

the RICE scale? Ex. See line 29-30 of pag 6

Pag 6 line 24 The Raoul tephra is a unequivocal volcanic event or not?

Pag 6 line 31-32 The layer counting stops at 343.72 because the annual layer is to fine (<6 cm), to identify seasonal signal needs at least 10-12 sample per year, a graph showing the number of sample analysed per year must be show, from the surface to the 344 m.

Pag 6 44-46 The record of overlap section must be shown to see the ratio noise/signal in the two cores.

Pag 7 line 10 "Several other records also displayed annual variability, but much less reliability" Why do use BC instead of H+ or both?

Pag7 line 11-16 The peak of proxies seasonality are quite different in time (isotope versus sea ice proxies, or photolysis) and most depend from the occurrence of snow fall. The use of ERA model does not look appropriate and the reference is still not published.

Pag 8 line 17-21 Geochemical composition of the tephra at RICE-WAIS and source must be show before any attribution of a tephra never reported in Antarctica before (Raoul 1964).

Pag 8 Line 21-36, it is not clear why some sulphate deposition is attributed to eruption and other no, and correlated to WAIS.

Pag 8 Analysis of comparison between manual and automated annual layer counting must be performed and show.

Pag 9 line 14-15. BC is used to date the 90% of the core analysed, but on the base of fig 4. is not the best proxies of seasonality, also as reported by Author pag 7 line 10. At line 26-27 is reported different use of the proxies the Authors contradict themself.

Pag 9 line 39-40. The geochemistry is not show; Pleiades volcano is not West Antarctica, but in Northern Victoria Land; Kurbatov et al., 2015 is not reported in reference, and it is not present in any database as reference for tephra layer reported; the 1252 tephra attributed to The Pleiade was iscovered the first time at Talos Dome and dated at 1254+- 2 by Narcisi et al., 2001; on my knowledge the analysis of the tephra at WAIS is still not published.

Pag. 10 line 5-18 RICE site is farer than other cores (WAIS-Byrd-Siple Dome, Taylor Dome and Talos Dome) from "many active volcanoes", it is very difficult to understand why RICE record is able to identify volcanic eruption those are not identified in the ice cores of the region with much lower ratio in noise/signature due to marine biogenic sulphate emissions.

Pag 10 line 19-25 Methane gas synchronization less precise than volcano matching, after 4 line "but methane it is better than volcanic matching", clarify.

Paragraph 3.3.1.1 The use of acidity and ECM to detect the volcanic signal is not a new tools. Hammer have used H+ in 1980 as proxies of volcanic signal.

Pag 10 line 43 a resolution of 9.5 cm (about 4 sample per yr) is very low to observe the seasonal variation, but enough to detect the important volcanic signal like Tambora, Kuave etc present a signal for 2-3 yr in Antarctic cores (from 8 to 12 sample).

Pag 10 line 25-32 On the base of figure 7a and Table 2 the volcanic events identified in RICE at 158.15 m and 160.77 m do not present any clear evidence in ECM, H+, due to the high background of ssSO4 at RICE. It is very difficult to understand why a site with this high noise/signal ratio could record local eruption sulphate does not observed at other site core. Paragraph 3.3.1.2 The Pleiades tephra horizon is discussed in several part of manuscript (parag 3.2.1, 3.2.4 and 3.3.1.2), but without provide any evidence on the base of geochemistry analysis. This tephra layer was reported for the first time in Talos Dome 1996 ice core and dated by Narcisi et al., 2001 at 1254+-2 and attributed as source to Melbourne Volcanic Province, probably "The Pleiades", located about 250 km from Talos Dome. This tephra was than identified in Siple Dome and Taylor Dome

by Dunbar et al., 2003. Moreover, Narcisi et al., 2012 pointed out that at TALDICE (a core drilled from 2004 to 2007) is present the tephra of 1254 as TD87a (86.20 m depth) close in composition to the previous of found in the ice core on 1996. However a subordinate set of glass shards (TD87b) is also trachytic but with a chemical signature inconsistent with The Pleiades products. Mount Berlin could thus be a suitable source of ash (Narcisi et al., 2012). Moreover an other tephra layer TD85 at 84.37 m depth younger than 25 yr has been reported by Narcisi et al., 2012 and the suggest source is Mt Melbourne volcanic province. Without any geochemistry analysis is impossible attribute unequivocally the tephra found in RICE.

Pag 11 Line 41-42 If the tephra layer identified is 1252+-2 yr, why use 1252+-13 for this horizon in RIC17? Pag 12 line 9-18 see above, more than 170 volcanic event most of them never see in closer core.

Pag 12 line 19-42 Volcanic event and Methane records are used for Synchronization or validation?

Pag 13 line 9-10 Which is the gas-age at RICE ? and compared to closer site as Siple and WAIS?

Pag 13 Line 24-29, What is the source of surface temperature of -22°C? Why is used this instead of -27.4°C, this value is also proposed in accompanied paper of Bertler et al., submitted. Why do you use a warmer temperature of 5.4°? Which implication on density and thinning model using 5.4° warmer?

Pag 13 line 30-40 Kingslake et al, 2014 instead of Raymond

Pag 13 line 40-41 The recent migration of ice divide, could be attributed to change in snow accumulation variability at ice divide? Due to the snow accumulation variability between the flank of the ice divide, which is the influence have on snow accumulation record and thinning function of RICE core? Kingslake et al., 2014 report that near-surface strain rates are compressive at ice divide than in the flanks 90% higher at

RICE. The migration of the ice divide respect to Raymond Bump position indicate a role of temporally changing in spatial snow accumulation distribution, as well as the role of along-ridge flow is un-clear and hampers a solid interpretation about thinning function and snow accumulation records.

Fig 8 The pRES measurement (Kingslake et al., 2014) was at ice divide and does not follow the Raymond bump features as reported in figure 8b.

Pag 14 line 11-20 On the base of which data the Authors construct a vertical velocity profile along the Raymond Bump?

Pag 15 line 14 "control point . . . . . . .of atmospheric oxygen isotope" at page 12 "Given the stability of the $\delta$18Oatm record over the last millennia, the synchronization was solely constrained by the observed variability in the methane records" as in several other part of the text none coherence exist between the paragraphs and some times also in the same paragraph.

Pag 15 Along all the paragraph it is not clear the process of adjustments of the counting layer respect to matching between RICE and WAIS.

Pag 16 line 13-16 High internal-annual variability in snow accumulation is normal issue (see eg Eisen et al., 2008 and reference within), 1.3% is a very low value.

Pag 16. Line 40-45 The three accumulation record of snow accumulation must be shown in the overlap time, the correlation coefficient of 0.85 and 0.87 indicate that the RICE annual are representative, but at pluriannual scale (see Eisen et al., 2008; Frezzotti et al., 2007). The comparison of the three cores can confirm only the stability of snow in the overlap time, not at secular or millennium scale.

Paragraph 4.3.2 and 4.3.2 The inflection point in accumulation of fig 9 with a trend in decrease is closer to age of the hypothesis of the stabilization of the ice divide at present position 1450 EC ( Pag 14). The uncertainty of change in accumulation must be taking in account also the spatial variability at ice divide. The topography position

of ice divide is probably linked also to spatial variability of snow accumulation in a feedback mechanism (see Drews et al., 2013; King et al., 2004; Matsuoka et al., 2015; Lenaerts et al., 2014). The uncertainty due to age scale and thinning function and ice divide migration must be tacking in account when is analysed the trend, uncertainties is not small amounts.

Pag 17 line 36-46 the interpretation of the reason of trend in accumulation differ from that hypothesis reported by Bertler et al., submitted paper.

Pag 18 line 1-4 the decrease of 6.6 cm/yr per century is not agree since 1950 with the paragraph 5.3 "Clausen et al. (1979) estimated the current (1954-1975) accumulation rate at the summit of Roosevelt Island to be 0.20 m w.e yr-1, whereas we here find the current accumulation rate (average of the last 50 years) to be 0.22±0.06 m w.e yr-"

Paragraph 5.1, Most of points reported are repetitions already pointed out in Methods and Result, see above for the comment, in particular for "we noted several strong volcanic imprints that seemingly have no counterpart in the WAIS Divide ice core data, and thus most likely originate from local West Antarctic volcanoes."

Pag 18 Line 31-44 The dipole effect change during the time, see Bertler paper, are this occurs in correspondence with presence or absence of RICE-WAIS volcanic event synchronization?

Pag 18 Line 38 "Absence of sulfate in RICE" with a higher background of 200 ng/g, exactly the opposite.

Pag 19 line 1-7 The tephra number of RICE is not unusual as presence compared to TALDICE or west Antarctic core, as the Authors have written few line after. Moreover, RICE present "several strong volcanic imprints that seemingly have no counterpart in the WAIS Divide ice core data, and thus most likely originate from local West Antarctic volcanoes.", but not tephra, this is very unusual if the volcanic event reported in Table 2 are true.

Page 19 line 3 "Only one exists within the last 2700 yr", but on the base of manuscript the tephra are two: Raoul 1964 and Pleiade 1252

Pag 19 Line 22 "longer-term trends are significantly different between the two locations" the text after describe similar trend with higher accumulation in the past respect to the present and change trend close at secular scale.

Paragraph 5.3 The result of RICE does not provide new information for the mass balance of RIS, taking in account the previous cores with similar SMB value and the high spatial variability of the rise and RIS.

---

## Referee Comment (RC2) · Anonymous Referee #2 · 23 Oct 2017

This manuscript provides a necessary timescale for the last 2700 years of the RICE ice core. It gathered many information obtained by Continuous Flow Analysis, especially chemical concentrations and acidity to identify annual cycles as well as volcanic peaks. The timescale is extensively compared to the WD2014 timescale which is one aspect not so clear from this manuscript: is this timescale tuned or not to WD2014 ? Finally, there is a discussion on the accumulation rate reconstruction and its evolution over the last 2700 years. Because of the strong uncertainties associated with the reconstruction, only the recent decrease can faithfully be discussed. While I believe that this manuscript will make an important contribution for following papers dealing with the RICE ice core, major comments should be addressed before its publication.

[Figure]

- The dating strategy is not clear. There is a mixing of layer counting constraints as well as use of volcanic peaks (+ nuclear bomb tests) but the uncertainty is only defined from layer counting at least on the upper part. It thus seems that the uncertainty is a bit overestimated ? Below 42.5 m, WD2014 seems to be taken as reference for the dating of the volcanic peaks as well as for adjusting the StratiCounter algorithm. Then WD2014 is used for "validation" of the timescale. By reading the methodology, it thus seems that there is something circular in the approach if WD2014 is used both for construction and validation of the timescale – could you please explain this better? - The methane constraint for the RICE timescale is not clear. First, it refers heavily to a paper that is in preparation (Lee et al., 2017). Second, the uncertainties associated with such tie-points are large and it is thus complicated to use them faithfully for timescale validation. Finally, the procedure mixing Monte-Carlo technique and manual adjustment is rather unclear. I imagine that everything will be in the Lee paper but details are missing to really make use of this part which is not very robust as written here. - Some parts are very long and not useful (most of section 3.2.3.2, part of section 3.3 on l. 10 or p. 15) – I suggest to reduce these sections and better concentrates on the method and associated uncertainties. - Accumulation rate is certainty an input of the firn model described in p. 13 while only forcing using a site temperature history is mentioned. It is very surprising that accumulation forcing is not mentioned here since one of the aim of this paper is to provide an accumulation scenario. We are thus expecting the use (or at least validation that everything is coherent with Dage or d15N measurements) of the accumulation rate scenario in the firn model. - What is the "model" mentioned in l. 45, p. 13 ? - The discussion on the ASL influence on the accumulation rate in the region is both in the "Results" section and in the "Discussion". This is also the case for other ideas that are repeated several times and a reorganization and simplification of the manuscript is needed. - The accumulation reconstruction should ideally have been compared to accumulation rate scenario used for the firn model as well as with water isotope profiles. It could strengthen the discussion and conclusion parts on the accumulation aspect that are rather short.

---

## Author Comment (AC1) · 8 Jun 2018

**Answers to reviewer 1:**

We thank the reviewer for the extensive comments and suggestions to the paper. Based on the reviewer comments, we have significantly restructured and revised the paper. We will not provide a detailed summary of these structural revisions below, but hope to be allowed to submit a revised version of the paper.

A point-to-point reply to the reviewer's comments are provided below (in blue), along with a short description of the adjustments to the paper relating to these comments.

**The manuscript is poorly written, too long with several repetition redundant, several contradictions between the same paragraph or other paragraphs, the data and the result are inaccurate present (see example tephra layers).**

The manuscript has been thoroughly restructured and revised, and redundancies have been removed.

We do not agree with the reviewer that we contradict ourselves, or that the data and result have been inaccurately presented, and we believe that this rely on misunderstandings. We hope that the reviewer agrees upon reading the revised version of the manuscript.

**The methods chapter are not well structured with several information reported two three times and not in the appropriate chapter as result and discussion. Most of the information about the methods is reported in the supplementary material, where are more clearly presented. The manuscript must be completely revised and shortening significantly.**

The manuscript has been thoroughly restructured and revised, and its content has been rewritten more concisely. Since the reviewers requested several expansions of the manuscript, the final version of the paper is, however, about the same length as the original version.

In the revised version of the manuscript, we have e.g. moved the section on timescale validation by comparison to WAIS Divide from "Method" to "Results". We believe that with this structural change, it should also become clearer that the RICE17 timescale is NOT synchronized to WAIS Divide, and that our matching of the two cores is only a basis for comparing the two timescales.

**Some references are uncorrected or mismatched.**

We have gone through all references and corrected these.

**Five accompanied papers of RICE core are submitted or in preparation, but their result are used to validate or as source of the result of the manuscript (ex. Lee et al., in preparation).**

It is correct that the Lee et al paper on methane matching of the RICE and WAIS Divide ice cores has not yet been submitted. We hope that this paper will be submitted very soon and will then be accessible as a discussion paper in Climate of the Past Discussions. In the present manuscript, we have revised the wording of the section on methane matching to improve its readability as a stand-alone text.

All other papers have been submitted and/or published now.

**To make this manuscript a significant contribution to the literature, the authors need to better justify their time scale and snow accumulation records.**

We hope that the reviewer will approve of the revised version.

**Clarify the use of the WAIS volcanic signal and methane with RICE17 chronology, in the text look like that is use as synchronisation (see 3.3.1.3), but several point is stressed that the accuracy is low and it is use only at posteriori as validation. All the process of comparison between RICE and WAIS must be clarify, it is repeated several time in different way.**

The RICE17 timescale is NOT synchronized to WAIS Divide.

We believe that part of the misunderstanding may be due to us inappropriately using the word "volcanic synchronization", where it more correctly should have been called "volcanic matching". This has been corrected in the new version of the manuscript.

**If the two records are synchronised by volcanic the age error must be the same closer the tie points, between one tie point to other can increase. The process must be revised.**

Since the two ice cores have not been synchronized (although they have been matched), the ages of the volcanic markers are not expected to be identical in the RICE and WAIS Divide records.

**The tephra layers where used to fix the chronology, but it is not reported the analysis of tephra particles (Raboul 1964 CE and Pleaide 1252 CE) and the analysis on WAIS ice record (up to now never published on my knowledge), that can be permit an unequivocal attribution.**

Results from geochemical analysis of the Pleiades tephra layers have been made available from the AntT database (http://antt.tephrochronology.org/I.html?id=AntT-15, http://antt.tephrochronology.org/I.html?id=AntT-16), which are now referenced in the text. Geochemistry of the Pleiades tephra horizon in the RICE and WAIS Divide ice cores is reported and discussed in Kalteyer (2015) and Dunbar et al (2010), which have now been included in the references:

Kalteyer, D.A., 2015. *Tephra in Antarctic Ice Cores*. Master Thesis, University of Maine. Available at: http://digitalcommons.library.umaine.edu/etd/2381/.

Dunbar, N.W., Kurbatov, A.V., Koffman, B.G., Kreutz, K.J., 2010. Tephra Record of Local and Distal Volcanism in the WAIS Divide Ice Core. 2010 WAIS Divide Science Meeting September 30-October 1, La Jolla, CA.

The 1965CE (Raoul) tephra layer from RICE and WAIS Divide is available from the Interdisciplinary Earth Data Alliance (IEDA) database (e.g. https://app.geosamples.org/sample/igsn/IESDW0026 and https://app.geosamples.org/sample/igsn/IESDW0016) with data reference Kurbatov (2015).

Results from the tephra analysis will be reported on in forthcoming publications, which we now refer to (see also comments below).

**The explanation because nssSO4 signal or acidity peak of major eruption reconnaissance in WAIS (Tambora, Unknow etc.) are not recorded in the RICE records is questionable, but Authors have attributed as unknown more than hundreds chemistry signal to volcanic eruption (123 event Table 2) and those are not observed in WAIS or others ice core in Ross Sea (Siple Dome, Taylor Dome, Talos Dome). Why RICE records is able to record 193 volcanic event, with all the problems pointed out in paragraph 3.3?**

We observe many small acidity peaks in the RICE records, which we in the previous draft related to volcanic events, despite these not being correlated to volcanic events in other ice cores. As the reviewer correctly points out, there is, however, a risk that we in Table 2 included acidity peaks derived from other events, e.g. extreme biogenic emission events.

For the new version of the manuscript, we have gone through the volcanic signatures in RICE and their matching to WAIS Divide, and have made the following changes:

- We established a new conductivity-to-Ca excess depth profile, directly calculated from the two CFA records. Previously, we only compared the two records visually, and refrained from calculating their differences, due to issues related to e.g. slight differences in depth assignment of the two records. However, as it turned out, having a directly calculated record of the non-sea-salt conductivity greatly simplified the volcanic matching between the two cores.

- With help of the new depth profile of non-sea-salt conductivity, we were able to match most of the prominent acidity peaks in WAIS Divide to volcanic peaks in RICE. The majority of the signals that were found in both cores corresponds to bipolar volcanic signals. This strengthens our trust in the reliability of the volcanic matching, since these are expected to deposit acidic material over an extended period, and therefore be most easily recognizable from the RICE records.

- The RICE acidity record still contains a large number of acid peaks that do not have a counterpart in WAIS Divide, but we refrain from stating that all these originate from volcanic events. Table 2 now only includes the acidity peaks that could be matched to a corresponding sulfate peak in WAIS Divide (65 acidity match points).

We consider the new volcanic matching to be very robust, and the main difference from the previous matching is that we have removed matches that we do not consider to be completely certain. However, the new depth profile of the conductivity-to-Ca excess allowed us to also identify a few new match points between RICE and WAIS Divide.

The text regarding the volcanic matching of the two cores has been revised accordingly.

**Black Carbon, on the base of figure 4 does not appear the best proxies of seasonal signal, H+ appear more conservative and less misleading of BC**

During development of the RICE17 timescale, we used all available proxies with annual signal, including black carbon and H+ (P. 9, line 26-27). Using multiple proxies will give us the most accurate timescale, as both records includes non-annual features that make annual layer interpretation based on a single record questionable.

**Authors report strong gradient in snow accumulation spatially ranged from 0.09 to 0.30 m we/yr and migration of the dome from 500 to 900 m. Can the Authors exclude any impact on the snow accumulation history due to migration of the dome ? and/or on thinning function?**

Over the last 2700 years, the Roosevelt Island ice divide has migrated only 500m from its present position (P13, L41), not 900m. We account for the migration of the ice divide in our derivation of the thinning function for the ice core, in that we vary the vertical velocity profile through time (P14, L11-20).

As suggested, we have added to the manuscript a brief discussion on how the migration of the ice divide could influence the obtained accumulation rate history:

*The recent period (~1500-1750 CE) of divide migration at Roosevelt Island may impact interpretation of the climate records from the RICE core. Present accumulation rates across Roosevelt Island show a distinct decrease on the downwind (western) side of the ice divide, with a gradient of ~0.5 cm/km yr$^{-1}$, although the trend is muted around the summit area. Ice recovered in the deeper part of the RICE core, deposited before divide migration, have originated west of the ice divide. Assuming a stable snowfall pattern through time relative to the divide, its migration would have caused reduced accumulation rates to be observed during the early part (until 1500 CE) of the RICE accumulation history. With an origin of the ice recovered in RICE of up to 500m west of the divide at time of deposition, our estimates of Roosevelt Island accumulation rates during this early period would therefore have a small negative bias of up to 0.25 cm/yr.*

*Correcting for the influence of ice divide migration, the main impact on the Roosevelt Island accumulation history is an earlier onset of the period with more rapid decrease in accumulation rates. The differences are small, however, and the overall pattern of trends in accumulation rate through time remains the same. In particular, ice divide migration has no impact on accumulation rate trends observed before and after the migration period.*

As mentioned, the impact on the accumulation history is small, and ice divide migration would have no impact on e.g. recent trends in accumulation rates.

**Paragraph 5.3 "Current mass balance. . .. "does not report any new valuable information for the mass balance of the RIS**

We have deleted this section from the manuscript.

**P. 3, line 44-47: How could explained stable ice divide flow with a migration of the ice divide position of around 500-900m?**

First of all, the ice divide has migrated only 500m, and this migration took place over a period of a few hundred years (roughly 1750-1500CE). During periods before and after, i.e. during the majority of the period, the ice divide flow was stable. We have accounted for the change in ice divide location over time in our modelling of the thinning function (P. 14, line 11-20).

We have revised the text to clarify this:

*To account for the changes in vertical strain rates at the drill site over time, we assumed the following divide-migration history, informed by the architecture of the Raymond stack (Fig. 6b): Until 500 years before ice core drilling (1512 CE), the divide was located 500 m east of the present position, as indicated by the position of the deeper Raymond arches. Since 500 years ago, the divide migrated westward, reaching its current position approximately 250 years ago (1762 CE), where it has since been stable.*

**P.4, line 33-34: RICE would be representative of East Ross Sea, not of Victoria Land, see accompanied RICE paper (Bertler et al, submitted)**

We have removed this sentence from the paper.

**Chapter methods: This part is too long and inappropriate as method chapter and most of the text must be moved to result chapter.**

As suggested by the reviewer, we have significantly restructured the paper.

The content of the methodology chapter is now split up into three parts: "Ice core processing and impurity analysis", "Constructing the Roosevelt Island Ice Core Chronology, RICE17, for the last 2700 years", and "Reconstructing past accumulation rates". We further have removed the section on comparison of the RICE17 timescale to WD2014 to the Results. We hope that this restructuring has made the paper more easily accessible.

**The does not provide information about the sample resolution along the core and the analysis performed and at which resolution ice (cm) and sample per year.**

Sample resolution of the various records is provided in Table 1. Further, as the CFA records are measured continuously, a discussion of their effective depth resolution is made on P.5, line 47 – P.6, line 5. We have made a slight addition to this paragraph to clarify the distinction between the two, so that it now reads:

*The CFA chemistry records are very densely sampled (1 data point per mm). Mixing in the tubing, however, as the meltwater sample travels from melthead to the analytical systems caused individual measurements to be correlated, and hence the effective depth resolution of the system is significantly less than the sampling resolution. This was especially the case for the RICE CFA set-up due to the relatively small fraction of total meltwater directed to the continuous measurement systems. Following the technique used in Bigler et al. (2011), we estimate the effective depth resolution for the CFA measurements to range from 0.8 cm (for conductivity) to 2.4 cm (for calcium) (see supplementary Table S1).*

The layer thickness is decreasing with depth, causing the number of independent samples per year to also change with depth (and varying between the individual chemical species). We therefore prefer to refrain from providing a general value for the number of samples per year in the manuscript.

Instead, in the section "Overview of the annual-layer counting strategy", we now mention the number of independent data points per year in the best resolved record in the very deepest part of the layer-counted timescale:

*At this depth, the annual layers are too thin (<6 cm, i.e. less than 8 independent data points/year in the best resolved records) for reliable layer identification in data produced by the RICE CFA set-up.*

**Percentage of missing record of CFA example must be reported and show.**

We have added the following paragraph to the paper (P. 5, line 46):

*Core breaks and/or contamination in the system caused some sections of missing data. The percentage of affected core varied between chemistry species, ranging from <1% (BC) to 15% (H+), the majority being small sections of missing data that did not severely impact annual layer interpretation of the records.*

**Most of the information about methods is relegate in the supplementary info.**

We are not sure whether the reviewer would like us to move more of the material in these three paragraphs into the supplementary, or if he/she would like us to include some of the supplementary material into the paper itself.

In our endeavor to shorten the paper, we have decided to keep the division between the two more or less as is. We have transferred some of the technicalities on the StratiCounter set-up to the supplementary.

However, we have included into the main paper the discussion of the annual layer signals observed in the various CFA records, as we consider these to be of general interest.

**P. 6, line 21-24: This paragraph present contraindiction in several points, along the entire text, correlation between RICE and WAIS "volcanic event" are used or not to tune the RICE scale? Ex. See line 29-30 of pag 6**

Apart from the top 42m (where the timescale can be constrained by historical events), the RICE17 timescale is a fully independent layer-counted timescale, i.e. it has not been tuned to the WAIS Divide timescale. The correlation to WAIS volcanic events is only used for subsequent comparison of the timescale to WD2014. We recognize that this essential aspect was not sufficiently clear from the manuscript, and, as previously mentioned, we have made several revisions to the paper for clarification, including the following:

- We have moved the comparison of the RICE17 timescale to WD2014 to the results section.
- A few instances of incorrect use of the word "synchronizing" has been removed, and the word "tiepoints" has been replaced with "matchpoints".

We have further revised the section "Overview of the annual-layer counting strategy" on page 6, so that it now reads as follows:

*The uppermost section (0-42.34 m) of the core was dated by manual identification of annual layers in records of water isotopes and ice impurities from the RICE main core as well as the RICE-12/13B shallow core. For this most recent period, several distinct marker horizons from well-known historical events were used to constrain the chronology.*

*Below 42.34m (1885 CE), the timescale was augmented using the StratiCounter layer-counting algorithm (Winstrup et al. 2012) applied to multiple CFA impurity records from the RICE main core. A previously-dated tephra layer at 165 m (Pleiades; 1251.6±2 CE according to WD2014) was used to optimize the algorithm settings, but other than that, RICE17 is a fully independent layer-counted ice-core chronology.*

**Page 6, line 24: The Raoul tephra is a unequivocal volcanic event or not?**

The Raoul tephra is an unequivocal tephra horizon, but as it is located above 42m it is not mentioned here. In the revised version of this paragraph (see above), we have removed the reference to unequivocal volcanic events.

**Page 6, line 31-32: The layer counting stops at 343.72, because the annual layer is to fine (<6cm), to identify seasonal signal needs at least 10-12 sample per year, a graph showing the number of sample analysed per year must be show, from the surface to the 344m**

As the CFA measurements are made continuously on the melt water stream, the notion of the number of samples per year is not a straight-forward measure for such data sets. Further, due to different amount of mixing in the various chemistry melt-water lines in the CFA set-up, the effective depth resolution differs between data sets. Instead of showing these in a graph, we have elected to mention the number of data points per year in the best resolved record in the very deepest part of the layer-counted timescale (P.6, line 31-32), as also mentioned above.

In the bottom part, we have less than 8 independent data points/year in the best resolved records. We note that this number is less than ~10-12 independent samples per year, mentioned by the reviewer to be required for annual layer identification. This lower number is somewhat counteracted by the near-continuity of the CFA records, which was employed on a depth scale with ~1mm resolution (i.e. resulting in 60 correlated samples/year in the deepest part of the timescale).

**Page 6, line 44-46: The record of overlap section must be shown to see the ratio noise/signal in the two cores**

As the reviewer also requests a significant shortening of the paper, we have elected not to extend the manuscript with such figure.

**Page 7, line 10: Several other records also displayed annual variability, but much less reliability" Why do use BC instead of H+ or both**

This is a misunderstanding. For all depth intervals, annual layer interpretation in the RICE17 chronology is based on the complete set of available chemistry and isotope records, as also mentioned in the section "Overview of the annual-layer counting strategy".

Our intention with the remark on page 7, line 10, was simply to note that some data series displayed a more reliable annual signal than others. In the deeper part, BC is the record with the most reliable annual signal, partly because of the high resolution of the record. Yet, also here annual layer counting is based on all CFA records, including H+.

**Page 7, line 11-16: The peak of proxies seasonality are quite different in time (isotope versus sea ice proxies, or photolysis), and most depend from the occurrence of snow fall. The use of ERA model does not look appropriate and the reference is still not published**

We observe from the data records that water isotopes and acidity signals peak simultaneously in most years (Figure 4), which we interpret as the maximum annual temperature and lowest sea ice extent taking place approximately at the same time. We have assigned the depths of these peaks to correspond to January 1st. Seasonal variations in snowfall will influence the precise depth location of the peaks in the various records, but it will do so similarly for all records.

In the text, we have shortened and reorganized the section, and removed the reference to ERA-interim data, so that the paragraph now reads:

*Summers could be identified as periods with high stable isotope ratios, high concentrations of nss-$SO_4^{2-}$ and associated acidity [originating from phytoplankton activity in the surrounding ocean during summer (Legrand et al. 1991; Udisti et al. 1998)], and low iodine concentrations [due to summertime photolysis of iodine in the snowpack (Frieß et al. 2009; Spolaor et al. 2014)]. Layer marks were placed according to the depths of concurrent summer peaks in water isotope ratios, nss-$SO_4^{2-}$ concentrations, and acidity levels, and assigned a nominal date of January 1st.*

**Page 8, line 17-21: Geochemical composition of the tephra at RICE-WAIS and source must be show before any attribution of a tephra never reported in Antarctica before (Raoul 1964)**

Results from the geochemical analysis of the Raoul tephra from RICE and WAIS Divide is available from the IEDA database, with data reference Kurbatov et al 2015 (which unfortunately had dropped out of the previous version of the manuscript):

Kurbatov, A. V. et al., 2015. Major element analyses of visible tephra layers in the Roosevelt Island Climate Evolution Project ice core (Antarctica). *Interdisciplinary Earth Data Alliance (IEDA)*. Available at: https://app.geosamples.org/sample/igsn/IESDW0025

The attribution to Raoul will be reported in a paper currently in preparation.

We have added these references to the text:

*A couple of volcanic horizons in RICE during this most recent part could be unambiguously related to well-known volcanic eruptions. Rhyolitic tephra located between 18.1-18.2m (Kurbatov et al. 2015) was found to have a similar geochemical composition as a tephra layer found in the WAIS Divide core, with a depth corresponding to late 1964 CE. The tephra likely originates from Raoul Island, New Zealand (Wheatley et al, in prep), which erupted from November 1964 to April 1965. This is consistent with the RICE17 chronology, according to which the tephra is located in early 1965 CE (Table 2).*

**Page 8, line 21-36: It is not clear why some sulphate deposition is attributed to eruption and others no, and correlated to WAIS**

For this most recent part of the timescale, for which historical volcanic eruptions constrained the timescale, we used all available unambiguous volcanic horizons. Apart from the Raoul tephra, however, we were only able to identify two volcanic horizons during this time interval: Santa Maria and Krakatau. We note that the ages of these volcanic horizons are constrained from historical records, and not because of synchronization to the WAIS Divide core, but agree with the reviewer that this was not clear from the previous version of the manuscript.

We have shortened this section, removed the references to WAIS Divide ages, and revised the wording to clarify these aspects. It now reads:

*Only two volcanic eruptions could be unambiguously identified in the acidity records over this period, namely the historical eruptions of Santa Maria (1902 CE; 37.45m) and Krakatau (1883 CE; 42.34m). These two horizons were used to constrain the deeper part of the manually-counted interval of RICE17, which terminates at the Krakatau acidity peak (Table 2). Deposition age of volcanic material for these events was assumed identical to observed in the WAIS Divide ice core (Sigl et al. 2013). Imprints from other large volcanic eruptions taking place during recent historical time, such as Agung and Pinatubo, did not manifest themselves sufficiently in the RICE records to be confidently identified.*

**Page 8: Analysis of comparison between manual and automated annual counting must be performed and show**

The manual counting performed was only a rough preliminary counting, with the sole purpose of producing a set of templates for the annual layers in the chemistry records, as required for initialization of StratiCounter. We therefore refrain from performing an analysis between the manual counts and the automated layer counts in the paper.

We have revised the wording in this section to ensure that the preliminary nature of the manual layer counts is better conveyed to the reader:

*StratiCounter was initialized based on a rough set of manual layer annotations in a short section of the data (40-150m). The manual annotations were used to produce a template for an annual layer in the various impurity records. To increase the independence of the StratiCounter timescale from the preliminary manual interpretation, in a final step the entire timescale was reevaluated using an improved set of layer templates derived from the output of the algorithm itself.*

In order to shorten the paper, these and other details regarding how the StratiCounter software was run has been moved to the Supplementary.

**Page 9, line 14-15: BC is used to date the 90% of the core analyzed, but on base of fig 4, is not the best proxies of seasonality, also as reported by Author pag 7 line 10. At line 26-27 is reported different use of the proxies the Authors contradict themselves**

This is a misunderstanding. We always use all available proxies for the dating (see e.g. P9, L25-27), see also our answer to the question reg. P7, L10.

We do not have IC or ICP-MS measurements (of e.g. S, as shown in figure 4) in sufficiently high resolution that these can be used for identifying layers below 40m. Fortunately, in the deeper part of the core, the annual signal in BC is better than for the top part, and, due to its high effective resolution, it maintains a good annual signal with depth.

We have now included a discussion on the annual layer signal in the various chemistry series in the main paper, along with figure S2 (now: figure 7), which shows the evolution in annual signal with depth of the various CFA species. See an updated version of the figure below.

The reason for the misunderstanding may lie in the paragraph on P9, L15-16, where we mention that we use the peak in BC for the annual markers. However, this is simply a matter of where the layer marks are placed, and it does not imply that BC is the only chemistry series used to identify the layers. To avoid misunderstandings, we have removed this sentence from the revised version of the paper.

[Figure]

*Figure 7: Average annual signals of 2 successive years in **a-c)** RICE acidity (H⁺), **d-f)** conductivity (Cond), **g-i)** calcium (Ca²⁺), and **j-l)** black carbon (BC) over three 100-year periods, calculated under the assumption of constant snowfall through the year. The line shows monthly-averaged median value of measured concentrations, and colored area signifies the 50% quantile envelope of the value distribution.*

**Page 9, line 39-40: The geochemistry is not show; Pleiades volcano is not West Antarctica, but in Northern Victoria Land; Kurbatov et al, 2015 is not reported in reference, and it is not present in any database as reference for tephra layer reported; the 1252 tephra attributed to The Pleiade was iscovered the first time at Talos Dome and dated at 1254+-2 by Narcisi et al, 2001; on my knowledge the analysis of the tephra at WAIS is still not published**

We have changed "West Antarctica" to "Northern Victoria Land".

Kurbatov (2015) has been added as reference, see also previous comments.

We have elected to shorten the discussion involving tephra in the manuscript, as this will be the focus of forthcoming publications. We have further moved the discussion about the Pleiades tephra layer to the Results, as it was primarily used during the matching of RICE to the WAIS Divide ice core. We have revised the text, and added additional references for the geochemistry analyses of the tephra.

Results from geochemical analysis of the Pleiades tephra layers have been made available from the AntT database (http://antt.tephrochronology.org/I.html?id=AntT-15, http://antt.tephrochronology.org/I.html?id=AntT-16), which are now referenced in the text. Geochemistry of the Pleiades tephra horizon in the RICE and WAIS Divide ice cores is reported and discussed in Kalteyer (2015) and Dunbar et al (2010), which have now been included in the references.

We also mention that tephra of similar composition has been found in Talos Dome (Narcisi et al, 2001), and TALDICE (Narcisi et al, 2012).

The section now reads:

*A visible tephra layer was found in RICE at 165m depth, with a RICE17 age of 1251.5±13 CE. Geochemistry of the tephra particles is consistent with an eruption from the Pleiades (Kalteyer 2015), a volcanic group located in Northern Victoria Land, Antarctica (Fig. 1). Tephra of similar geochemistry has been found in several other Antarctic cores dated to approximately the same time, including WAIS Divide (1251.6±2 CE; Dunbar et al. 2010) and Talos Dome/TALDICE (1254±2 CE; Narcisi, Proposito, and Frezzotti 2001; Narcisi et al. 2012). The Pleiades tephra horizon allowed a firm volcanic matching of the RICE and WAIS Divide ice cores at this depth (Fig. 8).*

**Page 10, line 5-18: RICE site is farer than other cores (WAIS-Byrd-Siple Dome and Talos Dome) from "many active volcanoes", it is very difficult to understand why RICE record is able to identify volcanic eruptions those are not identified in the ice cores of the region with much lower ratio in noise/signature due to marine biogenic sulphate emissions**

This is a good point. In the previous version of the manuscript, we had included several minor acidity peaks. There is, however, a risk that we had included acidity peaks of non-volcanic origin, e.g. extreme biogenic emission events. In the new version of the manuscript, we only include RICE acidity peaks that could be matched to the WAIS Divide volcanic record.

We note that one reason why we might observe more regional volcanoes in the RICE core is our use of acidity records to identify volcanic eruptions. By using acidity rather than sulfate, we might better observe the signals from regional eruptions, as discussed on P. 11, L1-9. This section has been moved to the discussion.

**Page 10, line 19-25: Methane gas synchronization less precise than volcano matching, after 4 line "but methane it is better than volcanic matching", clarify**

As described on P10, L5-31, the two methods are complementary: Methane matching is less precise (i.e. relative age differences is much better resolved using volcanic matching), but it provides better absolute age control, since there is less risk of misalignment of the records. The text has been revised to clarify this aspect.

**Paragraph 3.3.1.1: The use of acidity and ECM to detect volcanic signal is not a new tools. Hammer have used H+ in 1980 as proxies of volcanic signal**

Agreed. However, the volcanic record here is based on direct measurements of H+ in the ice core, whereas H+ used by Hammer (1980) was estimated from the ECM signal. Further, the volcanic proxy developed based on non-sea-salt conductivity is new. We have changed the title of the paragraph to "New and traditional ice-core tracers for volcanic activity" to reflect that it is not new to use ECM to detect volcanic signals.

**Page 10, line 43: A resolution of 9.5 cm (about 4 sample per yr) is very low to observe the seasonal variation, but enough to detect the important volcanic signal like Tambora, Kauve etc present a signal for 2-3 yr in Antarctic cores (from 8 to 12 sample)**

We agree with the reviewer that with this resolution, we would expect to be able to see some signal in discretely-sampled S record from large bipolar volcanoes that deposit material over several years. However, we observe from the retrieved sulfate records that it is usually very hard to distinguish volcanic eruptions in these. This is likely due to the large inter-annual variability in biogenic sulfate influx, shading the volcanic signatures. The volcanic eruptions are easier to identify from higher-resolution records, such as acidity.

We have added the following to the paper:

*Resolution of the discretely-sampled sulfur record was too low (below 67m: 5 cm, i.e. less than 4 samples/year), and even large volcanoes only left a vague imprint in form of slightly increased sulfur levels over a multi-year period (Fig. 8a). Detection of volcanic horizons in the RICE core therefore primarily relied on two new high-resolution tracers for volcanic activity; direct measurements of total acidity and estimated non-sea-salt liquid conductivity.*

To illustrate our point, we have added the sulfur records to figure 7 (now: figure 8), see below, so that the reader can verify our statements. We note that exactly this section is actually where the sulfur record displays the most distinct volcanic signal.

[Figure]

*Figure 8: a) The RICE volcanic proxy records: non-sea-salt-sulfur (nss-S; orange), ECM (purple), acidity (H⁺; red), and non-sea-salt conductivity (nss-cond; blue) based on the conductivity-to-calcium excess (grey and green). b) Matching of the RICE records to the WAIS Divide non-sea-salt sulfur record (Sigl et al. 2015). Vertical bars indicate volcanic match points (Table 2), with the red bar being the Pleiades tephra horizon (1251 CE).*

**Page 10, line 25-32: On the base of figure 7a and Table 2 the volcanic events identified in RICE at 158.15m and 160.77m do not present any clear evidence in ECM, H+, due to the high background of ssSO4 at RICE. It is very difficult to understand why a site with this high noise/signal ratio could record local eruption sulphate does not observed at other site core.**

We agree that these two volcanic eruptions are hard to identify from the figure. During the process of re-evaluating the volcanic matching between RICE and WAIS Divide, we have removed the volcanic event at 158.15m, which was not matched up to a volcanic horizon in WAIS Divide. In addition, we removed the volcanic event at 159.5m, as we did not consider this horizon to be a sufficiently clear-cut match to WAIS Divide, when considering all available volcanic proxies.

We have added to figure 7 (now: figure 8, see previous comment) the newly developed non-sea-salt conductivity record, which facilitates identification of volcanic eruptions from the difference between the conductivity and calcium records.

At 160.77m, the non-sea-salt conductivity record has predominantly positive values over a significant section, indicative of a volcanic event. This is backed up by significant, albeit small, peaks in the other volcanic proxies. We note that the age differences between this and surrounding volcanic events are identical to those obtained from matching this peak to the significant non-sea-salt sulfur peak in WAIS Divide at 184.5m. We hence consider this peak to be well-qualified as match point between the two cores.

We note that we do not believe that RICE records volcanic sulfate peaks not observed at other sites (such as WAIS Divide). However, by identifying volcanic eruptions based on the acidity rather than sulfate, we may be able to better identify regional eruptions, which also deposit other acids than sulfuric acids.

**Paragraph 3.3.1.2 The Pleiades horizon is discussed in several part of the manuscript (parag 3.2.1, 3.2.4, and 3.3.1.2), but without provide any evidence on the base of geochemistry analysis. This tephra layer was reported for the first time in Talos Dome 1996 ice core and dated by Narcisi et al, 2001, at 1254+-2 and attributed as source to Melbourne Volcanic Providence, probably "The Pleiades", located about 250km from Talos Dome. This tephra was than identified in Siple Dome and Taylor Dome by Dunbar et al, 2003. Moreover, Narcisi et al, 2012 pointed out that at TALDICE (a core drilled from 2004 to 2007) is present the tephra of 1254 as TD87a (86.2m depth) close in composition to the previous found in the ice core on 1996. However a subordinate set of glass shards (TD87b) is also trachytic but with a chemical signature inconsistent with The Pleiades products. Mount Berlin could thus be a suitable source of ash (Narcisi et al, 2012). Moreover an other tephra layer TD85 at 84.37m depth younger than 25 yr has been reported by Narcisi et al, 2012 and the suggest source is Mt Melbourne volcanic province. Without any geochemical analysis is impossible attribute unequivocally the tephra found in RICE"**

We now discuss the tephra horizon attributed to the Pleiades only in a single paragraph in the manuscript. In this paragraph, we now also mention that it has previously been found in other ice cores in the region, and provide additional references, see our reply to previous comments.

We prefer to not go into the tephra geochemistry in details here, as this will be the topic of forthcoming publications. As previously mentioned, geochemical analysis of this tephra layer is now published in the AntT database.

**Page 11, line 41-42: If the tephra layer identified is 1252+-2 yr, why use 1252+-13 for this horizon in RIC17?**

RICE17 is an independent timescale, and thus the uncertainty on the age of this tephra horizon reflects the uncertainty in layer counting in the RICE records.

**Page 12, line 9-18: See above, more than 170 volcanic event most of them never see in closer core**

This number has been reduced, see our previous comments.

**Page 12, line 19-42: Volcanic event and Methane records are used for Synchronization or validation?**

They are only used for validation, see our previous comments.

**Page 13, line 9-10: Which is the gas-age at RICE? And compared to closer site as Siple and WAIS?**

For the period discussed in the paper, Δage values for RICE is 145-171 years. This is slightly less than Δage values for the WAIS Divide ice core (174-206 years), and somewhat smaller than those for Siple Dome (233-255 years). TALDICE has significantly larger values of Δage.

**Page 13, line 24-29: What is the source of surface temperature of -22oC? Why is used this instead of 27.4oC, this value is also proposed in accompanied paper of Bertler et al, submitted. Why do you use a warmer temperature of 5.4oC? Which implication on density and thinning model using 5.4oC warmer?**

We thank the reviewer for pointing out this inconsistency regarding the current surface temperature at RICE. The -27.4°C temperature is derived from ERA-interim data, obtained for a location slightly south of the Roosevelt Island. It seems that there is a large temperature difference between this location and the RICE drill site, with borehole temperatures and AWS data from the RICE site showing an average temperature of -23.5°C. This difference in mean temperature based on the two methods is also discussed in Bertler et al. 2018 (published version). In the new version of the manuscript, we refrain from mentioning the too-cold ERA-interim temperatures.

As the surface temperature of -23.5°C was always used for the density and Δage modelling, these are correctly calculated in the manuscript.

**Page 13, line 30-40: Kingslake et al, 2014, instead of Raymond**

We have moved the Raymond 1983 reference to immediately after the phrase "Raymond arches".

**Page 13, line 40-41: The recent migration of ice divide, could be attributed to change in snow accumulation variability at ice divide? Due to the snow accumulation variability between the flank of the ice divide, which is the influence have on snow accumulation record and thinning function of RICE core? Kingslake et al., 2014 report that near-surface strain rates are compressive at ice divide than in the flanks 90% higher at RICE. The migration of the ice divide respect to Raymond Bump position indicate a role of temporally changing in spatial snow accumulation distribution, as well as the role of along-ridge flow is un-clear and hampers a solid interpretation about thinning function and snow accumulation records**

Neither the high frequency (shallow, <100m) or low frequency (deep, >50m) profiles suggest a significant change in accumulation pattern over time that could have driven the divide migration. Therefore, we believe the divide migration is likely caused by ice dynamics, potentially caused by changes in buttressing by the surrounding Ross Ice Shelf. The relatively small changes in divide position over time, however, suggest that neither has changed significantly over the past 2700 years. Since the reviewer has recommended a significant shortening of the manuscript, we have elected not to expand the discussion of the thinning function.

The applied thinning function takes account for the divide migration (see previous comments).

It is correct that the large gradient in accumulation rates across the ice divide could influence the obtained accumulation record from the ice core, and as previously mentioned, we now estimate this in the paper. Since the divide has only migrated a short distance, this does not significantly impact the derived trends in the accumulation rates.

**Fig 8: The pRES measurement (Kingslake et al., 2014) was at ice divide and does not follow the Raymond bump features as reported in figure 8b**

Correct. That is why we vary the vertical velocity profile through time (P14, L11-20).

**Page 14, line 11-20: On the base of which data the Authors construct a vertical velocity profile along the Raymond Bump?**

As stated, we are assuming that the ice in the core was not directly beneath the divide from 700BCE to 1450CE. The vertical velocity profile is allowed to change over time, based on a linear combination of the measured vertical velocity profiles from the flank and the topographic divide.

In the revised paper, this section has been clarified.

**Page 15, line 14: "Control point… of atmospheric oxygen isotope" at page 12 "Given the stability of the d18Oatm record over the last millennia, the synchronization was solely constrained by the observed variability in the methane records" as in several other part of the text none coherence exist between the paragraphs and some times also in the same paragraph"**

We used both $\delta^{18}O_{atm}$ and $CH_4$ for matching the two cores. However, given that the levels of $\delta^{18}O_{atm}$ were very stable over the last 3000 years, using this record did not provide many constraints to the synchronization. Essentially, therefore, the matching was based on the methane records. We have added the following to the text to make this clearer:

*The feature matching routine employed discretely-measured records of methane as well as isotopic composition of molecular oxygen ($\delta^{18}O_{atm}$). Over recent millennia, however, the $\delta^{18}O_{atm}$ concentrations are stable, and hence provided minimal matching constraints.*

**Page 15: Along all the paragraph it is not clear the process of adjustments of the counting layer respect to matching between RICE and WAIS**

RICE17 is an independent timescale, and hence we do not adjust the number of counted layers between match-points to fit the WD2014 timescale.

**Page 16, line 13-16: High internal-annual variability in snow accumulation is normal issue (see eg Eisen et al, 2008 and reference within), 1.3% is very low value**

We have removed this section from the manuscript.

**Page 16, line 40-45: The three accumulation record of snow accumulation must be shown in the overlap time, the correlation coefficient of 0.85 and 0.87 indicate that the RICE annual are representative, but at pluriannual scale (see Eisen et al., 2008; Frezzotti et al., 2007). The comparison of the three cores can confirm only the stability of snow in the overlap time, not at secular or millennium scale**

Agreed. However, while we can only say for certain that the accumulation is spatially consistent within the time period covered by these three cores, we still believe that the consistency between these records in the overlap period provides a basis for our general statement that the strong correlation between these indicates spatial consistency in accumulation rates.

We have added to the paper a figure comparing the accumulation records obtained from the three cores during their overlap period, see below. The three records show very similar inter-annual variability.

[Figure]

***Figure 12:*** *Accumulation reconstructions for the three Roosevelt Island ice cores.*

**Paragraph 4.3.2: "The inflection point in accumulation of fig 9 with a trend in decrease is closer to age of the hypothesis of the stabilization of the ice divide at present position 1450 EC ( Pag 14). The uncertainty of change in accumulation must be taking in account also the spatial variability at ice divide. The topography position of ice divide is probably linked also to spatial variability of snow accumulation in a feedback mechanism (see Drews et al., 2013; King et al., 2004; Matsuoka et al., 2015; Lenaerts et al., 2014).**

Firstly, we note that there was a small error in the previous version of the draft regarding our inferred timing of stabilization of the ice divide at its present location – we expect the migration to start 500 years ago, i.e. in 1512CE, not in 1450CE, and arrive to its present location 250 years ago (1762CE).

With these new ages, there is an even more significant difference in timing between the onset of a negative trend in the accumulation rates (1250CE) and the onset of divide migration (1512CE). As previously mentioned, the migration of the ice divide would give rise to a small negative bias in derived accumulation rates for the first part of the period. However, it does not significant impact the derived trends in the accumulation history.

Given the different timing as well as the small effect on the accumulation rates, we do not believe that migration of the ice divide has been the main cause behind the observed changes in accumulation rate over time.

**The uncertainty due to age scale and thinning function and ice divide migration must be tacking in account when is analysed the trend, uncertainties is not small amounts**

We note that the uncertainty associated with the change in ice divide migration (~0.25cm/yr) is small compared to the uncertainty in thinning function (~2 cm). Similarly, the small (correlated) uncertainties in the age scale do not significantly impact trends in the derived accumulation rates (P. 17, L10-15).

We have taken the approach to estimate uncertainty on the trend estimates based on a linear regression to the most likely accumulation rate history. We note that this is much more reasonable now that the uncertainties on the thinning function have been significantly reduced. Due to the high variability of the accumulation rate over time, we believe that this approach provides reasonable estimates for the uncertainty in observed trends.

**Page 17, line 36-46: The interpretation of the reason of trend in accumulation differ from that hypothesis reported by Bertler et al., submitted paper**

We now mention that the recent decline in accumulation at Roosevelt Island may be due to increased sea ice extent in the Eastern Ross Sea, as put forward by Bertler et al (2018).

**Page 18, line 1-4: The decrease of 6.6 cm/yr per century is not agree since 1950 with the paragraph 5.3 "Clausen et al. (1979) estimated the current (1954-1975) accumulation rate at the summit of Roosevelt Island to be 0.20 m w.e yr-1, whereas we here find the current accumulation rate (average of the last 50 years) to be 0.22±0.06 m w.e yr-**

The value of 6.6cm/century is derived from the RICE accumulation rate history, which shows a distinct decrease in recent accumulation rates. We agree that this may appear contradictory to the old cores finding slightly smaller accumulation rates than the present value obtained from RICE. The difference between accumulation rates in the two cores, however, is caused by the spatial variability in accumulation rates, with the RID75 core being drilled at a location with slightly lower accumulation rates. The two statements are therefore not contradictory.

**Paragraph 5.1: Most of points reported are repetitions already pointed out in Methods and Result, see above for the comment, in particular for "we noted several strong volcanic imprints that seemingly have no counterpart in the WAIS Divide ice core data, and thus most likely originate from local West Antarctic volcanoes.**

We have significantly reorganized the paper, so that we now in Discussion discuss the implications of using total acidity instead of sulfate, and how it impacts the volcanic record from RICE.

**Page 18, Line 31-44: The dipole effect change during the time, see Bertler paper, are this occurs in correspondence with presence or absence of RICE-WAIS volcanic event synchronization?**

It would be interesting to investigate the effect directly from a comparison of the two volcanic records, but it not possible to say from our matching between the two ice cores. Given the general challenge of establishing volcanic march-points in the cores, the identification of these partly depended on the density of reliable matches in the surrounding core sections. In other words, with a higher density of reliable match points, the more additional acidity peaks were sufficiently convincing to be annotated as a match-point. This positive feedback loop significantly influenced the frequency of identified match-points, overwhelming any effect caused by the dipole strength.

**Page 18, line 38: "Absence of sulfate in RICE" with a higher background of 200 ng/g, exactly the opposite**

We have changed this to:

*Absence of volcanic signal in the RICE core*

**Page 19, line 1-7: The tephra number of RICE is not unusual as presence compared to TALDICE or west Antarctic core, as the Authors have written few line after. Moreover, RICE present "several strong volcanic imprints that seemingly have no counterpart in the WAIS Divide ice core data, and thus most likely originate from local West Antarctic volcanoes.", but not tephra, this is very unusual if the volcanic event reported in Table 2 are true**

We have removed the sentence about the number of visible tephra layers in RICE.

**Page 19, line 3: "Only one exists within the last 2700 yr", but on the base of manuscript the tephra are two: Raoul 1964 and Pleiade 1252**

Correct. This paragraph has been removed.

**Page 19, line 22: "longer-term trends are significantly different between the two locations" the text after describe similar trend with higher accumulation in the past respect to the present and change trend close at secular scale**

We agree that this is not formulated very clearly. The text has been rewritten to clarify that while part of the long-term trends differ at the two locations (increasing at RICE until 1250CE, decreasing at WAIS Divide), the two records also show some similarities. For instance, the decline in WAIS accelerates around the same time that RICE accumulation rates starts to decline.

**Paragraph 5.3: The result of RICE does not provide new information for the mass balance of RIS, taking in account the previous cores with similar SMB value and the high spatial variability of the rise and RIS**

We have deleted this paragraph from the paper.

---

## Author Comment (AC2) · 8 Jun 2018

**Answers to reviewer 2**

We thank the reviewer for the comments and suggestions to the manuscript. Based on the reviewer comments, we have significantly restructured and revised the paper. We will not provide a detailed summary of these structural revisions below, but hope to be allowed to submit a revised version of the paper.

In the revised version of the manuscript, we have moved the section on timescale validation by comparison to WAIS Divide to "Results". We believe that with this structural change, it should become clearer that the RICE17 timescale is NOT synchronized to WAIS Divide, and that our matching of the two cores only forms basis for a comparison between the two timescales.

A point-to-point reply to the reviewer's comments are provided below (in blue), along with a short description of the adjustments to the paper relating to these comments.

**The timescale is extensively compared to the WD2014 timescale which is one aspect not so clear from this manuscript: is this timescale tuned or not to WD2014?**

The timescale is not tuned to WD2014. This has been clarified in the current version of the manuscript.

**Finally, there is a discussion on the accumulation rate reconstruction and its evolution over the last 2700 years. Because of the strong uncertainties associated with the reconstruction, only the recent decrease can faithfully be discussed.**

We found an error in the calculation of uncertainty on accumulation rates in the previous version of the manuscript, which has now been corrected. With this correction the uncertainties have been significantly reduced, allowing us to discuss the observed changes in accumulation over time with much more confidence.

**The dating strategy is not clear. There is a mixing of layer counting constraints as well as use of volcanic peaks (+ nuclear bomb tests) but the uncertainty is only defined from layer counting at least on the upper part. It thus seems that the uncertainty is a bit overestimated?**

The top part of the RICE17 timescale is manually counted while using constraints from historical events (incl. volcanic peaks) observable in the RICE ice core records. These constraints reduce the uncertainty of the most recent part of the timescale (corresponding to the upper 42.5m of the core). As we account for age constraints when developing the uncertainty of the age scale, we do not consider the uncertainty in the top part to be overestimated. We have revised the text to clarify this point, and now write:

*A confidence interval was assigned to the timescale by classifying layers as certain or uncertain (Fig. 4), while accounting for age constraints from marker horizons.*

In the deeper part, for which the timescale has not been constrained, we use the statistically-produced StratiCounter results for the age uncertainty.

**Below 42.5 m, WD2014 seems to be taken as reference for the dating of the volcanic peaks as well as for adjusting the StratiCounter algorithm. Then WD2014 is used for "validation" of the timescale. By reading the methodology, it thus seems that there is something circular in the approach if WD2014 is used both for construction and validation of the timescale – could you please explain this better?**

We hope that our approach is better explained in the revised version of the manuscript.

First of all; RICE17 and WD2014 are independent timescales.

In the top part, we used historical events to constrain the RICE timescale. Below 42.5m, the timescale has not been constrained.

The only "circularity" in our approach is that we used the Pleiades tephra horizon in the RICE core to select the optimal settings for StratiCounter. This tephra horizon has previously been dated to 1251+-2 years according to WD2014, but is also found in other ice cores with similar age. It happened to be that one version of the StratiCounter output produced exactly the same age of the Pleiades tephra horizon as found in WD2014, and we decided to select those settings for the algorithm.

However, we note that our adjustments to the StratiCounter settings gave rise to very small age differences (<10 years) at the depth of this tephra layer, and that all the StratiCounter-derived timescales were in very good agreement with each other. Hence, for all practical purposes, the RICE17 and WD2014 timescales are independent of each other.

We recognize that this essential aspect was not sufficiently clear from the previous version of the manuscript, and we have made several revisions and structural changes to the text for clarification.

**The methane constraint for the RICE timescale is not clear. First, it refers heavily to a paper that is in preparation (Lee et al., 2017).**

The details of the approach used for matching the RICE and WAIS Divide ice cores using their respective high-resolution methane records will be described in the Lee et al paper, which is ready to be submitted any day now, and will soon be accessible as discussion paper in Climate of the Past Discussions. In the present manuscript, we have revised the wording of the section to improve its readability on its own.

**Second, the uncertainties associated with such tie-points are large and it is thus complicated to use them faithfully for timescale validation.**

It is true that the uncertainties associated with the individual methane match-points are relatively large. It should be noted, however, that for the RICE and WAIS Divide ice cores, these uncertainties are much smaller than for most other Antarctic ice cores (P13, L9-14).

Despite of the uncertainty on the individual match points, it is worthwhile to validate the RICE17 timescale to WD2014 based on the methane synchronization of the two cores. For instance, in Figure 6 (now: Figure 9) we observe that all methane match points below 280m are associated with older ages in WD2014 than in RICE, which corroborates the volcanic synchronization between the two ice cores.

Due to the associated uncertainties, we only use the methane match points for validation of the absolute ages of the RICE17 timescale.

**Finally, the procedure mixing Monte-Carlo technique and manual adjustment is rather unclear. I imagine that everything will be in the Lee paper but details are missing to really make use of this part which is not very robust as written here.**

After matching the two methane records using a Monte Carlo approach, we observed that the two records in this top part would fit slightly better if the match-points were slightly adjusted. These adjustments fall

within the uncertainty of the gas-derived age control points. We report comparisons of the age-scale relative to the automated as well as the manually-adjusted methane match-points.

The new version reads:

*Ice cores can be stratigraphically matched using records of trapped gasses, which reflect global changes in atmospheric composition. Centennial-scale variations in methane concentrations observed in the RICE gas records are also found in similar records from WAIS Divide (WAIS Divide Project Members 2015; Mitchell et al. 2011). Matching up these records allows a comparison of the two ice-core timescales.*

*The gas records from RICE and WAIS Divide were matched using a Monte Carlo technique reported in Lee et al. (2018). The feature matching routine employed discretely-measured records of methane as well as isotopic composition of molecular oxygen ($\delta^{18}O_{atm}$). Over recent millennia, however, the $\delta^{18}O_{atm}$ concentrations are stable, and hence provided minimal matching constraints. An average spacing of 26 years between successive RICE methane samples contributes to the matching uncertainty. The matching routine identified 18 match-points over the past 2700 years, i.e. an average spacing of 150 years. Subsequent visual comparison of the methane profiles suggested minor manual refinements of the match-points (8 years on average, maximum 23 years; all within the uncertainty of the automated matching). These adjustments resulted in a slightly improved fit.*

**Some parts are very long and not useful (most of section 3.2.3.2, part of section 3.3 on l. 10 or p. 15) – I suggest to reduce these sections and better concentrates on the method and associated uncertainties.**

As suggested by the reviewer, we have significantly shortened these sections.

**Accumulation rate is certainty an input of the firn model described in p. 13 while only forcing using a site temperature history is mentioned. It is very surprising that accumulation forcing is not mentioned here since one of the aim of this paper is to provide an accumulation scenario. We are thus expecting the use (or at least validation that everything is coherent with Dage or d15N measurements) of the accumulation rate scenario in the firn model.**

It is correct that the usual form of the Herron-Langway firn densification model takes both temperature and accumulation history as an input, thereby providing lock-in-depth and delta-age as output. However, in our dynamic H-L firn model, the equations have been re-organized to account for the knowledge on past lock-in depth based on measurements of d15N-N2, after correction for the existence of a convective surface zone. Consequently, the H-L model used here is forced by the temperature history (based on water isotopes) and past firn column thickness based on measurements of d15N-N2 (P13, L2-5). Additional parameters include the surface density (fitted to the modern density profile) and density at the lock-in-depth (here estimated from the temperature). With this formulation of the H-L model, we obtain as output: 1) the age at the lock-in depth (Δage), and 2) a low-frequency accumulation history.

The section has been revised as follows:

*Methane feature matching allows a transfer of WAIS Divide ages to the RICE gas records, i.e. the RICE gas ages. To obtain the corresponding ice-core ice ages relevant for this study, Δage was calculated using a dynamic Herron-Langway firn densification model (Herron & Langway 1980) following Buizert et al. (2015), as described in detail in Lee et al. (2018). The model is forced using a site temperature history derived from the RICE stable water isotopes, and the firn column thickness is constrained by the isotopic*

*composition of molecular nitrogen ($\delta^{15}N$ of $N_2$). In addition to $\Delta$age, this formulation of the Herron-Langway densification model produces as output a low-resolution accumulation rate history.*

In the new version of the manuscript, we have included the firn-model based accumulation history as well as the measured d15N-N2 values in figure 9c (now: figure 10e, dashed line and black dots):

[Figure]

*Figure 10: a) Measured methane concentrations from RICE (blue, on the RICE17 timescale) and from WAIS Divide (red, on the WD2014 timescale). b) Bryan, Antarctic Peninsula (grey) (Thomas et al. 2015), c) Talos Dome (TD96, orange) (Stenni et al. 2002), d) WAIS Divide (WDC, red) (Fudge et al. 2016), e) Siple Dome (SDM-94, pink) (Kaspari et al. 2004), and f) RICE (blue) accumulation histories over the past 2700 years, in annual resolution and 20-year smoothed versions (thick lines). WAIS Divide accumulation rates have been corrected for ice advection. The shaded blue area indicates the 95% confidence interval of the RICE accumulation rates. The short-lived peak in accumulation rates around 320 BCE is likely to partly be an artefact caused by timescale inaccuracies in this period, during which RICE17 diverges from WD2014 (Fig. 9b). Also shown are the gas-derived accumulation rates for this time interval (f, blue dashed line), and measurements of $\delta^{15}N$ of $N_2$ informing on past firn column thickness (f, black dots; on the RICE gas timescale). g) RICE stable water isotope record ($\delta D$). Thick green line is a 20-year smoothed version of the isotope profile. Thin grey horizontal lines denote mean values of accumulation rates and $\delta D$ over the displayed period. The migration period of the Roosevelt Island ice divide is marked with a grey box.*

We note some discrepancy between this accumulation history and the one derived from the RICE17 layer counts and thinning function, and we discuss these differences in the manuscript:

*The RICE17 accumulation history shows reasonable agreement with low-resolution accumulation rate output from the dynamic Herron-Langway firn densification model (Fig. 10f, dashed line). The gas-based accumulation rate history does not resolve high-frequency variations, but shows a slow increase in accumulation rates of 0.16cm/century, similar to that obtained from the ice core chronology prior to 1250 CE. In contrast to the accumulation rate history derived based on the layer thicknesses and thinning function, however, the firn-based accumulation rates continue to increase until present-day. Further, the absolute value of the inferred gas-based accumulation rates tend to generally underestimate the accumulation rates by ~4cm (16%).*

*We speculate that these discrepancies may have to do with the shift in RICE water isotope levels occurring around 1500 CE (Fig. 10g), which in the firn model is used to represent temperature change. It has been suggested that this shift is due to other factors than temperature (Bertler et al. 2018). Assuming that the model is based on a slightly too-cold temperature input prior to 1500 CE, the model would compensate by decreasing the accumulation rates during this time, in order to preserve a constant thickness of the firn column, as indicated by steady values of $\delta^{15}N-N_2$ (Fig. 10f, black dots).*

**What is the "model" mentioned in l. 45, p. 13?**

The "model" referred to here is a Lliboutry vertical ice-flow velocity profile fitted to observed vertical velocities. In the new version, this sentence has been rewritten as follows:

*We produce a thinning function appropriate for RICE using vertical velocity profiles obtained by fitting a simple ice-flow model (Lliboutry 1979) to englacial velocities deduced from radar measurements (Kingslake et al. 2014).*

**The discussion on the ASL influence on the accumulation rate in the region is both in the "Results" section and in the "Discussion". This is also the case for other ideas that are repeated several times and a reorganization and simplification of the manuscript is needed.**

We have carefully gone through the manuscript and reorganized, simplified and shortened the text. All text about the influence of ASL on RICE accumulation rates has been moved to the discussion section.

**The accumulation reconstruction should ideally have been compared to accumulation rate scenario used for the firn model as well as with water isotope profiles. It could strengthen the discussion and conclusion parts on the accumulation aspect that are rather short.**

In the new version of the manuscript, we compare the accumulation reconstruction from the firn modelling to the accumulation history from the RICE17 timescale, see comment above.

We have further expanded the discussion regarding the accumulation rate history, which is now evaluated in context of regional climate drivers, including regional sea ice extent.

---

## Author Comment (AC4) · 11 Jun 2018

**Author response to editor and reviewers**

We thank the reviewers for their comments to the paper, and have revised the paper accordingly.

Main revisions include the following:

- The paper has been thoroughly revised, restructured and written for brevity.

- The volcanic matching between RICE and WAIS Divide has been revised:

  The revision was based on a new conductivity-to-Ca excess depth profile, directly calculated from the two CFA records. Due to slight differences in depth assignment of the two records, we previously refrained from calculating their differences, and instead compared the two records visually. However, having a directly calculated record of the non-sea-salt conductivity simplified the volcanic matching between the two cores.

  The main difference from the previous matching is that we have removed some of the volcanic matches that we did not consider to be completely certain. Using the new non-sea-salt conductivity record, we were also able to identify and match up many more of the volcanic peaks in RICE to a counterpart in WAIS Divide.

- The new RICE volcanic record (Table 2) only includes volcanic events that are visible both in the RICE and WAIS Divide ice cores. We therefore do not attribute peaks in the RICE acidity records that do not exist in WAIS Divide as clear evidence of volcanic eruptions.

- We corrected an error in the calculation of uncertainty on the RICE accumulation rates. With this correction, the uncertainty bounds on the accumulation rates become much smaller going back in time, allowing us to infer past trends in accumulation rate with greater certainty.

- The connection between climate drivers and RICE accumulation rates have been described in more detail.

Point-by-point replies to the two reviewers' comments are provided separately.

With these changes, we hope that you will agree to the submission of a revised version of the paper.

On behalf of the authors,

Mai Winstrup

---

## Author Response (AR3)

**Answers to reviewer 1**

We thank the reviewer for the detailed comments, and apologize for the confusion in the description of the thinning function. We found that answers to some of the issues were present in previous drafts of these sections but had been cut in accordance with the desire for a shorter and more streamlined text from the first set of reviews, so we appreciate the opportunity to explain them more. We have changed the description in the main text and added a new section in the supplementary material. We chose to put much of the detail in the supplement because out of the four reviews, only this referee showed particular interest in the thinning function. Therefore, we elected to keep the main text streamlined without adding too many details on the ice-flow modeling.

See below for detailed responses.

The paper estimates a record of snow accumulation for the last 2,700 years at Roosevelt Island, Antarctica. The record shows a decrease in accumulation since the mid-1960s that Bertler et al (2018) relate to an increase in sea ice extend.

The authors extract the information from an ice core following two main steps:

- Obtaining a detailed chronology that combines annual-layer counting, that follows the seasonality traces of a few chemical constituents, and a few historical constrains, like well dated volcanic eruptions.

- Estimating the past-accumulation records from the thickness of annual layers. For this, the authors have to estimate the influence of ice-flow from the time of deposition on the surface to the present: compaction from snow to ice, and compression.

In general, the document is clear and well written. I find a strong difference in level of detail between the sections related to ice-core analysis and establishing a chronology, and the sections related to ice-flow and deriving an accumulation record. More of that later.

I am not an expert on ice-core analysis so I will only say that the description of the methods in those sections reads well and they seem to be referenced. I have however serious concerns related to the reconstruction of past accumulation rates. I will group my concerns in terms of clarity, methodology and uncertainty.

Clarity
Sections 5.2 and 5.3 are simply not clear. Technical terms are not explained or referenced. For example: Raymond stack, divide migration, divide/flank flow transitional type flow or thinning function.

Indeed, the reference to Raymond 1983 had gotten cut from this section, and we have added it back. We have rewritten sections 5.2 and 5.3, and also added a more detailed description of divide and flank flow, and reference Nereson and Waddington (2002), which provides much greater detail on how divide migration affects the internal layering in an ice sheet.

A few important steps are not described or referenced. For example:

-How do the authors extract the information about the divide migration from the radar stratigraphy.

We infer the divide migration from the location of peak amplitude of the Raymond arch. The peak is nearly beneath the modern divide in the shallowest layers, but by about 120m depth is about 500m east. We have revised and added to the paragraph describing this line of argument, and reference Nereson and Waddington (2012) on the interpretation of internal layering in terms of divide migration.

-How do they derive the thinning function from a vertical velocity that varies in depth and in time.

We have more fully described that we use a transient 1-D model which tracks layer thinning with an annual time step. We iterate to find consistent values of the time-varying accumulation history and the modeled thinning.

-How do they calculate the present surface thinning. The authors state that they use an ice-flow model to match the dated architecture of the Raymond stack. What ice-flow model? 1D, 2D? Using the velocity profile that they describe below and the time-varying vertical velocity implied by the time-varying accumulation?

A full response to this comment will require a manuscript of its own (which we are working on), but we consider this to be beyond the scope of a paper focused on the timescale. To answer, we have two lines of support for the current thinning rate.

The first is from the fitting of the pRES vertical velocity profiles. The vertical velocity at the surface is equal to the sum of the accumulation rate and the ice-sheet thinning rate. We know that the current average accumulation rate is approximately 24cm ice eq/yr. So when fitting the vertical velocity profile to the measurements, any difference between the surface velocity of the fit and the accumulation rate is the ice-sheet thickness change. Kingslake et al. (2014) obtains the best fit for the divide with a surface velocity of 27cm/yr, implying 3-5cm/yr of thinning. However, with this profile, the Lliboutry fit yields too low vertical velocities at mid-depths (40-70% of ice thickness) compared to the measurements. The fit improves with more negative values of the Lliboutry parameter p, but this also requires a higher vertical surface velocity, which becomes unrealistic (more on that below). Kingslake et al. (2014) noted that there is a region of near-linear vertical velocity change in the pRES measurements (Kingslake, Figure 5). We found that continuing the slope starting at approximately 80% of the ice thickness yielded a surface velocity of 26 cm/yr, in line with the current accumulation rate and modest ice-sheet thinning.

The second line of support is from ice-flow modeling. We have used both 1D and 2D models, with both constant and time-varying accumulation rate histories. For 1D, we matched the bump amplitude (as in Conway et al., 1999, Figure 3). For 2D, we have matched the layers directly as well as the bump amplitude. The overall conclusion is that the divide was established at about 3ka and that there is a small amount of thinning; a couple of cm. However, a unique solution is elusive. The lack of velocity measurements near the surface creates uncertainty in the development of the Raymond Arch (and even if there were measurements in the firn column, distinguishing the effect of ice-flow thinning from firn compaction would be tricky). During development of the Raymond Arch, different choices of the vertical strain profile near the surface can be balanced by varying the accumulation history, these forming a similar depth-amplitude relationship. The lack of a strong constraint on the surface vertical velocity is a particular challenge. In addition to impacting the inference of ice-sheet thickness change and the vertical strain profile, it also impacts the implementation of the vertical velocities in 2D. For instance, the variations in near-surface vertical velocity inferred by Kingslake et al. (2014) in Figure 5 could be due to either spatial variations in the ice dynamics or in the accumulation rate at the surface.

Methodology
It is difficult to assess the quality of the methods without fully understanding them. However I want to make a few points with my current understanding of the methods but, in any case, the authors should discuss these points further.

-Divide Migration. The authors make a strong assumption about a divide migration of 500m amplitude between 500yr and 250yr ago. It has a strong effect on accumulation records, particularly when combined with the arbitrary vertical velocity profile assumed. I believe this is the main difference between the accumulation records between the initial and the revised version of the manuscript. 'The correction'.

The 'correction' referred to by the reviewer was the correction of a simple miscalculation of the accumulation rate uncertainties in the original manuscript. In the previous manuscripts, the uncertainty on the thinning rate was given as a percentage of the total thinning. By mistake, instead of using it as a percentage, in the first version of the manuscript these uncertainties were used as absolute values. Correspondingly, with the correct calculation in the revised manuscript, the uncertainties on the past accumulation rates were significantly reduced. There was no change in the most likely thinning function or accumulation history between the two manuscripts.

The assumption on the divide migration history does not impact the inferred accumulation rate history that much, as will be discussed more below. In the new version of the manuscript, we have expanded and recalculated the uncertainties on the thinning function, and here include the possibility that there has been no migration of the divide over time. The obtained uncertainties are reasonably similar to those previously estimated.

The authors assume that this should be evident by looking at Figure 6c. I disagree. In this particular case, we have radar-derived englacial strain-rates (Kingslake et al, 2014; Fig. 6b). The stronger near surface strain-rates are indicating where the weak strain-rates near the bed are at the present. Curiously enough, that area coincide in Figure 6b with the location of the Raymond stack. This shows that a) no migration has occurred as present Raymond area is on top of the bottom-most Raymond bumps and the tilt in Raymond stack is consequence of other factors, or b) the divide has gone back and forth recently. In any case, it is not evident what the authors are assuming and I have bough a new argument to the table.

We believe a recent migration of the divide, and divide-flow, is the most likely scenario given the observed tilt in the Raymond arch (e.g. Nereson & Waddington, 2002). However, we agree that the offset in the Kingslake et al. (2014) region of inferred maximum near-surface strain rate from the present topographic divide introduces complexity into the interpretation - the physical topographic divide and the location of maximum divide flow are not necessarily the same. We note that Kingslake find the strongest near-surface strain rates ~200m east of the drill site, whereas the peak of the Raymond arch is located ~500m east. However, as described below there is significant uncertainty in the derivation of near-surface strain rates from the pRES measurements.

In developing our most likely scenario, we first assessed the uncertainty of the vertical velocity profiles and inferred near-surface strain rates from the pRES measurements (Kingslake et al., 2014). We noted that different radar polarizations resulted in large differences in inferred near-surface strain (Figure 6 in Kingslake et al., 2014), with one near-maximum value west of the summit. Given that there are few measurements in the upper 150m (none in the upper 90m) and the near-surface strain rate is an extrapolation from the upper half of the ice sheet, we could not exclude that the current topographic divide has a vertical velocity and near-surface strain rate very similar to the representative divide vertical velocity profile. Thus we considered the most likely scenario that the divide and divide-flow are co-located and the tilt in the peak of the Raymond Arch reflect the divide migration.

For the purpose of this manuscript, the larger point raised by the reviewer is not the specifics of divide migration, but the impact on the inferred thinning function and hence accumulation rate history. We have expanded the uncertainty on the accumulation rate following a more detailed analysis of possible scenarios, as discussed in the supplement. As part of the new uncertainty estimate, we have recalculated the thinning function using the assumption of stable divide-flow, with the maximum strain rates located slightly east of the ice core drill site. The bump amplitude at mid-depths beneath the core site today is approximately 70% of the peak amplitude and the inferred (by Kingslake et al., 2014) near-surface vertical strain is also approximately 70% of the maximum measured surface strain rate. If the center of the divide-flow has been consistently offset from the physical divide, the ice in the core would have experienced transitional flow (70% divide-flow, 30% flank-flow) for all of the past 3ka, including the past 500 years.

A comparison of the new uncertainties to our previous estimates is shown in the figure below. For some sections, the new uncertainty estimate is smaller than our previous estimate, where we in a simple (and conservative) way accounted for the uncertainties associated with the vertical velocity profile. The uncertainties in the deeper part of the core has become slightly larger compared to the previous estimate, but otherwise the two uncertainty estimates are fairly similar.

[Figure]

-Vertical velocity at the divide. The authors assume that the ice-core site is currently at the divide position and they assume that that corresponds with the divide-type velocity (p=-1.22 in this paper; p=-0.78 in Kingslake et al (2014)). However a close inspection to Figure 6b reveals that the 'divide flow' in Figure 4 of Kingslake et al (2014) and Figure 6a in this paper is taken from about 500m east in the Figure of the summit position (green marker in Kinslake et al 2014). This position corresponds currently with the bottom-most position of the Raymond stack and not the ice-core location. This raises an important concern as, in summary, they are assuming that for the last 250yr the strain-rates are distributed following a distribution that does not correspond with the current conditions at the ice core site.

The divide-type flow profile is taken approximately 200m, not 500m, east of the ice core location.

As previously mentioned, the strain rates in Kingslake et al (2014), Figure 6b, have considerable uncertainty and were obtained from the depth interval ~130m-400m. Thus, we considered the vertical velocity profile presented by Kingslake et al. (2014) in Figure 5 as representative for the divide flow velocity profile at Roosevelt Island, regardless that it is not co-located with the current ice-divide. As discussed above, we have also presented scenarios that do not assume a recent transition to full-divide flow for the ice core.

-Vertical velocity at the flank. The authors use an arbitrary combination of two measured vertical velocities. From a combination of 0.7 this and 0.3 of that. Why? There is no explanation and it has a strong impact on the results before 250yrs ago.

For flank-flow, we use the flank-flow velocity profile found by Kingslake et al (2014), figure 4. For flow that is a mix between flank flow and pure divide flow, we use a linear combination of the velocity profiles appropriate for the two flow regimes, following Nereson and Waddington (2002). This allows a smooth transition as the flow regime transitions from partial to full divide-like flow.

The weighting of the two velocity profiles (70% divide flow, 30% flank flow) during the early period, is found from the relative amplitude of the Raymond Arch at the site: The ice core intersects the Raymond Arch at mid-depths at about 70% of the maximum arch amplitude, which we use as an indicator for the importance of divide-like flow. This is now better explained in the manuscript.

-Combination of flank/divide velocities. I have argued that I don't think there is a migration. However, even in that case, why do the authors use the wrong shape functions? There are, according to Kingslake et al (2014), 35 englacial profiles at Roosevelt Island, one of them is really close to the ice-core site. Wouldn't it make sense to use the englacial velocity from the site that is closest to the ice-core site?

First of all, there is considerable noise in individual englacial pRES velocity profiles. Secondly, what we want to obtain from the data in Kingslake et al. are typical "flank-flow" and "divide-flow" velocity profiles. We therefore choose to use the divide and flank profile that were presented in detail in Kingslake et al. (2014), both because these are the cleanest profiles and because they are the most accessible being shown explicitly in their Figure 4.

-Vertical velocity to thinning function. I mentioned earlier that the authors do not explain how do they calculate the thinning function. I just want to highlight here that this is not a trivial point as thinning function includes the cumulative effect of vertical compression from the surface that, in turn, depends on the time-varying accumulation that is unknown (e.g., Parrenin et al (2007) suggest an iterative method). The authors however assume a vertical velocity related to present values of surface accumulation and a varying shape function.

This is now more fully explained in the text. We use a 1-D model with an annual time-step with one iteration. In the first model run, we use a constant accumulation rate. We use the inferred thinning function to calculate the time-varying accumulation rate and then rerun the model. In this second iteration, at each timestep, the vertical velocity profile is scaled by the annual accumulation rate (plus the thickness change rate). We found that additional iterations yielding insignificant changes. While the surface vertical velocity likely does not vary at annual timescales, the resulting thinning functions are smooth, and no significant differences came from using a smoothed accumulation rate history.

Uncertainty.
It is difficult for me to predict the cumulative effect of all the factors I have just mention but, independently of them, the propagation of uncertainty has to be improved.

- I see no problem with the paper showing a favourite hypothesis, a divide migration, but the uncertainty should cover other other hypothesis: from no migration to a more recent migration. All this should be covered by repeating the estimation of accumulation with flank and divide velocities and showing them as two extreme cases within a range of scenarios.

- Also, I reiterate that I don't know how the authors estimate the thinning function. However, they should include in the uncertainty the effect of assuming constant the present accumulation rate and thinning rate in the derivation of the time-dependent accumulation. The authors currently estimate sensitivity by comparing the effect of assuming (0.24+0.02) m/yr with 0.24 m/yr. My view is that this is a clear underestimation of uncertainty. It should include, at least, the variability of the derived accumulation records through the time of interest, around 0.2m/yr to 0.28m/yr, to capture the uncertainty induced by this assumption.

As recommended by the reviewer, we have expanded our discussion of the uncertainty, and now calculate the uncertainty from an ensemble of different scenarios, including changes in the divide migration history over time, the extreme being no migration at all. As shown, the resulting uncertainties on the accumulation history are, however, relatively similar to our previous estimate.

The thinning function is calculated with a time-varying accumulation history which scales the vertical velocity profile. This is now explained better in the text, and we think we have satisfied this recommendation.

**Answers to reviewer 2**

We thank the reviewer for the thorough comments, and have revised and improved the manuscript accordingly. We have e.g. added a figure showing the annual layer counts for different depth sections, and expanded our discussion on the volcanic matching process in the manuscript. The paper now also includes two additional figures showing the volcanic matching in other depth sections (in the supplementary), and we have added to table 2 a measure for the statistical significance of peak size of the individual volcanic match-points in RICE.

See below for detailed responses.

Review of revised manuscript "A 2700-year annual timescale and accumulation history for an ice core from Roosevelt Island, West Antarctica" by Mai Winstrup et al.

In their revised manuscript the authors made a serious effort to address the issues raised in the first round of reviews. The manuscript has been restructured for clarity and several potential sources of misunderstanding have been eliminated. In particular the independence of the RICE17 age scale w.r.t the WAIS Divide ice core chronology is now made clear. Likewise the manuscript benefits from several related publications now being published or accessible in their discussion stage. This especially concerns the companion paper of Lee et al.

However, there are still two fundamental issues to be fully resolved before the manuscript can be considered for final publication in CP.

1) The automated annual layer counting by the StratiCounter algorithm is the backbone of the RICE17 chronology, but the authors provide no evidence of its performance in identifying annual layers, especially for the deeper and more thinned core sections. At present, the manuscript only contains one figure (Fig. 4) showing the manual assignment of annual layers in a very shallow section of the core, which is not enough. A separate figure would be needed showing exemplarily the automated layer identification. Of special interest here would be the depth section that has been manually counted to initialize the algorithm, as well as part of the deep sections with thinned layers.

As an added benefit, such a figure would also provide evidence of how clear an annual signal can be counted in the various proxies - here referring to the discussion in the first round of reviews about using BC vs. H+ for counting.

Although the main text would be adequate, I suppose the figure could also be placed in the supplement part S2 if the authors would not like to further increase the volume of the main manuscript.

We agree with the reviewer that the automated layer counting and the CFA data, on which it is based, ought to be illustrated in the manuscript. We have followed the reviewer's suggestion to add a figure showing the layer-counts in three different depth sections of the core, including a deep section and a section with manual layer counts (these also being shown). We eventually decided to place the figure in the main text, and now refer to it when discussing the clarity of annual signal in the various proxies.

The performance of StratiCounter has been thoroughly documented in several previous papers, e.g. Winstrup et al, 2012, Winstrup 2016. It has here been shown that the algorithm is able to produce an unbiased counting even in very noisy data series, for which it is hard to manually count layers in an unbiased fashion. We note that we initiated the algorithm on basis of a set of layer counts, which was known to have a bias, since according to this initial timescale, the age of the Pleiades tephra was significantly (25 years) younger from that in the WAIS Divide core. However, all StratiCounter timescales placed the age of this tephra layer within a narrow age interval (±10 years) around the WAIS Divide age. We took this as a sign that StratiCounter performed well in producing a layer counted timescale for RICE. This was further supported by the volcanic matching to WAIS Divide, described in the next comment.

2. The volcanic identification and matching against WAIS remains elusive to me and needs to be clarified further. The detection of a presumably volcanic signal in the various proxies seems to rest on manually selecting even small peaks in the signals. The manual approach appears somewhat arbitrary by comparison to the automated annual layer counting procedure. This is especially so since the authors are aware of the difficulties associated with the high background of non-volcanic sulphur/sulfate and acidity and state to have removed a number of peaks previously considered as volcanic markers. At present, it appears questionable and unclear from the manuscript how reliably volcanic signals are in fact recorded at RICE. Hence, selecting volcanic signals would benefit from a more quantitative detection technique, or at least from employing clearly stated criteria.

First, we note that the majority of peaks (>80%) that were removed from the original table were peaks in the RICE acidity/sulfur records, which were not related to any counterpart in the WAIS Divide core. These were removed from table 2, as they were of no use for evaluation the timescale, and as they may possibly originate from an oceanic rather than volcanic emission source. As we do not include volcanic peaks that – for one reason or another – could not be unambiguously matched to a corresponding peak in WAIS Divide, many peaks of likely volcanic origin in the RICE records are not included in table 2. This has been clarified in the manuscript.

The volcanic matching was indeed difficult for RICE, as also stated in the paper. The matching relied significantly on having accurate layer-counted timescales for both cores. Comparing the ice core records on their respective timescales, we were able to find in both cores a similar series of acidity peaks spaced with similar age differences. In other words, it is predominantly the pattern of acidity peaks that were matched between the two cores, rather than the individual peaks themselves. A similar approach was previously used to synchronize ice cores from Greenland and Antarctica by annual layer counting and linking of bipolar volcanic reference horizons (Svensson et al., 2013; now referenced in the text). In the manuscript, we have revised and extended the description of this approach, and for better illustration of the method, we have added to figure 9 (now figure 10) the age differences between the volcanic match-points in the two cores.

As the reviewer points out, a quantitative approach to identifying and matching volcanoes between the two cores would have been ideal, but a simple "over-the-threshold" technique would have identified many peaks of non-volcanic origin. Furthermore, no simple threshold method is able to account for e.g. changes in baseline and variability of the signal with depth as well as it is possible to do by eye. Even more importantly, a quantitative technique for selecting volcanic marker horizons to be matched to WAIS Divide should, as previously mentioned, not simply be based on the individual peaks, but combining this with some sort of pattern matching technique that could account for their relative age spacing, and the associated age uncertainties. To develop such a rigorous method is a significant undertaking, which we consider far beyond the scope of this work.

Instead, we have added to table 2 a measure for the average peak prominence in the RICE records corresponding to each of the volcanic match points. To account for changes in baseline and variability of the signals with depth (due to varying measurement quality as well as changes in signal with depth), smoothed versions of the four volcanic proxy records (acidity, nss-cond, nssS, ECM) were standardized using z-scores within a running 50-year window. The records were smoothed over a depth section corresponding to 1/4 year to remove noise while amplifying the volcanic signal. The average value of the peak prominence in the four volcanic proxy records is provided in Table 2. However, we observed that in some cases, the mean value provided in table 2 was not the best indicator for the peak prominence. In some cases, the volcanic peaks were clearly observed in some proxy records, but not others, which significantly decreased the associated average significance level.

The majority of the match points are associated with peaks exceeding two standard deviations of the overall variability of the record. However, while the majority of the volcanic marker horizons are prominently observed in the RICE records, also some smaller peaks were deemed reliable match points, where these formed part of a matching sequence of volcanic events in the two cores.

The development of the non-sea-salt conductivity (nss-cond) may provide remedy in this context, but is not convincingly shown to record volcanic signals. For instance, Figure 9 shows three distinct peaks in nss-cond (around 162, 164 and 168 m RICE depth). The other match points, however, have little or no sign of a peak, including the Pleiades tephra horizon. There is an additional peak around 169 m not considered. Moreover, judging from Figure 9 the signals of H+ and ECM appear to perform equal or even better (less noise) in indicating peaks at the presumed volcanic markers - which would argue against the need to use nss-cond. Based on Figure 9, I am not convinced that the acidity signal provides more information than the nss-S signal, other than its higher depth resolution.

While the non-sea-salt conductivity record would not necessarily be sufficient for identification of volcanic marker horizons on its own, the record adds valuable information to the other data series. In order to obtain the best volcanic matching of the RICE record, we used the information in all four records: ECM, nssS (available down to 249m), H+, and non-sea-salt conductivity. This has been clarified in the manuscript. Not only do the various records contain complementary information, the existence of multiple data series also helps to identify variability caused by measurement errors in a single data series, as well as to retrieve volcanic match points in sections where other data series are missing (we note in table 2 the peaks that are not based on all 4 data series due to missing data).

With the high depth resolution of the non-sea-salt conductivity and $H^+$-records, the combination of these two were particularly useful for distinguishing between measurement noise and acidic signal. However, the usefulness of the individual records varied with depth, as well as it depended on the individual volcanic horizon. For the deeper part, the nssS-record became significantly less useful as the much lower depth resolution became critical. During our revisions, we realized that the ECM record shown in the figure in the previous draft was not the ECM record, but a smoothed version of H+. This has now been corrected. The ECM record is generally very noisy, but some sections are better than others. This was primarily due to variability in the ice core quality, but variation between ECM operators was also a factor.

Regarding figure 10 (previously fig. 9):

For the Pleiades horizon: The identification of this match point was not based on chemical signals in the core, but was found by visual identification of a dusty layer in the core, later confirmed to be tephra. Indeed, the horizon cannot be observed in the CFA data for a very good reason: The section with tephra was removed before the CFA analysis, meaning that we do not have chemistry data for exactly this depth interval. We have made sure that this is now clear from the figure, which no-longer shows interpolated values across the missing data interval.

As the reviewer points out, two of the volcanic peaks are less prominent in the nss-conductivity record. We have changed the color of the two less prominent marker horizons mentioned by the reviewer (161m, 166.7m) to illustrate that in terms of peak height, these peaks are among the less significant peaks of all identified volcanic match points in the core, and they do not exceed a 2σ threshold for peak significance.

Our reasoning behind nevertheless identifying them as volcanic horizons (and match points to WAIS Divide) is as follows:

At 161m, we observe a broad peak in the nssS-record, which we identify as very likely to be a volcanic horizon: At this depth, the layer thickness (~17cm) is sufficiently large that ~3 IC measurements of S exist per year. As the biologic S-component is a summer signal, an extreme biological influx event should not express itself as a series of successive data points with elevated sulfur values, also through winter, as it is present here. In the remaining records, there is a (smaller) peak in the H+ and nss-cond records, with e.g. the peak in the normalized nss-conductivity record reaching 1.8σ. Since further this peak is located at exactly the right spot according to the respective timescales and the surrounding match points, we believe that we can safely assume this to be a reliable match point.

Also at 166.7m, nss-cond is the record out of the four, which, after smoothing and normalization, has the largest peak (1.5σ). However, we would not have identified this as a reliable match point, had it not been surrounded by two very clear match points, and being located exactly at the right spot between them.

The above illustrates the importance of looking for patterns of volcanic peaks to obtain reliable match points, as well as the ability of non-sea-salt conductivity to record volcanic marker horizons, even if the peaks do not always exceed a 2σ threshold.

The "peak" around 169m mentioned by the reviewer is very likely an artifact from the calculation of the non-sea-salt conductivity, and caused by a slight discrepancy in the depth alignment of the calcium and conductivity records, this causing a "heart-beat" pattern: a very thin peak immediately followed by very low (here: negative) values. There were several similarly shaped "heartbeat" patterns in the same figure (e.g. at 161.5m, 164.5m, 167m) and even more of smaller size. We emphasize that only broad peaks in the nss-cond should be considered possible volcanic horizons. In the revised version of the figure, we now show a more smoothed version of the nss-conductivity record. The smoothing removes these heart-beat shaped patterns, and hopefully makes the broader peaks more apparent to the reader. As in the previous version of the figure, we also have colored the area under the graph in order to focus the attention of the reader to the broader peaks.

The authors are aware of potential pitfalls associated with nss-cond calculated as a secondary quantity, but do not quantify how much (or how little) the signal is affected by these problems. For example, this concerns potential false peaks produced by uncertainty in depth-alignment between Cond and Ca++, as well as peaks through very low Ca++.

As mentioned in the previous comment, an incorrect depth alignment of the conductivity and $Ca^{2+}$ records gave rise to a very particular heart-beat pattern, which easily could be recognized. We further used several approaches to account for the various issues associated with the non-sea-salt conductivity being a secondary quantity: 1) We focused on the broader peaks as these are not affected by depth-scale issues, 2) We used smoothed versions of the conductivity and $Ca^{2+}$ records before calculating the nss-conductivity, since this reduces the impact of depth-scale differences, and, finally but most importantly, 3) we always double-checked against a direct plot of the two records overlaying each other to identify whether peaks in non-sea-salt conductivity were caused by obvious errors in alignment or measurement issues. The conductivity-to-calcium excess is now also shown as a green area on the figures.

Indeed, often peaks in the conductivity-to-calcium excess could often much easier be recognized when comparing the two records directly. This is perhaps best illustrated in Figure S4, second panel from the top, where e.g. the Krakatau volcanic horizon is visible as a distinct peak in conductivity (grey) that is not present in the $Ca^{2+}$ record (green). This peak cannot be explained by e.g. depth alignment issues or different smoothing in the two records, which may account for many of the other smaller peaks in the derived non-sea-salt conductivity record.

Consequently, I believe the volcanic identification and matching needs to be expanded and clarified, ideally including: i) a quantitative approach to peak detection, e.g. by employing a local "peak-over-threshold" criterion, ii) a more convincing demonstration of the new volcanic proxy nss-cond, including addressing the pitfalls of a secondary quantity, and iii) showing a more extended version of Figure 9 as a side-by-side comparison between RICE and WAIS volcanic signals, also at depths not including the distinct tephra marker of Pleiades.

Replies to i) and ii) were provided above. Regarding iii) we now have added an extra figure to the supplementary showing the volcanic matching of the two cores for a deeper section. For this section, the marker horizons are significantly more sparsely located, making it harder to see the details of the various records. The nssS-record for RICE does not exist below 249m, and is therefore not shown.

Detailed comments:

Introduction: The companionship to the Lee et al. paper should be mentioned explicitly when stating the scope of this work.

This has been included.

P6 L37-38: " Some uncertain layers were counted as a year in the timescale, while others were not." This procedure remains unclear. As I am sure the authors are aware, a typical approach here is to count uncertain layers as 0.5+/-0.5 years. Why was this procedure not adopted in this case? It is also not clear what uncertainty estimate was obtained and why this should be a 95% confidence interval.

Always increasing the timescale uncertainty by 1/2 year when encountering uncertain layers would give rise to an always-increasing uncertainty estimate with depth, which is not appropriate for timescales constrained by known-age marker horizons (as those present in the upper part of RICE).

Instead, our approach for estimating the uncertainty on the manually-layer counted part of the timescale was as follows: We identified the most likely set of layers, while any uncertainties in the layer identification (layers that possibly could be added or removed from the most likely set of annual layers, while still adhering to the age constraints) were reflected as an increase in the associated age uncertainty. The resulting age uncertainty is largest midways between two age constraints. We have revised the description of the process in the paper. We believe this to be a very conservative estimate of the uncertainty, roughly corresponding to a 95% confidence interval. We chose this approach as it produces an uncertainty estimate most similar to that obtained from StratiCounter (which is a 95% confidence interval), and we desired to keep the definition of the counting uncertainty along the core as similar as possible.

P8 L5-6: "The calcium and conductivity records frequently displayed multiple peaks per year, limiting their contribution to the annual layer interpretations."

This is another reason why it would be very important to see the performance of StratiCounter in such a case.

Sections of the calcium and conductivity records have been included in the new figure 6, along with the inferred annual layers from StratiCounter.

P8 L20-23: "Using ... the modelled density profile fits well the observed values". How were the parameters used here obtained, i.e. why these values selected? Also, from Figure S2 it can be seen that this statement is true only for the top 50 m. Below that depth there is a substantial misfit in the lower core - including depth of the firn-ice transition. This should at least be mentioned - I assume consequences of the misfit would be negligible if only an extrapolation of near-surface values was concerned.

The parameters used are observed values for accumulation rate, surface temperature, and initial snow density. This has now been clarified in the manuscript. Indeed, there is a small misfit between the model and the observations below 50m, but the top part fits well. It is unclear why there is this difference between modelled and observed densities, but as mentioned briefly in the supplementary (S3), it could potentially be due to the additional vertical strain present at divide locations.

Because of the misfit between model and observations, we have used the observed density values to correct the layer thickness for density changes with depth. Since we use the model only for correcting for the density in the very top part of the core, the misfit between modelled and observed densities in the deeper part should have no impact on our results.

P9 L15: "estimated using an ice-flow model to match the dated architecture of the Raymond stack" This is rather vague. What ice-flow model was used? Is this work done by the authors or the approach of Kingslake et al. (2014)? Please clarify.

This ambiguity in the text was also pointed out by reviewer #2, and we have provided a detailed answer there.

In short, we have used both 1D and 2D models, matching respectively the bump amplitude and the internal layering directly. This work will be presented in a paper by itself (which we are working on). For the present paper, we have added a comment that this is on-going work (see supplementary).

P9 L19-20: "a linear combination with divide-type velocities weighted by 0.7 and flank-type velocities weighted by 0.3."

Please state why these values were used. Was this an arbitrary selection? If so, would uncertainty in these values contribute to the uncertainty in thinning function? This approach remains unclear.

As requested by reviewer #2, we have rewritten the section on the ice-flow modelling to clarify our approach. We use a linear combination of flank and divide-like flow profiles to account for the migration of the ice divide, which causes the flow to become increasingly divide-like over time. The specific weights come from the relative magnitude of the Raymond arch at the ice core site: The amplitude of the Raymond Arch at the core site is approximately 70% of the peak amplitude, and we use this number to indicate the relative importance of the divide-like flow profile in the deeper part of the core.

Following the suggestions from reviewer 1, we have recalculated the uncertainty in the thinning function. It is now calculated based on the envelope of thinning functions obtained from 12 different scenarios, which includes the uncertainty in divide-migration over time, and how it has impacted the vertical velocity profile. The resulting uncertainties are in reasonable agreement with our previous estimates.

P9 L29-30: "Except for the uppermost part of the record, uncertainty in the thinning function dominates the total uncertainty, and only this factor will be considered here."

This is an important statement but is not supported by any evidence. Please explain.

Additional uncertainties could be caused by the density profile or misinterpretation of annual layers.

With regard to density: Except for the top 8m, we use measured densities, and the uncertainty associated with these should be minor. The extrapolation of densities to the surface is aided by the measurements of surface density in nearby snow pits. This, and the good agreement between modelled and measured densities in the top 50m, ensures that the density correction only causes minor uncertainty for the estimation of past accumulation rates.

Misinterpretation in annual layers may give rise to the annual accumulation for some years being significantly too high (e.g. twice as high if a layer is missing) or too low (e.g. half the actual amount, if there is a layer too many). However, due to the negligible bias of the timescale (as backed up by the volcanic matching to WAIS Divide), the uncertainty on average accumulation rates will be very small throughout the core. This argumentation has been added to the manuscript.

In the top part, the uncertainty in the thinning function is zero (no stain thinning has taken place), and hence the other two factors, while small, will dominate here. However, for the deeper part, it is a very good assumption to neglect the uncertainties of other factors than the thinning rate.

P9 L31 ff: This paragraph has some vague statements making it hard to fully comprehend. E.g. Was the near-surface uncertainty in density (it is only extrapolated) considered here? And: "(b) variation of the vertical velocity profile over time in ways not accounted for." E.g. I assume you are referring here to your change in weights of the linear combination?

The section only concerns the uncertainty associated with the thinning function, not the density correction. This section has been rewritten, and a section has been added to the supplementary, discussing the derivation of new estimates of the uncertainty on the accumulation rates.

P10 L3-4: "We suggest to interpret this uncertainty as a 95% (2σ) confidence interval." It is not clear why this needs to be exactly a 95% confidence level, or why this confinement to 95% is even needed.

We believe that our uncertainty bound is rather conservative, as it is calculated as the envelope of 12 scenarios, each of which represent (at least one) extreme parameter or scenario. As we wish to convey to the reader that this is a conservative estimate of the uncertainty, we have elected to keep the above sentence that we interpret the uncertainty as a 95% confidence interval.

P10 L13-14: "The CFA acidity record is driven primarily by the influx of non-sea-salt sulfur-containing compounds, as evident by its high resemblance to the IC non-sea-salt sulfate and ICP-MS non-sea-salt sulfur records in the top part of the core (Fig. 4)."

This may be true, but how can it be reconciled with the later statements about using acidity as a more reliable tracer of volcanic events, e.g. the statements about halide acids from regional volcanism? See comment for P19 L23 below.

Comparing the CFA acidity record with the non-sea-salt sulfur record, we observe more or less the same overall signal in the two records. However, at times there are peaks of excess acidity in the acidity record. We consider it likely that some of these peaks are caused by halogenic acids originating from regional volcanic eruptions. The two statements are therefore not contradictory. See also our reply to the last question, where we have provided some examples of distinct acidity peaks that are less prominent in the sulfur-record, although the remaining part of the records show fairly similar variability.

We note that we used all the available records for identifying volcanic eruptions in the RICE records. We do not as such consider the acidity record the most reliable tracer of volcanic events, but it was among the most useful records due to its high depth resolution.

P10 L28 and L33: Should be now Figure 9, not 8.

Corrected.

P13 L24: Please give a definition of how the conductivity-to-calcium excess was calculated.

The conductivity-to-calcium excess was calculated as follows:

nss-cond = cond – (a*Ca+b), with the coefficients a and b found from a linear fit for a running 2m window around the specific depth section. This definition has been added to the manuscript. The resulting record was subsequently smoothed in order to eliminate false peaks caused by slight differences in depth alignment between the two records.

P13 L27-28: "Nevertheless, peaks in the conductivity-to-calcium excess showed high consistency with peaks in total acidity, and it proved to be a reliable tracer for volcanic activity." Referring to main comment 2., this needs to be shown in more convincing detail. For instance, how does the excess signal look for the recent historical eruptions used in the paper, such as Krakatau?

We have added a figure to the supplementary showing the volcanic records (and the matching to WAIS Divide) in the top part of the record, including the Krakatau and Makian volcanic horizons. Both display a very distinct excess of conductivity relative to the calcium, which is especially visible when comparing the two records directly. The same peaks are visible in the acidity record.

P13 L36: The Kalteyer, 2015 reference is not accessible without a login from the University of Maine.

The reference has been removed.

P14 L44: "Fig 9a" should be Figure 10.

Corrected

P15 L40: "It has been suggested that this shift is due to other factors than temperature (Bertler et al., 2018),..."
Vague statement, please clarify what "other factors" are.

In Bertler et al, it is suggested that this change in isotope level may be related to changes in regional sea ice extent and atmospheric circulation. This has been added.

P16 L35-36: "Present accumulation rates show a distinct decrease on the downwind (western) side of the ice divide with a gradient of ~5 10-3 m w.e./km yr-1, although muted around the summit area." Where does this data come from? Is there a missing citation?

Citation has been added.

P19, L23 ff. "The halide acids are highly soluble, and will be removed from the atmosphere relatively quickly during transport. Hence, they will contribute to increased ice acidity in ice cores located close to the eruption site, whereas only sulfate is deposited from distant volcanic eruptions. By focusing on acidity as volcanic tracer instead of sulfur, the RICE volcanic proxies may thus be more sensitive to regional volcanism than to larger far-field eruptions."

As pointed out above, if acidity is dominated by non-sea-salt sulfur as claimed earlier in the manuscript, how would additional, presumably volcanic acids, leave a detectable imprint? In other words, if the H+

data shown in Figure 9 would be down-sampled to the depth-resolution of nss-S shown in the same Figure, would there be in fact a discernible difference between the two datasets, especially at the locations of the assumed eruptions? The reader is unable to evaluate this based on the presented data in Figure 9.

We have rewritten the section so that it becomes clear that while this is suggested from our preliminary investigations, we do not attempt to quantify the importance of the effect on the RICE records.

When comparing the acidity record, down-sampled to the same depth resolution as nssS, we observe a tendency towards a larger relative size of peaks in acidity for volcanic horizons that were not classified as originating from far-field eruptions. However, any further quantification is tricky due to: 1) changes in baseline with depth due to calibration issues, 2) the low resolution of the down-sampled records, and 3) a limited number of non-bipolar volcanic horizons for which both data sets are available.

Below is shown the nssS record and the acidity record down-sampled to the same depth resolution for two depth sections. We selected these two sections, as they both contained a volcanic horizon that could be matched to a non-bipolar volcanic horizon in the WAIS Divide ice core, i.e. the horizon was likely to represent a regional eruption. We would therefore expect the acidity peak for these eruptions to be relatively larger than the peak in nssS.

This is also what we observe: Comparing the two peaks at 164m (Samalas; bipolar) and 162m (not bipolar), the two peaks are of the same size in the acidity record, whereas the Samalas bipolar horizon is significantly larger in sulfate than the horizon at 162m. The second depth section contains only a single volcanic match-point horizon, namely at 180m. Compared to the overall range of variability in nssS and H+, there is also here an indication that the volcanic horizon has additional acidity compared to its sulfur content. We also observe that in general, the nssS and acidity show fairly similar (though not identical) variability throughout both sections.

[revised manuscript text omitted]

**Supplementary material**

**S1. Effective depth resolution of the CFA records**

Effective depth resolution of the CFA records wasere evaluated based on the time it took for the various measurement lines in the system to respond to an abrupt change in concentration level. Following the approach in Bigler et al. (2011), response times were calculated as the average time required for the system to transition from a blank water standard to the highest calibration standard plateau, using the 10% and 90% levels of the transition curve. From the employed melt rate of 3 cm/min, response times were then converted to equivalent response depths as a measure for the effective depth resolution. For the RICE CFA set-up, the conductivity record has the highest effective resolution of 0.8 cm, closely followed by black carbon (Table S1).

| | Response time (s) | Response depth (cm) | Missing data fraction |
|---|---|---|---|
| **Conductivity** | $15 \pm 2$ | $0.8 \pm 0.1$ | 6% |
| **Acidity** | $38 \pm 3$ | $1.9 \pm 0.2$ | 17% |
| **Calcium** | $49 \pm 6$ | $2.5 \pm 0.3$ | 9% |
| **Black carbon** | $20 \pm 3$ | $1.0 \pm 0.2$ | <1% |
| **Insoluble dust particles** | $17 \pm 3$ | $1.0 \pm 0.2$ | 8% |

**Table S1:** Response times for the CFA system to transition between the blank water level and a calibration standard plateau, and the equivalent response depths. Core breaks, contamination, measurement errors etc. gave rise to sections of missing data, these comprising from 1% to 17% of the total length of the record. Below 129 m, the dust record was extensively contaminated by drill liquid, and the missing data fraction is calculated for the uncontaminated top part only.

**S2. StratiCounter settings and procedure**

StratiCounter was initialized based on a preliminary set of manual layer annotations (Fig. 6a) within a selected depth interval (40-150 m). The manual annotations were used to produce a set of generalized templates for an annual layer in the various impurity records. We note that the manually-counted timescale was observed to have a bias towards counting too few layers, resulting in an age for the Pleiades tephra that is a few decades younger than observed in WAIS Divide. Applying the Expectation-Maximization algorithm (e.g. Gupta and Chen 2010), StratiCounter continuously updates and refines the statistical description of an annual layer, thereby allowing for changes in layer characteristics with depth (Winstrup, 2016; Winstrup et al., 2012). To further increase the independence of the StratiCounter timescale from the preliminary manual interpretation, in a final step the entire timescale was reevaluated using an improved set of layer templates derived from the algorithm output.

Rapid thinning of layers with depth in the RICE core necessitated slight changes in StratiCounter settings with depth. We therefore divided the record into four sections: an upper (42-180 m), an upper middle (165-250 m), a lower middle (240-300 m), and a lower section
(280-344 m). Overlap sections served as base for comparison between the runs, which were
found to contain only minor differences. Within these sections, the results from the deeper
section were used to produce the final timescale.

For the uppermost section (42-180 m), performance of the algorithm was tested using a variety
of algorithm settings, which all resulted in very similar timescales (±10 years at 165 m). The
final version was chosen as the timescale in best agreement with the WD2014 age of the
Pleiades tephra horizon (Dunbar et al., 2010) found at 165 m depth (Kalteyer, 2015) (Wheatley
and Kurbatov, 2017). Proceeding to the deeper sections, the algorithm settings were kept as
similar as possible to those employed for the upper part (Table S2).

The main change in settings with depth was the averaging distance employed to produce a
lower-resolution record from the original 1-mm resolution CFA records, performed before
automatic layer identification. Section delimitations were selected based on estimated layer
thicknesses obtained from methane matching to the WAIS Divide ice core (Lee et al., 2018),
and chosen so that an average layer consistsed of approximately 10-15 individual data points.
Accordingly, the averaging distance was successively reduced from 1.5 cm to 0.5 cm to account
for the general decrease in layer thicknesses with depth (Table S2). Note that the averaging
distance applied for the deepest section is less than the effective resolution of even the highest-
resolution impurity records (Table S1), meaning that successive averaged data points are
significantly correlated.

StratiCounter was run based on the full suite of CFA records: Black carbon, acidity, calcium,
conductivity, and dust (topmost section only). The dust record was excluded for the lower
sections due to drill liquid contamination. The impurity records were weighted so that records
of large similarity (e.g. the calcium and conductivity) were not treated as independent data
series, and with added emphasis on the black carbon, which displayed the most pronounced
annual signal (Table S2). Before analysis, extreme peaks caused by measurement noise and
processing errors were removed from the data series. These were further standardized using z-
scores based on the logarithm of the impurity concentrations in order to reduce inter-annual
variability in layer signal.

| Section | 42.34–180 m | 165-250 m | 240-300 m | 280-350 m |
|---|---|---|---|---|
| | (42.34-165 m) | (165-240 m) | (240-280 m) | (280-343.7 m) |
| **Depth resolution (cm)** | 1.5 | 1.0 | 0.75 | 0.5 |
| **Weights of impurity series** | | | | |
| **Black carbon** | 1 | 1 | 1 | 1 |
| **Acidity** | 0.5 | 0.5 | 0.5 | 0.5 |
| **Dust** | 0.5 | 0 | 0 | 0 |
| **Calcium** | 0.25 | 0.25 | 0.25 | 0.25 |
| **Conductivity** | 0.25 | 0.25 | 0.25 | 0.25 |

**Table S2:** StratiCounter settings for each depth range; (in parenthesis is given the interval for
which the results were used to produce the final combined timescale): Interpolated depth resolution of the impurity records before automated layer identification, and weighting of the
various impurity records in the StratiCounter algorithm.

**S3. The RICE density profile**

Measured densities for the RICE core (Fig. S1) show good agreement with a modelled density
profile calculated from a steady-state Herron-Langway densification model (Herron and
Langway, 1980) when using appropriate values for the initial snow density (410 kg m$^{-3}$; within
the range measured in adjacent snow pits), surface temperature (-23.5°C; consistent with
borehole temperatures (Bertler et al., 2018)), and accumulation rate (0.22 m w.e yr$^{-1}$). The
measured density profile was extended to the surface using the modelled densities. We note
that the initial densification in the Herron-Langway model is parameterized as a linear function
of depth, depending only on initial snow density and surface temperature.

At intermediate depths (50-120 m), the observed density profile has slightly denser snow than
predicted by the Herron-Langway model, and, as a result, the firn-ice transition is reached at
significantly shallower depths than predicted by the model. This difference may indicate that a
steady-state assumption is invalid, or it may be due to the additional vertical strain present at
divide locations (Kingslake et al., 2014).

[Figure]

**Figure S1:** Measured and modelled density profile for the RICE core.

**S4: Modeling of the thinning function for reconstructing the accumulation rate history**

**Model setup**
To calculate the thinning function, we employ a one-dimensional ice-flow model run at an
annual time step, which tracks the cumulative thinning of an ice layer. At each time step, the
full-depth vertical velocity profile is found by scaling the shape of the vertical profile (discussed
below) to the surface velocity. The vertical surface velocity is determined as the sum of the
accumulation rate and the rate of ice-sheet thinning, the latter assumed constant in time. Forced
with a time-dependent accumulation rate, the model computes the thinning function based on integration of the vertical strain over time. Starting from an assumption of constant accumulation, we iterate until the accumulation history and thinning function are consistent. While the surface vertical velocity likely does not vary at annual timescales, the resulting thinning functions are smooth, and no improvement comes from using a smoothed accumulation history.

Typical shapes of respectively "pure divide flow" and "flank flow" vertical velocity profiles at Roosevelt Island (Figure S2) were found by fitting Lliboutry-type ice-flow shape parametrizations (Lliboutry, 1979) to englacial velocities measured from repeat phase-sensitive radar measurements (Kingslake et al., 2014). However, no velocity measurements were possible in the upper 90m due to effects of firn compaction, meaning that some assumptions on the near-surface velocities were required. We therefore computed the thinning function using two different parametrizations of the velocity profile.

As discussed in the main text, the divide may have migrated in the past, and the vertical velocity profiles experienced by the ice in the core may thus have changed through time. To account for this, we follow Nereson and Waddington (2002) and describe the vertical velocity profiles as time-varying linear combinations of the divide and flank profiles. This allows a smooth variation in the vertical velocity profiles through time.

**Uncertainties**

For RICE, there are three primary sources of uncertainty in estimating the thinning function: 1) the shape of the vertical velocity profile where no measurements exist in the upper 90m, 2) the history of changes in the vertical velocity profile as the divide may have migrated, and 3) the rate of ice-sheet thickness change. Each source of uncertainty is discussed below.

The total uncertainty is estimated by calculating the thinning function using two possible parametrizations of the vertical velocity profiles, two divide-history scenarios, and three plausible rates of ice-sheet thinning, described below. The resulting accumulation histories are shown in Figure S3. We define the uncertainty as the full range of these 12 scenarios, which we interpret as a 95% confidence interval.

**The vertical velocity profile**

Following the work of Lliboutry (1979), the vertical ice-flow velocity profiles ($w$) at normalized depth ($\zeta$) can be parameterized using the vertical velocity at the surface ($w_s$) and a shape factor ($p$):

$$w(\zeta) = w_s \left( 1 - \frac{p+2}{p+1}\zeta + \frac{1}{p+1}\zeta^{p+2} \right)$$

Fitting to the vertical velocities measured for divide-like flow at Roosevelt Island (Fig. S2, red asterisks), (Kingslake et al., (2014) found the best fit with a surface velocity ($w_s$) of 0.27 m/yr and $p$ = -0.78 (Fig. S2, red dashed line). However, this fit has two limitations: i) it over-predicts the vertical velocity at mid-depths, and ii) given a recent accumulation rate of 0.24 m ice eq./yr, it implies a thinning of the ice sheet of 0.03m/yr. This amount of ice-sheet thickness change is at the upper limit of what is plausible given the observed structure of the Raymond Arch.

A more negative $p$ value, $p$ = -1.22, better matches the vertical velocity measurements, but requires a larger surface velocity, which can be excluded. As noted by (Kingslake et al. (,2014), the vertical velocity profile is near-linear in the upper part of the measurements. We therefore construct a second parametrization of the velocity profile as follows: We employ the Lliboutry fit using $p$ = -1.22, but replace the top part with a linear velocity increase towards the surface, starting at 155m ice equivalent depth (Fig. S2, solid red line). This is our preferred vertical velocity profile for divide-type flow at Roosevelt Island. The associated downwards surface velocity of 0.26m/yr corresponds to an ice-sheet thinning rate of 0.02 m/yr, which is consistent with the characteristics of the Raymond Arch.

For flank-type flow, the misfit to the measured vertical velocity measurements was minimized using $w_s$ = 0.24 m/yr and $p$ = 4.16 (Fig. S2, blue line), for which good agreement between measurements and model was obtained (Kingslake et al., 2014).

[Figure]

**Figure S2:** Vertical velocity profiles (scaled to the surface velocity, $w_s$) used for calculating the RICE thinning functions. Two shape parametrizations for divide flow were used (red lines), both derived from fitting to the measured vertical velocities (Kingslake et al., 2014). Our preferred fit (solid red line) improves the overall misfit and does not have a bias at mid-depth. For the flank-flow profile (blue), we used the fit from Kingslake et al. (2014). Also shown are vertical velocity profiles for transitional flow (black lines), calculated as a linear combination of the two profiles consisting of 70% divide flow and 30% flank flow, as appropriate for the majority of the RICE core.

**Divide migration history**
As discussed in the main text, there is ambiguity in the history of divide migration, and how it has influenced the vertical strain history at the core site. The main cause of uncertainty is the inference of maximum near-surface strain rates, i.e. the most "divide-like" flow, being slightly offset to the east of the present topographic divide (Kingslake et al., 2014; Fig 6b), where the ice core was drilled. It may therefore be that the drill site is not located at the present location of full divide-like flow. However, we note that there is considerable uncertainty in the pRES measurements of vertical strain rates, and that these do not extent to within 90m of the surface. In the near-surface layers, the peak of the Raymond Arch tilts towards to the modern summit, suggesting that the divide (and the location of maximum divide-flow) has migrated towards the current topographic summit in the most recent past. We consider this to be the most likely ice-divide migration and corresponding flow history.

In our preferred scenario with recent divide migration, the older ice in the core experienced a transitional flow regime somewhere between flank and ice-divide flow, but is now experiencing full divide flow. Starting from ~120m (~1500 CE), the amplitude of the Raymond Arch at the core site is approximately 70% of the peak amplitude. Prior to 1512 CE (500 years before the
core was drilled), we therefore used a vertical velocity profile appropriate for divide-flank
(70%/30%) transitional-type ice flow (Fig. S2; black lines). Over the following 250 years
(1512-1762 CE), the ice divide was assumed to migrate to its present position, while the vertical
velocity profile transitioned to full divide-type flow.

To account for the uncertainty in divide migration during the most recent past, the thinning
function was also calculated for the following second scenario: We assume that maximum
divide-flow has always been offset from the topographic summit, and thus that the ice in the
core has experienced entirely transitional flow.

**Ice-sheet thinning rate**
A third source of uncertainty is the amount of ice-sheet thickness change. Changes in ice-sheet
thickness will affect the vertical velocity, causing more vertical strain of layers in a thinning
ice sheet. For the deeper part of the ice core, changing the prescribed thinning rate is the most
important source of uncertainty for the accumulation rate reconstruction (Figure S3, blue lines).

We assessed the plausible range of ice-sheet thickness change both by fitting the measured
vertical velocity profiles and by modeling the amplitude of the Raymond Arch. While 2D
modeling of the dated internal stratigraphy of Roosevelt Island is on-going, we find that the
likely range of ice-sheet thickness change is 1 to 3 cm/year, with 2 cm/year being the preferred
amount.

[Figure]

**Figure S3:** Reconstructed accumulation histories according to the 12 scenarios described in the
text. The 95% confidence interval (grey area) is taken as the envelope of all accumulation
histories. Colored lines show the impact on the accumulation history from the various sources
of uncertainty, when changing one of these at a time relative to the preferred scenario (thick
black line).

**S5: Volcanic matching to WAIS Divide**

[Figure]

**Figure S4:** Volcanic matching between **a)** RICE and **b)** WAIS Divide for a recent section, including the Krakatau (1883 CE) and Makian (1861 CE) volcanic horizons. Vertical bars indicate volcanic match points (Table 2), and numbers denote the number of annual layers between match points in the two records according to their respective timescales. Mean peak height for Krakatau (white bar) does not exceed $2\sigma$ of the internal variability (mean of all available records), but is clearly visible when e.g. comparing the $Ca^{2+}$ and conductivity records directly. Green area shows the conductivity-to-calcium excess directly from the two records, with the resulting non-sea-salt conductivity record shown in the top panel.

[Figure]

**Figure S5:** Volcanic matching between **a)** RICE and **b)** WAIS Divide for the deepest part of the RICE record considered here. Vertical bars indicate volcanic match points (Table 2), and numbers denote the number of annual layers between match points in the two records according to their respective timescales. For this section, the RICE17 timescale shows a small, but distinct, bias towards counting fewer layers than present in WD2014.

**Supplementary references**

[revised manuscript text omitted]